# Breaking the Curse of Multiagency in Robust Multi-Agent Reinforcement Learning

## Abstract

Standard multi-agent reinforcement learning (MARL) algorithms are vulnerable to sim-to-real gaps. To address this, distributionally robust Markov games (RMGs) have been proposed to enhance robustness in MARL by optimizing the worst-case performance when game dynamics shift within a prescribed uncertainty set. Solving RMGs remains under-explored, from problem formulation to the development of sample-efficient algorithms. A notorious yet open challenge is if RMGs can escape the curse of multiagency, where the sample complexity scales exponentially with the number of agents. In this work, we propose a natural class of RMGs where the uncertainty set of each agent is shaped by both the environment and other agents' strategies in a best-response manner. We first establish the well-posedness of these RMGs by proving the existence of game-theoretic solutions such as robust Nash equilibria and coarse correlated equilibria (CCE). Assuming access to a generative model, we then introduce a sample-efficient algorithm for learning the CCE whose sample complexity scales polynomially with all relevant parameters. To the best of our knowledge, this is the first algorithm to break the curse of multiagency for RMGs.

## 1 Introduction

A flurry of problems naturally involve decision-making among multiple players with strategic objectives. Multi-agent reinforcement learning (MARL) serves as a powerful framework to address these challenges, demonstrating potential in various applications such as social dilemmas (Leibo et al., 2017; Baker, 2020; Zhang et al., 2024), autonomous driving (Lillicrap et al., 2015), robotics (Kober et al., 2013; Rusu et al., 2017), and games (Mnih et al., 2015; Vinyals et al., 2019). Despite the recent success of standard MARL, its transition from prototypes to reliable production is hindered by robustness concerns due to the complexity and variability of both the real-world environment and human behaviors. Specifically, environmental uncertainty can arise from sim-to-real gaps (Tobin et al., 2017), unexpected disturbance (Pinto et al., 2017), system noise, and adversarial attacks (Mahmood et al., 2018); agents' behaviors are subject to unknown bounded rationality and variability (Tversky & Kahneman, 1974). The solution learned at training time can fail catastrophically when faced with a slightly shifted MARL problem during testing, resulting in a significant drop in overall outcomes and each agent's individual payoff (Balaji et al., 2019; Zhang et al., 2020a; Zeng et al., 2022; Yeh et al., 2021; Shi et al., 2024; Slumbers et al., 2023).

To address robustness challenges, a promising framework is (distributionally) robust Markov games (RMGs) (Littman, 1994; Shapley, 1953). It is a robust counterpart to the common playground of standard MARL problems — Markov games (MGs) (Zhang et al., 2020c; Kardeş et al., 2011). In standard MGs, agents consider (competitive) personal objectives and simultaneously interact with each other within a shared unknown environment. The goal is to learn some solution concepts called equilibria, which are joint strategies/policies of agents that all of them stick with rationally with other agents fixed; for instance, Nash equilibria (NE) (Nash, 1951; Shapley, 1953), correlated equilibria (CE), and coarse correlated equilibria (CCE) (Aumann, 1987; Moulin & Vial, 1978). To promote robustness, RMGs differ from standard MGs by defining each agent's payoff (objective) as its worst-case performance when the dynamics of the game shift within a prescribed uncertainty set centered around a nominal environment.

## 1.1 THE CURSE OF MULTIAGENCY IN ROBUST MARL

Sample efficiency is a crucial metric for MARL due to the limited availability of data relative to the high dimensionality of the problem. In MARL, agents strive to learn a rationally optimal solution (equilibrium) through interactions with an unknown environment (Silver et al., 2016; Vinyals et al., 2019; Achiam et al., 2023). In contemporary applications, the environment is often extremely large-scale, while data acquisition can be prohibitively limited by high costs and stakes. As such, a notable challenge in terms of scalability for sample efficiency in MARL is known as *the curse of multiagency* — the sample complexity requirement scales exponentially with the number of agents (induced by the exponentially growing size of the joint action space). This issue has been recognized and studied in extensive MARL problems, but remains open for robust MARL. We concentrate on learning finite-horizon multi-player general-sum Markov games with a generative model (Kearns & Singh, 1999), where the number of agents is $n$, the episode length is $H$, the size of the state space is $S$, and the size of the $i$-th agent's action space is $A_i$, for $1 \leq i \leq n$.

• *Breaking the curse of multiagency in standard MARL.* A line of pioneering work (Jin et al., 2021; Bai & Jin, 2020; Song et al., 2021; Li et al., 2023) has recently introduced a new suite of algorithms using adaptive sampling that provably break the curse of multiagency in standard MGs. In particular, to find an $\varepsilon$-approximate CCE, Li et al. (2023) requires a sample complexity no more than

$$\widetilde{O}\left(\frac{H^4 S \sum_{i=1}^n A_i}{\varepsilon^2}\right) \tag{1}$$

up to logarithmic factors, which depends only on the sum of individual actions, rather than the number of joint actions.

• *The persistent curse of multiagency in robust MARL.* The development of provable sample-efficient algorithms for RMGs is largely underexplored, with only a few recent studies (Zhang et al., 2020c; Kardeş et al., 2011; Ma et al., 2023; Blanchet et al., 2023; Shi et al., 2024). Focusing on a class of RMGs with uncertainty sets satisfying the $(s, \boldsymbol{a})$-*rectangularity condition*, existing works all suffer from the curse of multiagency, significantly limiting their scalability. For example, using the total variation (TV) distance as the divergence function, the state-of-the-art (Shi et al., 2024), using non-adaptive sampling, finds an $\varepsilon$-approximate robust CCE with a sample complexity no more than

$$\widetilde{O}\left(\frac{H^3 S \prod_{i=1}^n A_i}{\varepsilon^2} \min\left\{H, \frac{1}{\min_{1 \leq i \leq n} \sigma_i}\right\}\right) \tag{2}$$

up to logarithmic factors, where $\sigma_i \in [0, 1)$ is the uncertainty level for the $i$-th agent. As a result, the sample size requirement becomes prohibitive when the number of agents is large.

Consequently, there is a significant desire to explore paths that could break through the curse of multiagency in RMGs, which is much more involved than its standard counterpart due to complicated non-linearity introduced by planning for worst-case performances. Nevertheless, the family of RMGs is a much richer class of problems because of the flexibility in choosing the uncertainty sets to capture different robust design considerations. While convenient, the $(s, \boldsymbol{a})$-*rectangularity condition* prevalent in current approaches can be overly restricted in practice, as each agent's uncertainty set is assumed to be independent of other agents' strategies and can be decoupled into independent subsets for each state-joint action pair $(s, \boldsymbol{a})$, suggesting it might be challenging to break the curse of multiagency in the existing framework. Given these limitations, we are motivated to develop new classes of RMGs that can provide robust solutions applicable to more realistic MARL problems with sample-efficient algorithms. This raises an open question:

*Can we design RMGs with practically-meaningful uncertainty sets that come with sample complexity guarantees breaking the curse of multiagency?*

## 1.2 CONTRIBUTIONS

We propose a new class of RMGs with a *fictitious* uncertainty set that explicitly captures uncertainties in the environment in view of other agents' strategies, making it suitable for complex real-world scenarios. We begin by verifying the game-theoretic properties of the proposed class of RMGs to ensure the existence of robust variants of well-known standard equilibria notions, robust NE and robust CCE. Next, due to the general intractability of learning NE, we focus on designing algorithms

| Algorithm | Uncertainty set | Equilibria | Sample complexity |
|---|---|---|---|
| P$^2$MPO (Blanchet et al., 2024) | $(s, \boldsymbol{a})$-rectangularity | robust NE | $S^4 \left(\prod_{i=1}^n A_i\right)^3 H^4/\varepsilon^2$ |
| DR-NVI (Shi et al., 2024) | $(s, \boldsymbol{a})$-rectangularity | robust NE/CE/CCE | $\frac{SH^3 \prod_{i=1}^n A_i}{\varepsilon^2} \min\left\{H, \frac{1}{\min_{1 \leq i \leq n} \sigma_i}\right\}$ |
| Robust-Q-FTRL **(this work)** | fictitious $(s, a_i)$-rectangularity | robust CCE | $\frac{SH^6 \sum_{1 \leq i \leq n} A_i}{\varepsilon^4} \min\left\{H, \frac{1}{\min_{1 \leq i \leq n} \sigma_i}\right\}$ |

Table 1: We compare our results with prior work on finding an $\varepsilon$-approximate equilibrium in finite-horizon multi-agent general-sum robust MG, omitting logarithmic factors in the sample complexities. Our result is the only computationally tractable algorithm that breaks the curse of multiagency.

that can provably overcome the curse of multiagency in learning an approximate robust CCE, referring to a joint policy where no agent can improve their benefit by more than $\varepsilon$ through rational deviations.. Specifically, for sampling mechanisms to explore the unknown environment, we assume access to a generative model that can only draw samples from the nominal environment (Shi et al., 2024). The main contributions are summarized as follows.

• We introduce a new class of robust Markov games using fictitious uncertainty sets with *policy-induced $(s, a_i)$-rectangularity condition* (see Section 2.2 for details), which is a natural adaptation from robust single-agent RL to robust MARL. The uncertainty set for each agent $i$ can be decomposed into independent subsets over each state and its own action tuple $(s, a_i)$, where each subset is a "ball" around the expected nominal transition determined by other agents' policies and the nominal transition kernel, a divergence function $\rho$, and the radius/uncertainty level $\sigma_i$. We verify several essential facts of this class of RMGs: the existence of the desired equilibrium — robust NE and robust CCE for this new class of RMGs using game-theoretical tools such as fixed-point theorem; the existence of best-response policies and robust Bellman equations.

• We consider the total variation (TV) distance as the divergence function $\rho$ for uncertainty sets due to its popularity in both practice (Pan et al., 2023; Lee et al., 2021; Szita et al., 2003) and theory (Panaganti & Kalathil, 2022; Blanchet et al., 2023; Shi et al., 2024). We propose Robust-Q-FTRL that can find $\varepsilon$-approximate robust CCE with high probability, as long as the sample size exceeds

$$\widetilde{O}\left(\frac{SH^6 \sum_{i=1}^n A_i}{\varepsilon^4} \min\left\{H, \frac{1}{\min_{1 \leq i \leq n} \sigma_i}\right\}\right) \qquad (3)$$

up to logarithmic factors. To the best of our knowledge, this is the first algorithm to break the curse of multiagency in RMGs. It provably finds an $\varepsilon$-approximate robust CCE using a sample size that is polynomial to all salient parameters. Table 1 provides a detailed comparison to prior works in robust MARL, where our results significantly improve upon prior art (2) (Shi et al., 2024) by reducing the exponential dependency on the size of each agent's action space to a linear dependency. To achieve this, we utilize adaptive sampling and online adversarial learning tools, coupled by a tailored design and analysis for robust MARL due to the nonlinearity of the robust value function, which contrasts with the linear payoff functions in standard MARL with respect to the transition kernel.

**Notation.** In this paper, we use the notation $[T] := 1, 2, \ldots, T$ for any positive integer $T > 0$, and $\Delta(S)$ for the simplex over the set $S$. For any policy $\pi$ and function $Q(\cdot)$ defined over a domain $B$, the variance of $Q$ under $\pi$ is given by $\mathsf{Var}_\pi(Q) := \sum_{a \in B} \pi(a)[Q(a) - \mathbb{E}_\pi[Q]]^2$. We define $x = [x(s, \boldsymbol{a})]_{(s,\boldsymbol{a}) \in S \times A} \in \mathbb{R}^{SA}$ as any vector that represents values for each state-action pair, and $x = [x(s, a_i)]_{(s,a_i) \in S \times A_i} \in \mathbb{R}^{SA_i}$ as any vector representing agent-wise state-action values. Similarly, we denote $x = [x(s)]_{s \in S}$ as any vector representing values for each state. For $\mathcal{X} := (S, \{A_i\}_{i \in [n]}, H, \{\sigma_i\}_{i \in [n]}, \frac{1}{\varepsilon}, \frac{1}{\delta})$, let $f(\mathcal{X}) = O(g(\mathcal{X}))$ denote that there exists a constant $C_1 > 0$ such that $f \leq C_1 g$, with $\widetilde{O}(\cdot)$ similarly defined but omitting logarithmic factors.

## 2 PRELIMINARIES

In this section, we begin with some background on multi-agent general-sum standard Markov games (MGs) in finite-horizon settings, followed by a general framework of a robust variant of standard MGs —- distributionally robust Markov games.

## 2.1 STANDARD MARKOV GAMES

A finite-horizon *multi-agent general-sum Markov game* (MG) is characterized by the tuple $\mathcal{MG} = \{\mathcal{S}, \{\mathcal{A}_i\}_{1 \le i \le n}, P, r, H\}$. This setup features $n$ agents each striving to maximize their individual long-term cumulative rewards within a shared environment. At each time step, all agents observe the same state over the state space $\mathcal{S} = \{1, \cdots, S\}$ within the shared environment. For each agent $i$ ($i \in [n]$), $\mathcal{A}_i = \{1, \cdots, A_i\}$ denotes its action space containing $A_i$ possible actions. The joint action space for all agents (resp. the subset excluding the $i$-th agent) is defined as $\mathcal{A} := \mathcal{A}_1 \times \cdots \times \mathcal{A}_n$ (resp. $\mathcal{A}_{-i} := \prod_{j \ne i} \mathcal{A}_j$ for any $i \in [n]$). We use the notation $\boldsymbol{a} \in \mathcal{A}$ (resp. $\boldsymbol{a}_{-i} \in \mathcal{A}_{-i}$) to denote a joint action profile involving all agents (resp. all except the $i$-th agent). In addition, the probability transition kernel $P = \{P_h\}_{1 \le h \le H}$, with each $P_h : \mathcal{S} \times \mathcal{A} \mapsto \Delta(\mathcal{S})$, describes the dynamics of the game: $P_h(s' \,|\, s, \boldsymbol{a})$ is the probability of transitioning from state $s \in \mathcal{S}$ to state $s' \in \mathcal{S}$ at time step $h$ when agents choose the joint action profile $\boldsymbol{a} \in \mathcal{A}$. The reward function of the game is $r = \{r_{i,h}\}_{1 \le i \le n, 1 \le h \le H}$, with each $r_{i,h} : \mathcal{S} \times \mathcal{A} \mapsto [0, 1]$ normalized to the unit interval. For any $(i, h, s, \boldsymbol{a}) \in [n] \times [H] \times \mathcal{S} \times \mathcal{A}$, $r_{i,h}(s, \boldsymbol{a})$ represents the immediate reward received by the $i$-th agent in state $s$ when the joint action profile $\boldsymbol{a}$ is taken. Last but not least, $H > 0$ represents the horizon length.

**Markov policies and value functions.** In this work, we concentrate on Markov policies that the action selection rule depends only on the current state $s$, independent from previous trajectory. Namely, the $i$-th ($i \in [n]$) agent chooses actions according to $\pi_i = \{\pi_{i,h} : \mathcal{S} \mapsto \Delta(\mathcal{A}_i)\}_{1 \le h \le H}$. Here, $\pi_{i,h}(a \,|\, s)$ represents the probability of selecting action $a \in \mathcal{A}_i$ in state $s$ at time step $h$. As such, the joint Markov policy of all agents can be denoted as $\pi = (\pi_1, \ldots, \pi_n) : \mathcal{S} \times [H] \mapsto \Delta(\mathcal{A})$, i.e., given any $s \in \mathcal{S}$ and $h \in [H]$, the joint action profile $\boldsymbol{a} \in \mathcal{A}$ of all agents is chosen following the distribution $\pi_h(\cdot \,|\, s) = (\pi_{1,h}, \pi_{2,h} \ldots, \pi_{n,h})(\cdot \,|\, s) \in \Delta(\mathcal{A})$.

To continue, for any given joint policy $\pi$ and transition kernel $P$ of a $\mathcal{MG}$, the $i$-th agent's long-term cumulative reward can be characterized by the value function $V_{i,h}^{\pi,P} : \mathcal{S} \mapsto \mathbb{R}$ (resp. Q-function $Q_{i,h}^{\pi,P} : \mathcal{S} \times \mathcal{A} \mapsto \mathbb{R}$) as below: for all $(h, s, a) \in [H] \times \mathcal{S} \times \mathcal{A}$,

$$V_{i,h}^{\pi,P}(s) := \mathbb{E}_{\pi,P}\Big[\sum_{t=h}^{H} r_{i,t}(s_t, \boldsymbol{a}_t) \,|\, s_h = s\Big], \quad Q_{i,h}^{\pi,P}(s, \boldsymbol{a}) := \mathbb{E}_{\pi,P}\Big[\sum_{t=h}^{H} r_{i,t}(s_t, \boldsymbol{a}_t) \,|\, s_h = s, \boldsymbol{a}_h = \boldsymbol{a}\Big].$$
(4)

In this context, the expectation is calculated over the trajectory $\{(s_t, \boldsymbol{a}_t)\}_{h \le t \le H}$ produced by following the joint policy $\pi$ under the transition kernel $P$.

## 2.2 DISTRIBUTIONALLY ROBUST MARKOV GAMES

A general distributionally robust Markov game (RMG) is represented by the tuple

$$\mathcal{RMG} = \big\{\mathcal{S}, \{\mathcal{A}_i\}_{1 \le i \le n}, \{\mathcal{U}_\rho^{\sigma_i}(P^0)\}_{1 \le i \le n}, r, H\big\}.$$

Here, $\mathcal{S}, \{\mathcal{A}_i\}_{1 \le i \le n}, r, H$ are defined in the same manner as those in standard MGs (see Section 2.1). RMGs differ from standard MGs: for each agent $i$ ($1 \le i \le n$), the transition kernel is not fixed but can vary within its own prescribed uncertainty set $\mathcal{U}_\rho^{\sigma_i}(P^0)$ centered around some *nominal* kernel $P^0 : \mathcal{S} \times \mathcal{A} \mapsto \Delta(\mathcal{S})$ that represents a reference (such as the training environment). The shape and the size of the uncertainty set $\{\mathcal{U}_\rho^{\sigma_i}(P^0)\}_{i \in [n]}$ are further specified by a divergence function $\rho$ and the uncertainty levels $\{\sigma_i\}_{i \in [n]}$, serving as the "distance" metric and the radius respectively.

Various choices of the divergence function have been considered in robust RL literature, including but not limited to $f$-divergence (such as total variation, $\chi^2$ divergence, and Kullback-Leibler (KL) divergence) (Yang et al., 2022; Zhou et al., 2021; Lu et al., 2024; Wang et al., 2024) and Wasserstein distance (Xu et al., 2023). Different uncertainty sets lead to distinct RMGs, as they address distinct types of uncertainty and game-theoretical solutions. This paper focuses on variability in environmental dynamics (transition kernels), though uncertainty in agents' reward functions could also be considered similarly but is omitted for brevity.

**Robust value functions and best-response policies.** For any RMG, each agent seeks to maximize its worst-case performance in the presence of other agents' behaviors despite perturbations in the environment dynamics, as long as the kernel transitions remain within its prescribed uncertainty set $\mathcal{U}_\rho^{\sigma_i}(P^0)$. Mathematically, given any joint policy $\pi : \mathcal{S} \times [H] \mapsto \Delta(\mathcal{A})$, the worst-case performance

of any agent $i$ is characterized by the *robust value function* $V_{i,h}^{\pi,\sigma_i}$ and the *robust Q-function* $Q_{i,h}^{\pi,\sigma_i}$: for all $(i,h,s,a_i) \in [n] \times [H] \times \mathcal{S} \times \mathcal{A}_i$,

$$V_{i,h}^{\pi,\sigma_i}(s) := \inf_{P \in \mathcal{U}_\rho^{\sigma_i}(P^0)} V_{i,h}^{\pi,P}(s) \qquad \text{and} \qquad Q_{i,h}^{\pi,\sigma_i}(s,a_i) := \inf_{P \in \mathcal{U}_\rho^{\sigma_i}(P^0)} Q_{i,h}^{\pi,P}(s,a_i). \tag{5}$$

Note that different from (4), here the Q-function for any $i$-th agent is defined only over its own action $a_i \in \mathcal{A}_i$ rather than the joint action $\boldsymbol{a} \in \mathcal{A}$.

To continue, we denote $\pi_{-i}$ as the policy for all agents except for the $i$-th agent. By optimizing the $i$-th agent's policy $\pi_i' : \mathcal{S} \times [H] \to \Delta(\mathcal{A}_i)$ (independent from $\pi_{-i}$), we define the maximum of the robust value function as

$$V_{i,h}^{\star,\pi_{-i},\sigma_i}(s) := \max_{\pi_i':\mathcal{S}\times[H]\mapsto\Delta(\mathcal{A}_i)} V_{i,h}^{\pi_i'\times\pi_{-i},\sigma_i}(s) = \max_{\pi_i':\mathcal{S}\times[H]\mapsto\Delta(\mathcal{A}_i)} \inf_{P \in \mathcal{U}_\rho^{\sigma_i}(P^0)} V_{i,h}^{\pi_i'\times\pi_{-i},P}(s) \tag{6}$$

for all $(i,h,s) \in [n] \times [H] \times \mathcal{S}$. The policy that achieves the maximum of the robust value function for all $(i,h,s) \in [n] \times [H] \times \mathcal{S}$ is called a *robust best-response policy*.

**Solution concepts for robust Markov games.** In view of the conflicting objectives between agents, establishing equilibrium becomes the goal of solving RMGs. As such, we introduce two kinds of solution concepts — robust NE and robust CCE — robust variants of standard NE and CCE (usually considered in standard MGs) specified to the form of RMGs.

• *Robust NE.* A product policy $\pi = \pi_1 \times \pi_2 \times \cdots \times \pi_n : \mathcal{S} \times [H] \mapsto \prod_{i=1}^n \Delta(\mathcal{A}_i)$ is said to be a *robust NE* if

$$V_{i,1}^{\pi,\sigma_i}(s) = V_{i,1}^{\star,\pi_{-i},\sigma_i}(s), \quad \forall(s,i) \in \mathcal{S} \times [n]. \tag{7}$$

Given the strategies of the other agents $\pi_{-i}$, when each agent wants to optimize its worst-case performance when the environment and other agents' policy stay within its own uncertainty set $\mathcal{U}_\rho^{\sigma_i}(P^0)$, robust NE means that no player can benefit by unilaterally diverging from its present strategy.

• *Robust CCE.* A distribution over the joint product policy $\xi := \{\xi_h\}_{h\in[H]} : [H] \mapsto \Delta(\mathcal{S} \mapsto \prod_{i\in[n]} \Delta(\mathcal{A}_i))$ is said to be a *robust CCE* if it holds that

$$\mathbb{E}_{\pi\sim\xi}\left[V_{i,1}^{\pi,\sigma_i}(s)\right] \geq \mathbb{E}_{\pi\sim\xi}\left[V_{i,1}^{\star,\pi_{-i},\sigma_i}(s)\right], \quad \forall(i,s) \in [n] \times \mathcal{S}. \tag{8}$$

Considering all agents follow the policy drawn from the distribution $\xi$, i.e., $\pi_h \sim \xi_h$ for all $h \in [H]$, when the distribution of all agents but the $i$-th agent's policy is fixed as the marginal distribution of $\xi$, robust CCE indicates that no agent can benefit from deviating from its current policy.

Note that, for standard MGs, CCE is defined as a possibly correlated joint policy $\pi^{\mathsf{CCE}} : \mathcal{S} \times [H] \mapsto \Delta(\mathcal{A})$ (Moulin & Vial, 1978; Aumann, 1987) if it holds that

$$V_{i,1}^{\pi^{\mathsf{CCE}},P}(s) \geq \max_{\pi_i':\mathcal{S}\times[H]\to\Delta(\mathcal{A}_i)} V_{i,1}^{\pi_i'\times\pi_{-i}^{\mathsf{CCE}},P}(s), \qquad \forall(s,i) \in \mathcal{S} \times [n]. \tag{9}$$

This correlated policy $\pi^{\mathsf{CCE}}$ can also be viewed as a distribution $\xi$ over the product policy space since each joint action $\boldsymbol{a}$ can be seen as a deterministic product policy. Careful readers may note that the definition (9) of CCE in standard MGs is in a different form from the one (8) in RMGs, as the latter does not include the expectation operator $\mathbb{E}_{\pi\sim\xi}[\cdot]$ with respect to the policy distribution ($\xi$) over the value function. We emphasize that the definition with the expectation operator outside of the value (or cost) function with respect to a distribution of product pure strategies in (8) is a natural formulation originating from game theory (Moulin et al., 2014; Moulin & Vial, 1978). In standard MARL and previous robust MARL studies, the definition in (9) is typically used because (9) and (8) are identical in those situations, as the expectation operator and the corresponding value functions are linear with respect to the joint policy, allowing them to be interchanged (Li et al., 2023; Shi et al., 2024).

# 3 ROBUST MARKOV GAMES WITH FICTITIOUS UNCERTAINTY SETS

Given the definition of general RMGs, a natural question arises: what kinds of uncertainty sets should we consider to achieve the desired robustness in our solutions? To address this, we focus on a specific class of RMGs characterized by a type of natural yet powerful uncertainty sets.

### 3.1 A NOVEL UNCERTAINTY SET DEFINITION IN RMGs

We propose a new class of uncertainty sets, named *fictitious* uncertainty sets, which count in the uncertainty induced by both the environment and agents' behaviors in a correlated manner. Before introducing the uncertainty sets, we provide some auxiliary notations as below. We denote a vector of any transition kernel $P : \mathcal{S} \times \mathcal{A} \mapsto \Delta(\mathcal{S})$ or $P^0 : \mathcal{S} \times \mathcal{A} \mapsto \Delta(\mathcal{S})$ respectively as $P_{h,s,\boldsymbol{a}} :=$ $P_h(\cdot \,|\, s, \boldsymbol{a}) \in \mathbb{R}^{1 \times S}, P^0_{h,s,\boldsymbol{a}} := P^0_h(\cdot \,|\, s, \boldsymbol{a}) \in \mathbb{R}^{1 \times S}$, for all $(s, \boldsymbol{a}) \in \mathcal{S} \times \mathcal{A}$. Then for any (possibly correlated) joint Markov policy (defined in section 2.1) $\pi : \mathcal{S} \times [H] \mapsto \Delta(\mathcal{A})$, we define the expected nominal transition kernel conditioned on the situation that the $i$-th agent chooses some action $a_i \in \mathcal{A}_i$ and other agents play according to the conditional policy (i.e., $\boldsymbol{a}_{-i} \sim \pi_h(\cdot \,|\, s, a_i)$) given $s \in \mathcal{S}$ and $a_i$ as below: for all $(h, s, a_i) \in [H] \times \mathcal{S} \times \mathcal{A}_i$,

$$P^{\pi_{-i}}_{h,s,a_i} = \mathbb{E}_{\boldsymbol{a} \sim \pi_h(\cdot \,|\, s, a_i)} \left[ P^0_{h,s,\boldsymbol{a}} \right] = \sum_{\boldsymbol{a}_{-i} \in \mathcal{A}_{-i}} \frac{\pi_h(a_i, \boldsymbol{a}_{-i} \,|\, s)}{\pi_{i,h}(a_i \,|\, s)} \left[ P^0_{h,s,\boldsymbol{a}} \right]. \tag{10}$$

Armed with the above definitions, now we are in a position to define the *fictitious* uncertainty sets, denoted as $\left\{ \mathcal{U}^{\sigma_i}_\rho(P^0, \cdot) \right\}_{i \in [n]}$, which satisfy a *policy-induced $(s, a_i)$-rectangularity condition*.

**Definition 1.** *For any joint policy $\pi : \mathcal{S} \times [H] \mapsto \Delta(\mathcal{A})$, divergence function $\rho : \Delta(\mathcal{S}) \times \Delta(\mathcal{S}) \mapsto \mathbb{R}^+$ and accessible uncertainty levels $\sigma_i \geq 0$ for all $i \in [n]$, the fictitious uncertainty sets $\left\{ \mathcal{U}^{\sigma_i}_\rho(P^0, \pi) \right\}_{i \in [n]}$ satisfy the* policy-induced $(s, a_i)$-rectangularity *condition: for all $i \in [n]$ and $\forall (h, s, a_i) \in [H] \times \mathcal{S} \times \mathcal{A}_i$,*

$$\mathcal{U}^{\sigma_i}_\rho(P^0, \pi) := \otimes \, \mathcal{U}^{\sigma_i}_\rho \left( P^{\pi_{-i}}_{h,s,a_i} \right), \quad \mathcal{U}^{\sigma_i}_\rho \left( P^{\pi_{-i}}_{h,s,a_i} \right) := \left\{ P \in \Delta(\mathcal{S}) : \rho \left( P, P^{\pi_{-i}}_{h,s,a_i} \right) \leq \sigma_i \right\}, \tag{11}$$

*where $\otimes$ represents the Cartesian product.*

In words, conditioned on a fixed joint policy $\pi$, the uncertainty set $\mathcal{U}^{\sigma_i}_\rho(P^0, \pi)$ for each $i$-th agent can be decomposed into a Cartesian product of subsets over each state and agent-action pair $(s, a_i)$. Each uncertainty subset $\mathcal{U}^{\sigma_i}_\rho(P^{\pi_{-i}}_{h,s,a_i})$ over $(s, a_i)$ is defined as a "ball" around a reference — the expected nominal transition kernel $P^{\pi_{-i}}_{h,s,a_i}$ conditioned on both transition kernel and agents' behavior $\pi$.

**Further discussions of fictitious uncertainty sets.** It is in order to remark on the proposed type of uncertainty sets, in comparison with prior works.

• *A natural adaptation from single-agent robust RL.* When agents follow some joint policy $\pi : \mathcal{S} \times [H] \mapsto \Delta(\mathcal{A})$, fixing other agents' policy $\pi_{-i}$, from the perspective of each individual agent $i$, RMGs with our policy-induced $(s, a_i)$-rectangularity condition will degrade to a single-agent robust RL problem with the widely used $(s, a_i)$-rectangularity condition in the single-agent literature (Iyengar, 2005; Zhou et al., 2021). Namely, from any agent $i$'s viewpoint, in a RMG, it has an "overall environment" player that can not only manipulate the environmental dynamics but also other players' policy $\pi_{-i}$.

• *Allowing uncertainty from both the environment and agents' behaviors in a correlated manner.* One essential feature of our proposed uncertainty set is that it is shaped by both the environment and agents' strategies in a (possibly) correlated manner. Specifically, for any agent $i$ and a given policy $\pi$, any uncertainty subset $\mathcal{U}^{\sigma_i}_\rho \left( P^{\pi_{-i}}_{h,s,a_i} \right)$ (over any $(h, s, a_i)$) is constructed as a neighborhood around a nominal center $P^{\pi_{-i}}_{h,s,a_i}$ (see (10)) that depends on both the nominal environment $P^0$ and other agents' conditional strategies $\pi_h(\cdot \,|\, s, a_i)$.

• *Comparisons to prior works.* Prior works on provable sample-efficient algorithms have focused on a different type of uncertainty sets with $(s, \boldsymbol{a})$-*rectangularity condition* (Ma et al., 2023; Blanchet et al., 2023; Shi et al., 2024). This class of uncertainty sets decouples the uncertainty into independent subsets for each state-joint action pair $(s, \boldsymbol{a})$, accounting for the uncertainty of the environment and agents' strategies independently. In comparison, the proposed uncertainty set lifts this independence assumption across subsets over different $(s, a_i, \boldsymbol{a}_{-i})$ for any $\boldsymbol{a}_{-i} \in \mathcal{A}_i$, enabling the environment and agents' strategies to shape the uncertainty set in a correlated manner.

### 3.2 PROPERTIES OF RMGs WITH FICTITIOUS UNCERTAINTY SETS

Throughout the paper, we focus on the class of RMGs with the proposed fictitious uncertainty sets, represented as

$$\mathcal{RMG}_\pi = \big\{ \mathcal{S}, \{\mathcal{A}_i\}_{1 \leq i \leq n}, \{\mathcal{U}_\rho^{\sigma_i}(P^0, \cdot)\}_{1 \leq i \leq n}, r, H \big\}$$

and abbreviated as fictitious RMGs in the remaining of the paper. In this section, we present key facts about fictitious RMGs related to best-response policies, equilibria, and the corresponding one-step lookahead robust Bellman equations. The proofs are postponed to Appendix C.

First, we introduce the following lemma, which verifies the existence of a robust best-response policy that achieves the maximum robust value function (cf. (6)).

**Lemma 1.** *For any $i \in [n]$, given $\pi_{-i} : \mathcal{S} \times [H] \mapsto \Delta(\mathcal{A}_i)$, there exists at least one policy $\widetilde{\pi}_i : \mathcal{S} \times [H] \to \Delta(\mathcal{A}_i)$ for the $i$-th agent that can simultaneously attain $V_{i,h}^{\widetilde{\pi}_i \times \pi_{-i}, \sigma_i}(s) = V_{i,h}^{\star, \pi_{-i}, \sigma_i}(s)$ for all $s \in \mathcal{S}$ and $h \in [H]$. We refer this policy as the* robust best-response policy.

**Existence of robust NE and robust CCE.** fictitious RMGs can be viewed as hierarchical games with $n + nS \sum_{i=1}^n A_i$ agents. This includes the original $n$ agents and $n$ additional sets of $S \sum_{i=1}^{\widetilde{n}} A_i$ independent adversaries, each determining the worst-case transitions for one agent over a state plus agent-wise-action pair. Considering the solution concepts — robust NE and robust CCE — introduced in Section 2.2, the following theorem verifies the existence of them for any fictitious RMGs using Kakutani's fixed-point theorem (Kakutani, 1941), focusing on robust NE firstly.

**Theorem 1** (Existence of robust NE). *For any $\mathcal{RMG}_\pi = \big\{ \mathcal{S}, \{\mathcal{A}_i\}_{1 \leq i \leq n}, \{\mathcal{U}_\rho^{\sigma_i}(P^0, \cdot)\}_{1 \leq i \leq n}, r, H \big\}$ with an uncertainty set defined in Definition 1, there exists at least one robust NE.*

Analogous to standard Markov games, since {robust NE} $\subseteq$ {robust CCE}, Theorem 1 indicates the existence of robust CCEs directly.

Fortunately, the class of fictitious RMGs feature a robust counterpart of the Bellman equation — *robust Bellman equation*, which is detailed in Appendix B.2.

## 4 SAMPLE-EFFICIENT LEARNING: ALGORITHM AND THEORY

In this section, we focus on designing sample-efficient algorithms for solving fictitious RMGs when agents need to collect data by interacting with the unknown shared environment in order to learn the equilibria. To proceed, we shall first specify the data collection mechanism and the divergence function for the uncertainty set. Then we propose a sample-efficient algorithm Robust-Q-FTRL that leverages a carefully-designed adaptive sampling strategy to break the curse of multiagency.

### 4.1 PROBLEM SETTING AND GOAL

Recall that the uncertainty sets are constructed by specifying a divergence function $\rho$ and the uncertainty level to control its shape and size. In this work, we focus on using the TV distance as the divergence function $\rho$ for the uncertainty set, following Szita et al. (2003); Lee et al. (2021); Pan et al. (2023), defined by

$$\forall P, P' \in \Delta(\mathcal{S}) : \quad \rho_{\mathsf{TV}}(P, P') \coloneqq \frac{1}{2} \|P - P'\|_1. \tag{12}$$

For convenience, throughout the paper, we abbreviate $\mathcal{U}^{\sigma_i}(\cdot) \coloneqq \mathcal{U}_{\rho_{\mathsf{TV}}}^{\sigma_i}(\cdot)$ when there is no ambiguity.
**Data collection mechanism: a generative model.** We assume the agents interact with the environment through a generative model (simulator) (Kearns & Singh, 1999), which is a widely used sampling mechanism in both single-agent RL and MARL (Zhang et al., 2020b; Li et al., 2022). Specifically, at any time step $h$, we can collect an arbitrary number of independent samples from any state and joint action tuple $(s, \boldsymbol{a}) \in \mathcal{S} \times \mathcal{A}$, generated based on the true *nominal* transition kernel $P^0$: for $i = 1, 2, \ldots, s_{h,s,\boldsymbol{a}}^i \overset{i.i.d}{\sim} P_h^0(\cdot \,|\, s, \boldsymbol{a})$.

**Goal.** Consider any fictitious RMGs $\mathcal{RMG}_\pi = \big\{ \mathcal{S}, \{\mathcal{A}_i\}_{1 \leq i \leq n}, \{\mathcal{U}^{\sigma_i}(P^0, \cdot)\}_{1 \leq i \leq n}, r, H \big\}$. While learning exact robust equilibria is computationally challenging and may not be necessary in practice, instead in this work, we focus on finding an approximate robust CCE (defined in (8)). Namely, a

distribution $\xi := \{\xi_h\}_{h \in [H]} : [H] \mapsto \Delta(\mathcal{S} \mapsto \prod_{i \in [n]} \Delta(\mathcal{A}_i))$ is said to be an $\varepsilon$-*robust CCE* if

$$\mathsf{gap}_{\mathsf{CCE}}(\xi) := \max_{s \in \mathcal{S}, 1 \le i \le n} \left\{ \mathbb{E}_{\pi \sim \xi} \left[ V_{i,1}^{\star, \pi_{-i}, \sigma_i}(s) \right] - \mathbb{E}_{\pi \sim \xi} \left[ V_{i,1}^{\pi, \sigma_i}(s) \right] \right\} \le \varepsilon. \tag{13}$$

Armed with a generative model of the nominal environment, the goal is to learn a robust CCE using as few samples as possible.

### 4.2 ALGORITHM DESIGN

With the sampling mechanism over a generative model in hand, we propose an algorithm called Robust-Q-FTRL to learn an $\varepsilon$-*robust CCE* in a sample-efficient manner, summarized in Algorithm 2 in the appendix. Robust-Q-FTRL draws inspiration from Q-FTRL developed in the standard MG literature (Li et al., 2022), but empowers tailored designs for learning in fictitious RMGs to achieve a robust equilibrium and to tackle statistical challenges arising from agents' nonlinear objectives. Overall, Robust-Q-FTRL takes a single pass to learn recursively from the final time step $h = H$ to $h = 1$. At each time step $h \in [H]$, an online learning process with $K$ iterations will be executed. Before introducing the algorithm, we first concentrate on two essential steps customized for learning in fictitious RMGs.

**Constructing the empirical model via $N$-sample estimation.** For each time step $h$, we denote $\pi_{i,h}^k$ as the current learning policy of the $i$-th agent before the beginning of the $k$-th iteration for any $k \in [K]$. And we denote the joint product policy as $\pi_h^k = (\pi_{1,h}^k, \cdots, \pi_{n,h}^k)$. During each iteration $k$, for each agent $i \in [n]$, we require to generate $N$ independent samples from the generative model over each $(s, a_i) \in \mathcal{S} \times \mathcal{A}_i$ to obtain an empirical model, detailed in Algorithm 1. It includes an empirical reward function represented by $r_{i,h}^k \in \mathbb{R}^{SA_i}$ and transition kernels denoted by $P_{i,h}^k \in \mathbb{R}^{SA_i \times S}$. Note that different from standard MGs, we need to generate $N$ samples instead of 1 sample per iteration to handle the additional statistical challenges induced by the non-linear objective of agents ($N$ will be specified momentarily).

**Estimating robust Q-function of the current policy $\pi_h^k$.** We denote $\widehat{V}_{i,h} \in \mathbb{R}^S$ as the estimation of the $i$-th agent's robust value function at time step $h$. For any agent $i$, with the empirical reward function $r_{i,h}^k$, empirical kernel $P_{i,h}^k$, and the estimated robust value function $\widehat{V}_{i,h+1}$ at the next step in hand, the robust Q-function $\{q_{i,h}^k\}$ of current policy $\pi_h^k$ can be estimated as:

$$\forall (i, h, s, a_i) \in [n] \times [H] \times \mathcal{S} \times A_i : \quad q_{i,h}^k(s, a_i) = r_{i,h}^k(s, a_i) + \inf_{\mathcal{P} \in \mathcal{U}^{\sigma_i}(P_{i,h,s,a_i}^k)} \mathcal{P}\widehat{V}_{i,h+1}. \tag{14}$$

Unlike the linear function w.r.t. $P_{i,h}^k$ in standard MGs, (14) lacks a closed form and introduces an additional inner optimization problem. Solving (14) directly is computationally challenging due to the need to optimize over an $S$-dimensional probability simplex, with complexity growing exponentially with the state space size $S$. Fortunately, by applying strong duality, we can solve (14) equivalently via its dual problem with tractable computation (Iyengar, 2005):

$$
\begin{aligned}
q_{i,h}^k(s, a_i) = \; & r_{i,h}^k(s, a_i) \\
& + \max_{\alpha \in [\min_s \widehat{V}_{i,h+1}(s), \max_s \widehat{V}_{i,h+1}(s)]} \left\{ P_{i,h}^k \big[ \widehat{V}_{i,h+1} \big]_\alpha - \sigma_i \big( \alpha - \min_{s'} \big[ \widehat{V}_{i,h+1} \big]_\alpha(s') \big) \right\}, \tag{15}
\end{aligned}
$$

where $[V]_\alpha$ denotes the clipped version of any vector $V \in \mathbb{R}^S$ determined by some level $\alpha \ge 0$, namely, $[V]_\alpha(s) := \begin{cases} \alpha, & \text{if } V(s) > \alpha, \\ V(s), & \text{otherwise.} \end{cases}$ This is a key component of Robust-Q-FTRL, serving for constructing nonlinear robust objectives in the online learning process and ensuring the desired statistical accuracy.

**Overall pipeline of Robust-Q-FTRL.** With these technical modules in place, we introduce Robust-Q-FTRL, which follows a similar online learning procedure as Q-FTRL for standard MGs (Li et al., 2022). The complete procedure is summarized in Algorithm 2. We denote $Q_{i,h}^k \in \mathbb{R}^{SA_i}$ as the estimated robust Q-function of the equilibrium for the $i$-th agent at the $k$-th iteration of time step $h$. To begin with, Robust-Q-FTRL initialize the robust value function, robust Q-function $\widehat{V}_{i,H+1}(s) = Q_{i,h}^0(s, a_i) = 0$, and the policy $\pi_{i,h}^1(a_i \mid s) = 1/A_i$ for all $i \in [n]$. Then subsequently from the final time step $h = H$ to $h = 1$, for each step $h$, a $K$ iterations online learning process will be

executed. At each $k$-th iteration, given current policy $\pi_h^k$, as described above, an empirical model ($\{r_{i,h}^k\}_{i\in[n]}$ and $\{P_{i,h}^k\}_{i\in[n]}$) is constructed by $N$-sample estimation (cf. algorithm 1). Then the robust Q-function $\{q_{i,h}^k\}_{i\in[n]}$ of the current policy $\pi_h^k$ is estimated by (15).

Now we are ready to specify the loss objective and proceed the online learning procedure. With the current one-step update $\{q_{i,h}^k\}$, we update the Q-estimate as $Q_{i,h}^k = (1-\alpha_k)Q_{i,h}^{k-1} + \alpha_k q_{i,h}^k$. Here, $\{\alpha_k\}_{k\in[K]}$ is a series of rescaled linear learning rates with some $c_\alpha \geq 24$, where for all $k \in [K]$,

$$\alpha_k = \frac{c_\alpha \log K}{k - 1 + c_\alpha \log K} \quad \text{and} \quad \alpha_k^n = \begin{cases} \alpha_k \prod_{i=k+1}^n (1-\alpha_i), & \text{if } 0 < k < n \leq K \\ \alpha_n & \text{if } k = n \end{cases}. \quad (16)$$

Let the Q-estimate be the online learning loss objective at this moment, we apply the Follow-the-Regularized-Leader strategy (Shalev-Shwartz, 2012; Li et al., 2022) to update the corresponding policy as: $\pi_{i,h}^{k+1}(a_i \,|\, s) = \frac{\exp\left(\eta_{k+1} Q_{i,h}^k(s,a_i)\right)}{\sum_{a'} \exp\left(\eta_{k+1} Q_{i,h}^k(s,a')\right)}$ with $\eta_{k+1} = \sqrt{\frac{\log K}{\alpha_k H}}$ for $k = 1, 2, \dots$. This is a widely used adaptive sampling and learning procedure for MARL problems.

After completing $K$ iterations for time step $h$, we finalize the robust value function estimation by setting it to its confidence upper bound, incorporating carefully designed optimistic bonus terms $\{\beta_{i,h}\}$ as: for all $(i, h, s) \in [n] \times [H] \times \mathcal{S}$,

$$\beta_{i,h}(s) = c_{\mathsf{b}} \sqrt{\log^3(\frac{KS\sum_{i=1}^n A_i}{\delta})} \sqrt{\frac{1}{KH}\sum_{k=1}^K \alpha_k^K \left\{\mathsf{Var}_{\pi_{i,h}^k(\cdot|s)}\left(q_{i,h}^k(s,\cdot)\right) + H\right\}}, \quad (17)$$

where $c_{\mathsf{b}}$ denotes some absolute constant, $\delta \in (0,1)$ is the high probability threshold, Finally, after the recursive learning process ends for all time steps $h = H, H-1, \cdots, 1$, we output a distribution of product policy $\widehat{\xi} = \{\widehat{\xi}_h\}_{h\in[H]}$ over all the policies $\{\pi_h^k = (\pi_{1,h}^k \times \cdots \times \pi_{n,h}^k)\}_{h\in[H],k\in[K]}$ occurs during the process that defined as $\xi_h(\pi_h^k) := \alpha_k^K$ for all $(h,k) \in [H] \times [K]$.

### 4.3 THEORETICAL GUARANTEES

In this section, we provide the theoretical guarantees for the sample complexity of our proposed algorithm Robust-Q-FTRL (Algorithm 2), shown as below:

**Theorem 2** (Upper bound). *Using the TV uncertainty set defined in* (12). *Consider any $\delta \in (0,1)$ and any fictitious RMGs $\mathcal{RMG}_\pi = \left\{\mathcal{S}, \{\mathcal{A}_i\}_{1\leq i\leq n}, \{\mathcal{U}^{\sigma_i}(P^0,\cdot)\}_{1\leq i\leq n}, r, H\right\}$ with $\sigma_i \in (0,1]$ for all $i \in [n]$. For any $\varepsilon \leq \sqrt{\min\left\{H, \frac{1}{\min_{1\leq i\leq n}\sigma_i}\right\}}$, Algorithm 2 can output an $\varepsilon$-robust CCE $\widehat{\xi}$, i.e.,*

$$\mathsf{gap}_{\mathsf{CCE}}(\widehat{\xi}) := \max_{s\in\mathcal{S}, 1\leq i\leq n}\left\{\mathbb{E}_{\pi\sim\widehat{\xi}}\left[V_{i,1}^{\star,\pi_{-i},\sigma_i}(s)\right] - \mathbb{E}_{\pi\sim\widehat{\xi}}\left[V_{i,1}^{\pi,\sigma_i}(s)\right]\right\} \leq \varepsilon$$

*with probability at least $1-\delta$, as long as $N \geq \frac{C_1 H^2}{\epsilon^2}\min\left\{\frac{1}{\min_{1\leq i\leq n}\sigma_i}, H\right\}$ and $K \geq \frac{C_1 H^3}{\epsilon^2}$. Here $C_1$ is some universal large enough constant. Namely, it is sufficient if the total number of samples acquired in the learning process obeys*

$$N_{\mathsf{all}} := HKNS\sum_{1\leq i\leq n}A_i \geq \frac{(C_1)^2 H^6 S\sum_{1\leq i\leq n}A_i}{\varepsilon^4}\min\left\{H, \frac{1}{\min_{1\leq i\leq n}\sigma_i}\right\}.$$

Before we jump into more discussions of the above theorem, in addition, we introduce the information-theoretic minimax lower bound for this problem as well.

**Lower bound for learning in fictitious RMGs.** Considering the instances of fictitious RMGs that the action space for all the agents except the $i$-th agent contains only a single action, i.e., $A_j = 1$ for all $j \neq i$. As such, all the agents $j \neq i$ will take a fixed action and the game reduces to a single-agent robust MDP with $(s,a)$-*rectangularity condition* (Zhou et al., 2021). So the goal of finding the robust equilibrium — robust NE/CCE also degrades to finding the optimal policy of the $i$-th agent. Invoking the results from Shi et al. (2024, Theorem 2), the lower bound for the class of fictitious RMGs is achieved directly: consider any tuple $\left\{S, \{A_i\}_{1\leq i\leq n}, \{\sigma_i\}_{1\leq i\leq n}, H\right\}$ obeying $\sigma_i \in (0, 1-c_0]$ with $0 < c_0 \leq \frac{1}{4}$ being any small enough

positive constant, and $H > 16\log 2$. Let $\varepsilon \leq \begin{cases} \frac{c_1}{H}, & \text{if } \sigma_i \leq \frac{c_1}{2H}, \\ 1 & \text{otherwise} \end{cases}$ for any $c_1 \leq \frac{1}{4}$. We can construct a set of fictitious RMGs $\mathcal{M} = \{\mathcal{RMG}_i\}_{i \in [I]}$, such that for any dataset generated from the nominal environment with in total $N_{\mathsf{all}}$ independent samples over all state-action pairs, we have $\inf_{\widehat{\xi} \in [H] \mapsto \Delta(\mathcal{S} \mapsto \prod_{i=1}^n \mathcal{A}_i)} \max_{\mathcal{RMG}_i \in \mathcal{M}} \left\{ \mathbb{P}_{\mathcal{MG}_i}\left( \mathsf{gap}_{\mathsf{CCE}}(\widehat{\xi}) > \varepsilon \right) \right\} \geq \frac{1}{8}$, provided that

$$N_{\mathsf{all}} \leq \frac{C_2 SH^3 \max_{1 \leq i \leq n} A_i}{\varepsilon^2} \min\left\{ H, \frac{1}{\min_{1 \leq i \leq n} \sigma_i} \right\}. \tag{18}$$

Here, the infimum is taken over all estimators $\widehat{\xi}$, $\mathbb{P}_{\mathcal{RMG}_i}$ denotes the probability when the game is $\mathcal{MG}_i$ for all $\mathcal{MG}_i \in \mathcal{M}$, and $C_2$ is some small enough constant.

Armed with both the upper bound (Theorem 2) and lower bound in (18), we are now ready to discuss the implications of our sample complexity results.

**Breaking the curse of multiagency in the sample complexity for RMGs.** Theorem 2 demonstrates that for any fictitious RMGs, Robust-Q-FTRL algorithm finds an $\epsilon$-robust CCE when the total number of samples exceeds

$$\widetilde{O}\left( \frac{SH^6 \sum_{1 \leq i \leq n} A_i}{\epsilon^4} \min\left\{ H, \frac{1}{\min_{1 \leq i \leq n} \sigma_i} \right\} \right). \tag{19}$$

To the best of our knowledge, Robust-Q-FTRL with the above sample complexity in (19) is the first algorithm for RMGs breaking the curse of multiagency, regardless of the types of uncertainty sets. Our sample complexity depends linearly on the sum of each agent's actions $\sum_{i=1}^n A_i$ rather than their product $\prod_{i=1}^n A_i$—making the algorithm highly scalable as the number of agents increases. Nonetheless, there still exist gaps between our upper bound and the lower bound—especially in terms o the dependency on the horizon length $H$ and the accuracy level $\varepsilon$—an interesting direction to investigate in the future.

**Comparisons with prior works.** All prior works focus on learning equilibria for a different kind of robust MGs with $(s, \boldsymbol{a})$-rectangular uncertainty sets (Ma et al., 2023; Blanchet et al., 2023; Shi et al., 2024). However, the state-of-the-art sample complexity $\widetilde{O}\left( \frac{SH^3 \prod_{i=1}^n A_i}{\varepsilon^2} \min\left\{ H, \frac{1}{\min_{1 \leq i \leq n} \sigma_i} \right\} \right)$ (Shi et al., 2024) still suffers from the curse of multiagency with an exponential dependency on the number of agents when all agents have equal action spaces, which uses nonadaptive sampling. Our work circumvents the curse of multiagency by resorting to a tailored adaptive sampling and online learning procedure, together with the introduction of a new class of fictitious RMGs, providing a fresh perspective to learning RMGs.

**Technical insights.** For sample complexity analysis, while previous works have addressed the curse of multiagency in sequential games like standard Markov games (MGs) and Markov potential games, these methods are not directly applicable to RMGs. Prior approaches assume a linear relationship between the value function and the transition kernel, allowing statistical errors across $K$ iterations to cancel out. However, in RMGs, the robust value function, due to its distributionally robust requirement, is highly nonlinear and often lacks a closed form, making it impossible to linearly aggregate statistical errors. To tackle the nonlinear challenges in RMGs, we design a variance-style bonus term through non-trivial decomposition and control of auxiliary statistical errors caused by nonlinearity, resulting in a tight upper bound on regret during the online learning process.

## 5 CONCLUSION

Robustness in MARL presents greater challenges than in single-agent RL due to the strategic interactions between agents in a game-theoretic setting. This work proposes a new class of RMGs with fictitious uncertainty sets that naturally extends from robust single-agent RL and addresses more realistic scenarios where each agent's uncertainty is influenced by both the environment and the behavior of others. We then propose Robust-Q-FTRL, the first algorithm to break the curse of multiagency in robust Markov games regardless of the uncertainty set definitions, with sample complexity scaling polynomially with all key parameters. This opens up new research directions in MARL, such as uncertainty set selection, equilibrium refinement, and sample-efficient algorithm design.

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

---

**Algorithm 1:** $N$-sample estimation $\big(\pi_h = \{\pi_{j,h}\}_{j\in[n]}, i, h\big)$.

---

1 **Initialization:** the reward $\widehat{r} = 0 \in \mathbb{R}^{SA_i}$ and the transition model $\widehat{P} = 0 \in \mathbb{R}^{SA_i \times S}$.

2 **for** $(s, a_i) \in \mathcal{S} \times \mathcal{A}_i$ **do**

3      **for** $t = 1$ **to** $N$ **do**

4          Sample $\boldsymbol{a}^t(s, a_i) = [a_j(s, a_i)]_{1 \leq j \leq n}$ constructed by independent actions drawn from policy:

$$a_j(s, a_i) \overset{\text{ind.}}{\sim} \pi_{j,h}(\cdot \mid s) \quad (j \neq i) \qquad \text{and} \qquad a_i(s, a_i) = a_i. \tag{20}$$

5          Sample from the generative model:

$$r_{i,h}^t(s, a_i) = r_{i,h}(s, \boldsymbol{a}^t(s, a_i)), \qquad s_{s,a_i}^t \sim P_h\big(\cdot \mid s, \boldsymbol{a}^t(s, a_i)\big). \tag{21}$$

6      Set $\widehat{r}(s, a_i) = \frac{1}{N} \sum_{t\in[N]} r_{i,h}^t(s, a_i)$ and $\widehat{P}\big(s' \mid s, a_i\big) = \frac{1}{N} \sum_{t\in[N]} \mathbb{1}\big\{s_{s,a_i}^t = s'\big\}$.

7 **Return:** empirical model $\big(\widehat{r}, \widehat{P}\big)$.

---

**Algorithm 2:** Robust-Q-FTRL

---

1 *Input:* learning rates $\{\alpha_k\}$ and $\{\eta_{k+1}\}$, number of iterations $K$ per time step, and number of samples $N$ per iteration.

2 *Initialization:* $\widehat{V}_{i,H+1}(s) = Q_{i,h}^0(s, a_i) = 0$ and $\pi_{i,h}^1(a_i \mid s) = 1/A_i$ for all $i \in [n]$ and then all $(h, s, a_i) \in [H] \times \mathcal{S} \times \mathcal{A}_i$.

// start recursive learning process.

3 **for** $h = H, H-1, \cdots, 1$ **do**

4      **for** $k = 1, 2, \cdots, K$ **do**

5          **for** $i = 1, 2, \cdots, n$ **do**

             // construct empirical models and estimate current robust Q-function

6              $\big(r_{i,h}^k, P_{i,h}^k\big) \leftarrow N$-sample estimation $\big(\pi_h^k = \{\pi_{j,h}^k\}_{j\in[n]}, i, h\big)$. (Algorithm 1)

7              Estimate the robust Q-function $q_{i,h}^k$ of current $\pi_h^k$ according to (15).

             // Online learning procedure

8              Update the Q-estimate $Q_{i,h}^k = (1 - \alpha_k)Q_{i,h}^{k-1} + \alpha_k q_{i,h}^k$ and apply FTRL:

$$\forall(s, a_i) \in \mathcal{S} \times \mathcal{A}_i: \quad \pi_{i,h}^{k+1}(a_i \mid s) = \frac{\exp\big(\eta_{k+1} Q_{i,h}^k(s, a_i)\big)}{\sum_{a'} \exp\big(\eta_{k+1} Q_{i,h}^k(s, a')\big)}.$$

     // set the final robust value estimate at time step $h$.

9      **for** $i = 1, 2, \cdots, n$ **do**

10          For all $s \in \mathcal{S}$: set $\beta_{i,h}(s)$ to be the optimistic bonus term in (17) and

$$\widehat{V}_{i,h}(s) = \min\left\{ \sum_{k=1}^{K} \alpha_k^K \big\langle \pi_{i,h}^k(\cdot \mid s), q_{i,h}^k(s, \cdot)\big\rangle + \beta_{i,h}(s), \ H - h + 1\right\}, \tag{22}$$

11 *Output:* a set of policies $\{\pi_h^k = (\pi_{1,h}^k \times \cdots \times \pi_{n,h}^k)\}_{k\in[K], h\in[H]}$ and a distribution $\widehat{\xi} = \{\widehat{\xi}_h\}_{h\in[H]}$ over them. For any time step $h$, $\widehat{\xi}_h$ is the distribution over $\{\pi_h^k\}_{k\in[K]}$ so that $\widehat{\xi}_h(\pi_h^k) = \alpha_k^K$.

---

Ziyuan Zhou and Guanjun Liu. Robustness testing for multi-agent reinforcement learning: State perturbations on critical agents. *arXiv preprint arXiv:2306.06136*, 2023.

## A    RELATED WORK

**Breaking curse of multiagency for standard Markov games.** Breaking the curse of multiagency is a major and prevalent challenge in sequential games. In standard multi-agent general-sum MGs,

it has been shown that learning a Nash equilibrium requires an exponential sample complexity (Song et al., 2021; Rubinstein, 2017; Bai & Jin, 2020). However, for other types of equilibria, such as CE and CCE, many works have successfully broken the curse of multiagency. Specifically, for finite-horizon general-sum MGs in the tabular setting with finite state and action spaces, Jin et al. (2021) developed the V-learning algorithm for learning CE and CCE with the sample complexity of $\widetilde{O}(H^6 S (\max_{i \in [n]} A_i)^2 / \epsilon^2)$ and $\widetilde{O}(H^6 S \max_{i \in [n]} A_i / \epsilon^2)$, respectively; Daskalakis et al. (2023) achieved a sample complexity of $\widetilde{O}(H^{11} S^3 \max_{i \in [n]} A_i / \epsilon^3)$ for learning a CCE. Beyond tabular settings, Wang et al. (2023) and Cui et al. (2023) extended these results to linear function approximation, achieving sample complexities of $\widetilde{O}(d^4 H^6 (\max_{i \in [n]} A_i^5) / \epsilon^2)$ and $\widetilde{O}(H^{10} d^4 \log (\max_{i \in [n]} A_i) / \epsilon^4)$, respectively, where $d$ is the dimension of the linear features. For Markov potential games, a subclass of MGs, Song et al. (2021) provided a centralized algorithm that learns a NE with a sample complexity of $\widetilde{O}(H^4 S^2 \max_{i \in [n]} A_i / \epsilon^3)$.

**Finite-sample analysis for distributionally robust Markov games.** Robust Markov games under environmental uncertainty are largely underexplored, with only a few provable algorithms (Zhang et al., 2020a; Kardeş et al., 2011; Ma et al., 2023; Blanchet et al., 2023; Shi et al., 2024). Existing sample complexity analyses all suffer from the daunting curse of multiagency issues, or impose an extremely restricted uncertainty level that can fail to deliver the desired robustness (Ma et al., 2023; Blanchet et al., 2024; Shi et al., 2024). Specifically, they all consider a class of RMGs with the $(s, \boldsymbol{a})$-*rectangularity condition*, where the uncertainty sets for each agent can be decomposed into independent sets over each $(s, \boldsymbol{a})$ pair. Shi et al. (2024) considered the generative model with an uncertainty set measured by the TV distance, Blanchet et al. (2023) treated a different sampling mechanism with offline data for both the TV distance and KL divergence. In addition, Ma et al. (2023) required the uncertainty level be much smaller than the accuracy-level and an instance-dependent parameter (i.e., $\sigma_i \leq \max\{\frac{\varepsilon}{SH^2}, \frac{p_{\min}}{H}\}$ for all $i \in [n]$). This can thus fail to maintain the desired robustness, especially when the accuracy requirement is high (i.e., $\varepsilon \to 0$) or the RMG has small minimal positive transition probabilities (i.e., $p_{\min} \to 0$).

**Robust MARL.** Standard MARL algorithms may overfit the training environment and could fail dramatically due to the perturbations and variability of both agents' behaviors and the shared environment, leading to performance drop and large deviation from the equilibrium. To address this, this work considers a robust variant of MARL adopting the distributionally robust optimization (DRO) framework that has primarily been investigated in supervised learning (Rahimian & Mehrotra, 2019; Gao, 2020; Bertsimas et al., 2018; Duchi & Namkoong, 2018; Blanchet & Murthy, 2019) and has attracted a lot of attention in promoting robustness in single-agent RL (Nilim & El Ghaoui, 2005; Iyengar, 2005; Badrinath & Kalathil, 2021; Zhou et al., 2021; Shi & Chi, 2024; Wang et al., 2024; Shi et al., 2023). Beyond the RMG framework considered in this work, recent research has advanced the robustness of MARL algorithms from various perspectives, including resilience to uncertainties or attacks on states (Han et al., 2022; Zhou & Liu, 2023), the type of agents (Zhang et al., 2021), other agents' policies (Li et al., 2019; Kannan et al., 2023), offline data poisoning (Wu et al., 2024; McMahan et al., 2024), and nonstationary environment (Szita et al., 2003). A recent review can be found in Vial et al. (2022).

# B PRELIMINARIES

Denoting the vectors $x = [x_i]_{1 \leq i \leq n}$ and $y = [y_i]_{1 \leq i \leq n}$, we use the notation $x \leq y$ (or $x \geq y$) to signify that $x_i \leq y_i$ (or $x_i \geq y_i$) for every $1 \leq i \leq n$. The Hadamard product of two vectors $x$ and $y$ in $\mathbb{R}^S$ is denoted as $x \circ y = [x(s) \cdot y(s)]_{s \in \mathcal{S}}$. In addition, for any series of vectors $\{x_i\}_{i \in [S}$, $\mathrm{diag}(x_1, x_2, \cdots, x_S)$ denote a block diagonal matrix by placing each given vector $x_i$ along the diagonal, with zeros filling the off-diagonal blocks. 0 (or 1) represents the all-zero (or all-one) vector, while $e_i \in \mathbb{R}^S$ denotes a basis vector of dimension $S$ with 1 in the $i$-th position and 0 elsewhere.

## B.1 ADDITIONAL MATRIX AND VECTOR NOTATION

Before continuing, we introduce or recall some matrix and vector notation that will be used throughout the paper. In particular, for any joint policy $\pi : \mathcal{S} \times [H] \mapsto \Delta(\mathcal{A})$ and any $(i, h) \in [n] \times [H]$:

**Matrices for policy.** We introduce three matrices associated with $\pi$, i.e., $\Pi_h^\pi \in \mathbb{R}^{S \times S \prod_{i=1}^n A_i}$, $\Pi_h^{\pi_{-i}} \in \mathbb{R}^{S \times S \prod_{j \neq i} A_j}$, and $\Pi_h^{\pi_i} \in \mathbb{R}^{S \times SA_i}$, which are defined as block diagonal matrices that adhere to the following properties:

- The matrix $\Pi_h^\pi$ is given by $\mathrm{diag}\left(\pi_h(1)^\top, \pi_h(2)^\top, \ldots, \pi_h^\top(S)\right)$, where $\pi_h(s) = [\pi_h(\mathbf{a} \,|\, s)]_{\mathbf{a} \in \mathcal{A}} \in \Delta(\mathcal{A})$ for each $s \in \mathcal{S}$ represents the joint policy vectors across all agents.

- The matrix $\Pi_h^{\pi_{-i}}$ can be expressed as $\mathrm{diag}\left(\pi_{-i,h}(1)^\top, \pi_{-i,h}(2)^\top, \ldots, \pi_{-i,h}^\top(S)\right)$, where $\pi_{-i,h}(s) = [\pi_h(\mathbf{a}_{-i} \,|\, s)]_{\mathbf{a}_{-i} \in \mathcal{A}_{-i}} \in \Delta(\mathcal{A}_{-i})$ for all $s \in \mathcal{S}$ denotes the joint policy vectors from all agents except agent $i$.

- The matrix $\Pi_h^{\pi_i}$ is defined as $\mathrm{diag}\left(\pi_{i,h}(1)^\top, \pi_{i,h}(2)^\top, \ldots, \pi_{i,h}^\top(S)\right)$, where $\pi_{i,h}(s) = [\pi_{i,h}(a_i \,|\, s)]_{a_i \in \mathcal{A}_i} \in \Delta(\mathcal{A}_i)$ for each $s \in \mathcal{S}$ represents the policy of the $i$-th agent.

**Reward vectors.** We recall the definition of $r_{i,h}$ and introduce the reward vectors $r_{i,h}^\pi$ and $r_{i,h}^{\pi_{-i}}$ as follows:

- Let $r_{i,h} = [r_{i,h}(s, \mathbf{a})]_{(s,\mathbf{a}) \in \mathcal{S} \times \mathcal{A}} \in \mathbb{R}^{S \prod_{i=1}^n A_i}$ represent the reward function for the $i$-th player at time step $h$, where $\mathcal{S}$ is the state space and $\mathcal{A}$ is the action space.
- The reward vector $r_{i,h}^\pi \in \mathbb{R}^S$ corresponds to the joint policy $\pi = \{\pi_h\}_{h \in [H]}$ at time step $h$. Specifically, for each $s \in \mathcal{S}$, $r_{i,h}^\pi(s) = \mathbb{E}_{\mathbf{a} \sim \pi_h(s)}[r_{i,h}(s, \mathbf{a})]$, where the expectation is taken over the actions $\mathbf{a}$ drawn from policy $\pi_h$ in state $s$.
- The reward vector $r_{i,h}^{\pi_{-i}} \in \mathbb{R}^{SA_i}$ corresponds to the joint policy $\pi_{-i} = \{\pi_{-i,h}\}_{h \in [H]}$ at time step $h$, excluding agent $i$. Specifically, for all $s \in \mathcal{S}$ and $a_i \in A_i$, $r_{i,h}^{\pi_{-i}}(s, a_i) = \mathbb{E}_{\mathbf{a}_{-i} \sim \pi_{-i,h}(s)}[r_{i,h}(s, \mathbf{a})]$, where the expectation is over the actions $\mathbf{a}_{-i}$ drawn from the joint policy $\pi_{-i,h}$ for all agents except agent $i$.

**Matrices for transition variants.** We first introduce the following notations related to transitions associated with the nominal transition kernel and the policy $\pi$:

- Define $P_h^0 \in \mathbb{R}^{S \prod_{i=1}^n A_i \times S}$, the matrix representing the nominal transition kernel at time step $h$. Specifically, for any $(s, \mathbf{a}) \in \mathcal{S} \times \mathcal{A}$, $P_{h,s,\mathbf{a}}^0 \in \mathbb{R}^{1 \times S}$ represents the row corresponding to the state-action pair $(s, \mathbf{a})$.
- Define $P_h^{\pi_{-i}} \in \mathbb{R}^{SA_i \times S}$, the matrix representing the nominal transition kernel at time step $h$, associated with the joint policy $\pi_{-i}$. Specifically, for all $s, s' \in \mathcal{S}$ and $a_i \in \mathcal{A}_i$, $P_{h,s,a_i}^{\pi_{-i}}(s') = \mathbb{E}_{\mathbf{a}_{-i} \sim \pi_{-i,h}(s)}[P_{h,s,\mathbf{a}}^0(s')]$. Here, $P_{h,s,a_i}^{\pi_{-i}} \in \mathbb{R}^{1 \times S}$ represents the row corresponding to the state-action pair $(s, a_i)$.
- Let $\widehat{P}_{i,h}^{\pi_{-i}} \in \mathbb{R}^{SA_i \times S}$ denote the empirical transition kernel matrix at time step $h$, associated with the joint policy $\pi_{-i}$ and agent $i$. Similarly, $\widehat{P}_{h,s,a_i}^{\pi_{-i}} \in \mathbb{R}^{1 \times S}$ represents the row corresponding to the state-action pair $(s, a_i)$.
- Define $\underline{P}_h^\pi \in \mathbb{R}^{S \times S}$ as $\underline{P}_h^\pi := \Pi_h^\pi P_h^0$, where $\Pi_h^\pi$ is the policy matrix at time step $h$ under joint policy $\pi$.
- Define $\widehat{\underline{P}}_{i,h}^\pi \in \mathbb{R}^{S \times S}$ as $\widehat{\underline{P}}_{i,h}^\pi := \Pi_h^{\pi_i} \widehat{P}_{i,h}^{\pi_{-i}}$, where $\Pi_h^{\pi_i}$ denotes the policy matrix at time step $h$ under policy $\pi_i$.

We introduce matrix notations for transitions that are associated not only with the nominal transition and policy $\pi$, but also with value functions:

- For time step $h \in [H]$, joint policy $\pi$, and a value vector $V \in \mathbb{R}^S$, we define $P_{i,h}^{\pi_{-i},V} \in \mathbb{R}^{SA_i \times S}$ as the matrix representing the worst-case transition probability kernel within the uncertainty set for agent $i$, centered around the nominal kernel. The row corresponding to the state-action pair $(s, a_i)$ in $P_{i,h}^{\pi_{-i},V}$, denoted as $P_{i,h,s,a_i}^{\pi_{-i},V} \in \mathbb{R}^S$, is given by:

$$P_{i,h,s,a_{-i}}^{\pi_{-i},V} = \mathrm{argmin}_{\mathcal{P} \in \mathcal{U}_\rho^{\sigma_i}(P_{h,s,a_i}^{\pi_{-i}})} \mathcal{P}V. \tag{23a}$$

We also define the transition matrices for specific value vectors as:

$$P_{i,h}^{\pi,V} := P_{i,h}^{\pi_{-i}, V_{i,h+1}^{\pi,\sigma_i}} \quad \text{and} \quad P_{i,h,s,a_i}^{\pi,V} := P_{i,h,s,a_i}^{\pi_{-i}, V_{i,h+1}^{\pi,\sigma_i}} = \mathrm{argmin}_{\mathcal{P} \in \mathcal{U}_\rho^{\sigma_i}(P_{h,s,a_i}^{\pi_{-i}})} \mathcal{P}V_{i,h+1}^{\pi,\sigma_i}. \tag{23b}$$

Finally, we define square matrices $\underline{P}_{i,h}^{\pi,V} \in \mathbb{R}^{S \times S}$ as: $\underline{P}_{i,h}^{\pi,V} := \Pi_h^{\pi_i} P_{i,h}^{\pi_{-i},V}$.

- By replacing the nominal transition kernel with the empirical transition kernel, we similarly define $\widehat{P}_{i,h}^{\pi_{-i},V}$ as the worst-case probability transition kernel within the uncertainty set for agent $i$, centered around the empirical kernel $\widehat{P}_{i,h}^{\pi_{-i}}$. The row corresponding to the state-action pair $(s, a_i)$ in $\widehat{P}_{i,h}^{\pi_{-i},V}$ is denoted as $\widehat{P}_{i,h,s,a_i}^{\pi_{-i},V} \in \mathbb{R}^S$ and is defined as:

$$\widehat{P}_{i,h,s,a_{-i}}^{\pi_{-i},V} = \mathrm{argmin}_{\mathcal{P} \in \mathcal{U}_\rho^{\sigma_i}(\widehat{P}_{i,h,s,a_i}^{\pi_{-i}})} \mathcal{P}V. \tag{23c}$$

The transition matrices $\widehat{P}_{i,h}^{\pi,V}$ for specific value vectors are defined as:

$$\widehat{P}_{i,h}^{\pi,V} := \widehat{P}_{i,h}^{\pi_{-i},V_{i,h+1}^{\pi,\sigma_i}} \quad \text{and} \quad \widehat{P}_{i,h,s,a_i}^{\pi,V} := \widehat{P}_{i,h,s,a_i}^{\pi_{-i},V_{i,h+1}^{\pi,\sigma_i}} = \mathrm{argmin}_{\mathcal{P} \in \mathcal{U}_\rho^{\sigma_i}(\widehat{P}_{i,h,s,a_i}^{\pi_{-i}})} \mathcal{P}V_{i,h+1}^{\pi,\sigma_i}, \tag{23d}$$

Additionally, we define square matrices $\underline{\widehat{P}}_{i,h}^{\pi,V} \in \mathbb{R}^{S \times S}$ as: $\underline{\widehat{P}}_{i,h}^{\pi,V} := \Pi_h^{\pi_i} \widehat{P}_{i,h}^{\pi_{-i},V}$.

**Variance.** We now introduce notations for variance corresponding to a specific probability distribution. For a probability vector $P \in \mathbb{R}^{1 \times S}$ and a vector $V \in \mathbb{R}^S$, we denote the variance of $V$ with respect to $P$ as $\mathsf{Var}_P(V)$, defined as:

$$\mathsf{Var}_P(V) := P(V \circ V) - (PV) \circ (PV), \tag{24}$$

Additionally, for a transition kernel $P^{\pi_{-i}} \in \mathbb{R}^{SA_i \times S}$ and a vector $V \in \mathbb{R}^S$, we define $\mathsf{Var}_{P^{\pi_{-i}}}(V) \in \mathbb{R}^{SA_i}$ as a vector of variances. The entry corresponding to $(s, a_i)$ in $\mathsf{Var}_{P^{\pi_{-i}}}(V)$ is given by:

$$\mathsf{Var}_{P^{\pi_{-i}}}(s, a_i) := \mathsf{Var}_{P_{s,a_i}^{\pi_{-i}}}(V), \tag{25}$$

where $P_{s,a_i}^{\pi_{-i}}$ denotes the row of the transition matrix corresponding to state $s$ and action $a_i$.

### B.2 ROBUST BELLMAN EQUATIONS FOR RMGS WITH FICTITIOUS UNCERTAINTY SETS

For any joint policy $\pi : \mathcal{S} \times [H] \mapsto \Delta(\mathcal{A})$, the robust value function can be expressed as

$$V_{i,h}^{\pi,\sigma_i}(s) = \inf_{\mathcal{U}_\rho^{\sigma_i}(P^0,\pi)} \mathbb{E}\Big[ \sum_{t=h}^H r_i(s_t, a_t) \,|\, s_h = s \Big] = \mathbb{E}_{\boldsymbol{a} \sim \pi_h(s)}\Big[ r_{i,h}(s, \boldsymbol{a}) + \inf_{\mathcal{U}_\rho^{\sigma_i}\left(P_{h,s,a_i}^{\pi_{-i}}\right)} PV_{i,h+1}^{\pi,\sigma_i} \Big]. \tag{26}$$

It can be verified directly by definition. The robust Bellman equation described above is intrinsically linked to the *policy-induced $(s, a_i)$-rectangularity* condition (cf. (11)) of the uncertainty set. This condition leads to a well-posed and computationally-tractable class of RMGs by allowing the decomposition from an overall uncertainty set to independent subsets across different agents, time steps, and each state-action pair $(s, a_i)$.

Note that the specified robust Bellman equation is different for a joint correlated policy and a joint product policy, induced by different expected nominal transition kernels. In particular, for any joint product policy $\pi : \mathcal{S} \times [H] \mapsto \prod_{i \in [n]} \Delta(\mathcal{A}_i)$, the expected nominal transition kernel conditioned on the $i$-th agent's action $a_i \in \mathcal{A}_i$, current state $s \in \mathcal{S}$, and the policy $\pi$ can be expressed by

$$P_{h,s,a_i}^{\pi_{-i}} = \mathbb{E}_{\boldsymbol{a} \sim \pi_h(\cdot \,|\, s, a_i)}\big[ P_{h,s,\boldsymbol{a}}^0 \big] = \mathbb{E}_{\boldsymbol{a}_{-i} \sim \pi_{-i,h}(\cdot \,|\, s)}\big[ P_{h,s,(a_i,\boldsymbol{a}_{-i})}^0 \big] \tag{27}$$

for any $(i, h, s, a_i) \in [n] \times [H] \times \mathcal{S} \times \mathcal{A}_i$, where the last equality holds since the policy $\pi$ is a product policy, and the distribution of $\boldsymbol{a}_{-i}$ is independent of $a_i$. It is observed that the expected nominal transition kernel $P_{h,s,a_i}^{\pi_{-i}}$ for a product policy $\pi$ is independent of the $i$-th agent's policy given $(s, a_i)$. This differs from (10) for a possibly correlated policy, where (10) can generally depend on the $i$-th agent's policy.

### B.3 PRELIMINARY FACTS ABOUT FTRL

Our proposed algorithm (see Algorithm 2) is inspired by online adversarial learning. In this section, we introduce the formulation of online learning and review key aspects of a widely-used algorithm, the Follow-the-Regularized-Leader (FTRL) algorithm.

**Problem setting: online learning for weighted average loss.** We consider an online learning problem over $K$ steps, commonly found in adversarial learning settings (Lattimore & Szepesvári, 2020). The learner is presented with an action set $\mathcal{A}$, and loss functions $f_1, \ldots, f_K : \mathcal{A} \to \mathbb{R}_{\geq 0}$ are provided for each step. At each time step $k$, the learner selects a distribution over the action set, $\pi_k \in \Delta(\mathcal{A})$, and observes the loss function $f_k(\pi_k)$. The goal of the learner is to minimize the weighted average loss over the $K$ steps, which is defined as: $L^K = \sum_{k=1}^{K} \alpha_k^K f_k(\pi_k)$. To evaluate the learner's performance, the regret for the online learning process is defined as:

$$R^K = \sum_{k=1}^{K} \alpha_k^K f_k(\pi_k) - \left[ \min_{\pi \in \Delta(\mathcal{A})} \sum_{k=1}^{K} \alpha_k^K f_k(\pi) \right]. \tag{28}$$

**FTRL and its regret bound.** A widely-used method for solving the online learning problem described above is the Follow-the-Regularized-Leader (FTRL) algorithm, introduced by Shalev-Shwartz & Singer (2007); Shalev-Shwartz (2007). At each step $k + 1$, the learner selects a soft-greedy action by solving:

$$\pi_{k+1} = \arg \min_{\pi \in \Delta(\mathcal{A})} \left[ \sum_{i=1}^{k} \alpha_i^k f_i(\pi) + F_k(\pi) \right], \quad k = 1, 2, \ldots, \tag{29}$$

where $F_k(\pi)$ represents a convex regularization function. The following theorem provides a refined regret bound for the FTRL algorithm when the loss function is linear with respect to the policy.

**Theorem 3** (Theorem 3 in Li et al. (2022)). *For all $k \in [K]$ and policy $\pi$, the loss function is defined as $f_k(\pi) = \langle \pi_k, l_k \rangle$, where $l_k \in \mathbb{R}^{|\mathcal{A}|}$ represents a loss vector. The learner's choice $\pi_{k+1}$ in episode $k + 1$ is updated according to the FTRL algorithm:*

$$\pi_{k+1}(a) = \arg \min_{\pi \in \Delta(\mathcal{A})} \left\{ \langle \pi, L_k \rangle + F_k(\pi) \right\} = \frac{\exp\left( -\eta_{k+1} L_k(a) \right)}{\sum_{a' \in \mathcal{A}} \exp\left( -\eta_{k+1} L_k(a') \right)}, \quad \text{for all } a \in \mathcal{A}, \tag{30}$$

*where the regularization function is given by $F_k(\pi) = \sum_{a \in \mathcal{A}} \frac{1}{\eta_{k+1}} \pi(a) \log(\pi(a))$. Suppose $0 < \alpha_1 \leq 1$ and $\eta_1 = \eta_2(1 - \alpha_1)$, and for all $k \geq 2$, assume $0 < \alpha_k < 1$ and $0 < \eta_{k+1}(1 - \alpha_k) \leq \eta_k$. Define:*

$$\widehat{\eta}_k := \begin{cases} \eta_2, & \text{if } k = 1, \\ \frac{\eta_k}{1 - \alpha_k}, & \text{if } k > 1. \end{cases} \tag{31}$$

*Then, the regret of the FTRL algorithm is bounded by:*

$$R_n \leq \max_{a \in \mathcal{A}} \left[ \sum_{k=1}^{K} \alpha_k^K \langle \pi_k, l_k \rangle - \sum_{k=1}^{K} \alpha_k^K l_k(a) \right]$$

$$\leq \frac{5}{3} \sum_{k=1}^{n} \alpha_k^n \widehat{\eta}_k \alpha_k \mathsf{Var}_{\pi_k}(l_k) + \frac{\log A}{\eta_{n+1}} + 3 \sum_{k=1}^{n} \alpha_k^n \widehat{\eta}_k^2 \alpha_k^2 \|l_k\|_\infty^3 \mathbb{I}\left( \widehat{\eta}_k \alpha_k \|l_k\|_\infty > \frac{1}{3} \right). \tag{32}$$

## C   PROOF FOR SECTION 3

### C.1   PROOF OF THEOREM 1

**Step 1: preliminaries.** First, we introduce some useful definition and existing facts that are standard in real analysis and game theory literature.

**Definition 2** (Upper semi-continuous). *A point-to-set mapping $x \in \mathcal{X} \mapsto \phi(x) \in \mathcal{Y}$ is upper semi-continuous if $\lim_{n \to \infty} x^n = x_0, y^n \in \phi(x^n), \lim_{n \to \infty} y^n = y_0$ imply that $y^0 \in \phi(x_0)$.*

**Theorem 4** (Kakutani's fixed point Theorem (Kakutani, 1941)). *If $X$ is a closed, bounded, and convex set in a Euclidean space, and $\phi$ is a upper semi-continuous correspondence mapping $X$ into the family of all closed convex subsets of $X$, then there exists $x \in X$ so that $x \in \phi(x)$.*

**Step 2: constructing an auxiliary single-step game.** Focusing on finite-horizon RMG $\mathcal{MG}_{\text{rob}} = \{\mathcal{S}, \{\mathcal{A}_i\}_{1 \leq i \leq n}, \{\mathcal{U}_\rho^{\sigma_i}(P^0)\}_{1 \leq i \leq n}, r, H\}$, we shall verify the theorem by firstly consider a one-step game and then apply the results recursively to the sequential Markov games.

Without loss of generality, we focus on any of the steps $h \in [H]$ and construct an auxiliary one-step game. Towards this, we first introduce a fixed value function $V_{i,h+1} \in \mathbb{R}^S$ with $0 \leq V_{i,h+1} \leq H$ for the $i$-th agent, representing the possible value function obtained at the next time step $h+1$. Focusing on time step $h$, for any joint product policy $\pi : \mathcal{S} \mapsto \prod_{i \in [n]} \Delta(\mathcal{A}_i)$, we abuse the notation defined in (27) to denote the expected nominal transition kernel over each $(s, a_i)$ as:

$$P_{h,s,a_i}^{\pi_{-i}} = \mathbb{E}_{\pi(\boldsymbol{a}_{-i} \,|\, s, a_i)} \left[ P_{h,s,(a_i, \boldsymbol{a}_{-i})}^0 \right] = \mathbb{E}_{\pi_{-i}(\boldsymbol{a}_{-i} \,|\, s)} \left[ P_{h,s,(a_i, \boldsymbol{a}_{-i})}^0 \right]. \tag{33}$$

Armed with this, for any joint product policy $\pi : \mathcal{S} \mapsto \prod_{i \in [n]} \Delta(\mathcal{A}_i)$, we can define the payoffs to maximize for the players as below:

$$\forall s \in \mathcal{S}: \quad f_{i,s}(\pi_i(s), \pi_{-i}(s); V_{i,h+1}) = \mathbb{E}_{\boldsymbol{a} \sim \pi(s)}[r_{i,h}(s, \boldsymbol{a})] + \mathbb{E}_{a_i \sim \pi_i(s)} \left[ \inf_{\mathcal{U}^{\sigma_i}\left(P_{h,s,a_i}^{\pi_{-i}}\right)} PV_{i,h+1} \right], \tag{34}$$

which is defined analogous to the robust Bellman equation (cf. (26)) by replacing a real robust value function vector (associated with some policy) to some fixed vector $V_{i,h+1}$.

Now we are ready to introduce the following useful mapping: for any $\pi : \mathcal{S} \mapsto \prod_{i \in [n]} \Delta(\mathcal{A}_i)$,

$$\phi(\pi) \coloneqq \left\{ u \,|\, u_i(s) \in \text{argmax}_{\pi_i'(s) \in \Delta(\mathcal{A}_i)} \, f_{i,s}(\pi_i'(s), \pi_{-i}(s); V_{i,h+1}), \forall (i, s) \in [n] \times \mathcal{S} \right\}. \tag{35}$$

**Step 3: the existence of NE of the auxiliary game.** To apply Theorem 4, there are three required conditions. First, we know that the space of product policy is $X = \{\pi : \mathcal{S} \mapsto \prod_{i \in [n]} \Delta(A_i)\}$ is a closed, bounded and convex set in Euclidean space.

- **Verifying that $\phi(\pi)$ is an upper semi-continuous correspondence.** Before starting, we introduce the following two useful lemmas with the proof postponed to Appendix C.2.2 and C.2.3.
  **Lemma 2.** *The set of function $\{f_{i,s}(\pi_i'(s), \pi_{-i}(s); V_{i,h+1}), 0 \leq V_{i,h+1}) \leq H\}$ is equicontinuous with respect to $\pi_i'(s), \pi_{-i}(s)$ for all $(i, s) \in [n] \times \mathcal{S}$.*
  **Lemma 3.** *For any $i \in [n]$ and then $x_{-i} : \mathcal{S} \mapsto \prod_{j \neq i, j \in [n]} \Delta(\mathcal{A}_j)$, the functions*

  $$\forall s \in \mathcal{S}: \quad g_{i,s}(x_{-i}(s), V_{i,h+1}) \coloneqq \max_{\pi_i' \in \Delta(\mathcal{S})} f_{i,s}(\pi_i'(s), x_{-i}(s); V_{i,h+1}) \tag{36}$$

  *are continuous with respect to $x_{-i}(s)$ and the set $\{g_{i,s}(\cdot, V) | V \in \mathbb{R}^S, 0 \leq V \leq H\}$ is equicontinuous.*
  Armed with above lemmas, we are in the position to prove this condition. We suppose there are two sequence $\lim_{n \to \infty} x^n = x^0, y^n \in \phi(x^n), \lim_{n \to \infty} y^n = y^0$. Recall the definition of a upper semi-continuous correspondence (cf. Definition 2), we are supposed to show that $y^0 \in \phi(x^0)$, i.e.,

  $$\forall (i, s) \in [n] \times \mathcal{S}: \quad f_{i,s}(y_i^0(s), x_{-i}^0(s); V_{i,h+1}) = \max_{\pi_i' \in \Delta(\mathcal{S})} f_{i,s}(\pi_i'(s), x_{-i}^0(s); V_{i,h+1}). \tag{37}$$

  Towards this, we have

  $$|f_{i,s}(y_i^0(s), x_{-i}^0(s); V_{i,h+1}) - g_{i,s}(x_{-i}^0(s), V_{i,h+1})|$$
  $$\leq |f_{i,s}(y_i^0(s), x_{-i}^0(s); V_{i,h+1}) - f_{i,s}(y_i^n(s), x_{-i}^n(s); V_{i,h+1})|$$
  $$\quad + |f_{i,s}(y_i^n(s), x_{-i}^n(s); V_{i,h+1}) - g_{i,s}(x_{-i}^0(s), V_{i,h+1})|$$
  $$\overset{(i)}{=} |f_{i,s}(y_i^0(s), x_{-i}^0(s); V_{i,h+1}) - f_{i,s}(y_i^n(s), x_{-i}^n(s); V_{i,h+1})|$$
  $$\quad + |g_{i,s}(x_{-i}^n(s), V_{i,h+1}) - g_{i,s}(x_{-i}^0(s), V_{i,h+1})| \to 0 \quad \text{as} \quad n \to \infty, \tag{38}$$

  where the first inequality follows from the triangle inequality, (i) holds by the assumption $y^n \in \phi(x^n)$ so that $f_{i,s}(y_i^n(s), x_{-i}^n(s); V_{i,h+1}) = \max_{\pi_i' \in \Delta(\mathcal{S})} f_{i,s}(\pi_i'(s), x_{-i}^n(s); V_{i,h+1})$, and the last line can be verified by the continuity implied by Lemma 2 and Lemma 3.

- **Verifying $\phi(\pi)$ is convex for any $\pi \in X$.** Finally, we gonna work on the convexity of $\phi(\pi)$ for any $\pi \in X$. To begin with, by the definition of $\phi(\pi)$ in (35), we know that $\phi(\pi) \subseteq X$ and the maximum of the continuous function $f_{i,s}(\pi_i(s), \pi_{-i}(s); V_{i,h+1})$ (cf. Lemma 2) on a compact set exists, i.e., $\phi(x) \neq \emptyset$.
  Suppose there exists two Nash equilibrium $z : \mathcal{S} \mapsto \prod_{i \in [n]} \Delta(\mathcal{A}_i), v : \mathcal{S} \mapsto \prod_{i \in [n]} \Delta(\mathcal{A}_i)$ and $z, v \in \phi(\pi)$. Then we have that for any $(i, s) \in [n] \times \mathcal{S}$,

$$
f_{i,s}(z_i(s), \pi_{-i}(s); V_{i,h+1}) = f_{i,s}(v_i(s), \pi_{-i}(s); V_{i,h+1})
$$
$$
= \max_{u_i(s) \in \Delta(\mathcal{A}_i)} f_{i,s}(u_i(s), \pi_{-i}(s); V_{i,h+1}). \tag{39}
$$

To continue, for any $0 \leq \lambda \leq 1$, one has

$$
\max_{u_i(s) \in \Delta(\mathcal{A}_i)} f_{i,s}(u_i(s), \pi_{-i}(s); V_{i,h+1})
$$
$$
= \lambda f_{i,s}(z_i(s), \pi_{-i}(s); V_{i,h+1}) + (1 - \lambda) f_{i,s}(v_i(s), \pi_{-i}(s); V_{i,h+1})
$$
$$
= \lambda \left( \mathbb{E}_{a_i \sim z_i(s)} \left[ r_{i,h}^{\pi_{-i}}(s, a_i) \right] + \mathbb{E}_{a_i \sim z_i(s)} \left[ \inf_{\mathcal{U}^{\sigma_i} \left( P_{h,s,a_i}^{\pi_{-i}} \right)} P V_{i,h+1} \right] \right)
$$
$$
+ (1 - \lambda) \left( \mathbb{E}_{a_i \sim v_i(s)} \left[ r_{i,h}^{\pi_{-i}}(s, a_i) \right] + \mathbb{E}_{a_i \sim v_i(s)} \left[ \inf_{\mathcal{U}^{\sigma_i} \left( P_{h,s,a_i}^{\pi_{-i}} \right)} P V_{i,h+1} \right] \right)
$$
$$
= \mathbb{E}_{a_i \sim [\lambda z_i(s) + (1-\lambda) v_i(s)]} \left[ r_{i,h}^{\pi_{-i}}(s, a_i) \right] + \mathbb{E}_{a_i \sim [\lambda z_i(s) + (1-\lambda) v_i(s)]} \left[ \inf_{\mathcal{U}^{\sigma_i} \left( P_{h,s,a_i}^{\pi_{-i}} \right)} P V_{i,h+1} \right]
$$
$$
= f_{i,s}(\lambda z_i(s) + (1 - \lambda) v_i(s), \pi_{-i}(s); V_{i,h+1}). \tag{40}
$$

where we denote $r_{i,h}^{\pi_{-i}}(s, a_i) := \mathbb{E}_{\boldsymbol{a}_{-i} \sim \pi_{-i}(s)} [r_{i,h}(s, (a_i, \boldsymbol{a}_{-i}))]$. Hence, we show that $\lambda z_i(s) + (1 - \lambda) v_i(s) \in \phi(\pi)$ for all $(i, s) \in [n] \times \mathcal{S}$ and $0 \leq \lambda \leq 1$, thus verify that $\phi(\pi)$ is convex for any $\pi \in X$.

**Step 4: the existence of robust NE in RMGs.** Armed with above results, now we consider a general form to show that there exists a policy $\pi : [H] \times \mathcal{S} \mapsto \prod_{i \in [n]} \Delta(\mathcal{A}_i)$ that satisfies

$$
\forall (i, h, s) \in [n] \times [H] \times \mathcal{S} : \quad V_{i,h}^{\pi, \sigma_i}(s) = V_{i,h}^{\star, \pi_{-i}, \sigma_i}(s). \tag{41}
$$

We shall prove this by induction.

- **The base case.** Starting with the final step $h = H$, we recall that by definition,

$$
\forall (i, s) \in [n] \times \mathcal{S} : \quad V_{i,H+1}^{\pi, \sigma_i}(s) = 0. \tag{42}
$$

  To apply the results in the one-step game constructed in Step 2, we consider the one-step game at $h = H$ and using the payoff function (cf. (34))

$$
\forall s \in \mathcal{S} : \quad f_{i,s}(\pi_i(s), \pi_{-i}(s); V_{i,H+1}^{\pi, \sigma_i}) = \mathbb{E}_{\boldsymbol{a} \sim \pi(s)}[r_{i,h}(s, \boldsymbol{a})]. \tag{43}
$$

  We know that there exists a policy $\pi$ so that

$$
\forall (i, s) \in [n] \times \mathcal{S} : \quad V_{i,H}^{\pi, \sigma_i}(s) = V_{i,H}^{\star, \pi_{-i}, \sigma_i}(s) \tag{44}
$$

  by setting $\pi_H$ as the NE of the one-step game.
- **Induction.** Assuming that there exists a policy $\pi$ so that for subsequent steps $h+1, \cdots, H$,

$$
\forall (i, h, s) \in [n] \times \{h + 1, \cdots, H\} \times \mathcal{S} : \quad V_{i,h}^{\pi, \sigma_i}(s) = V_{i,h}^{\star, \pi_{-i}, \sigma_i}(s), \tag{45}
$$

  which are achieved by determining certain policies for $\{\pi_{h+1}, \pi_{h+2}, \cdots, \pi_H\}$. We are supposed to prove that at time step $h$, we can ensure our policy $\pi$ satisfying

$$
\forall (i, s) \in [n] \times \mathcal{S} : \quad V_{i,h}^{\pi, \sigma_i}(s) = V_{i,h}^{\star, \pi_{-i}, \sigma_i}(s) \tag{46}
$$

  by choosing a proper policy $\pi_h$ at the time step $h$.
  Towards this, it is observed that

$$
V_{i,h}^{\star, \pi_{-i}, \sigma_i}(s)
$$

$$= \max_{\pi'_i:\mathcal{S}\times[H]\mapsto\Delta(\mathcal{A}_i)} V_{i,h}^{\pi'_i\times\pi_{-i},\sigma_i}(s)$$

$$= \max_{\pi'_i:\mathcal{S}\times[H]\mapsto\Delta(\mathcal{A}_i)} \mathbb{E}_{\boldsymbol{a}\sim\pi'_{i,h}(s)\times\pi_{-i,h}(s)}[r_{i,h}(s,\boldsymbol{a})]$$

$$+ \mathbb{E}_{a_i\sim\pi'_{i,h}(s)}\left[\inf_{P\in\mathcal{U}^{\sigma_i}\left(P_{h,s,a_i}^{\pi_{-i}}\right)} PV_{i,h+1}^{\pi'_i\times\pi_{-i},\sigma_i}\right]$$

$$= \max_{\pi'_{i,h}(s)\in\Delta(\mathcal{A}_i)} \mathbb{E}_{\boldsymbol{a}\sim\pi'_{i,h}(s)\times\pi_{-i,h}(s)}[r_{i,h}(s,\boldsymbol{a})]$$

$$+ \max_{\pi'_{i,h}(s)\in\Delta(\mathcal{A}_i)} \mathbb{E}_{a_i\sim\pi'_{i,h}(s)} \max_{\pi'_{i,h+}:\mathcal{S}\times h^+\mapsto\Delta(\mathcal{A}_i)}\left[\inf_{P\in\mathcal{U}^{\sigma_i}\left(P_{h,s,a_i}^{\pi_{-i}}\right)} PV_{i,h+1}^{\pi'_i\times\pi_{-i},\sigma_i}\right]$$

$$= \max_{\pi'_{i,h}(s)\in\Delta(\mathcal{A}_i)} \mathbb{E}_{\boldsymbol{a}\sim\pi'_{i,h}(s)\times\pi_{-i,h}(s)}[r_{i,h}(s,\boldsymbol{a})] + \mathbb{E}_{a_i\sim\pi'_{i,h}(s)}\left[\inf_{P\in\mathcal{U}^{\sigma_i}\left(P_{h,s,a_i}^{\pi_{-i}}\right)} PV_{i,h+1}^{\star,\pi_{-i},\sigma_i}\right].$$

$$(47)$$

where we denote $h^+ = \{h+1, h+2, \cdots, H\}$ as the set that includes all the time steps after $h$ until the end of the episode, and the last equality follows from the fact

$$\max_{\pi'_{i,h+}:\mathcal{S}\times h^+\mapsto\Delta(\mathcal{A}_i)} \inf_{\mathcal{U}^{\sigma_i}\left(P_{h,s,a_i}^{\pi_{-i}}\right)} PV_{i,h+1}^{\pi'_i\times\pi_{-i},\sigma_i} = \inf_{\mathcal{U}^{\sigma_i}\left(P_{h,s,a_i}^{\pi_{-i}}\right)} P \max_{\pi'_{i,h+}:\mathcal{S}\times h^+\mapsto\Delta(\mathcal{A}_i)} V_{i,h+1}^{\pi'_i\times\pi_{-i},\sigma_i}$$

$$= \inf_{\mathcal{U}^{\sigma_i}\left(P_{h,s,a_i}^{\pi_{-i}}\right)} PV_{i,h+1}^{\star,\pi_{-i},\sigma_i}, \quad (48)$$

which holds by the definition of $V_{i,h+1}^{\star,\pi_{-i},\sigma_i}$. Now invoking the results in the auxiliary one-step game with $V_{i,h+1} = V_{i,h+1}^{\star,\pi_{-i},\sigma_i}$, one has that there exists a policy with $\pi_h$ that satisfies

$$\forall(i,s)\in[n]\times\mathcal{S}: \quad V_{i,h}^{\pi,\sigma_i}(s) = V_{i,h}^{\star,\pi_{-i},\sigma_i}(s). \quad (49)$$

Combining the results in the base case and induction, we complete the proof by recursively choosing $\pi_h : \mathcal{S} \mapsto \prod_{i\in[n]}\Delta(\mathcal{A}_i)$ for $h = H, H-1, \cdots, 1$ as the NE of the corresponding one-step game at time step $h$ and arrive at

$$\forall(i,s)\in[n]\times\mathcal{S}: \quad V_{i,1}^{\pi,\sigma_i}(s) = V_{i,1}^{\star,\pi_{-i},\sigma_i}(s). \quad (50)$$

## C.2 PROOF OF AUXILIARY FACTS

### C.2.1 PROOF OF LEMMA 1

The proof is obtained by recursively showing that for each $(h, s)$, there exist a policy. Then the product policy of them will be that final policy

Without loss of generality, we consider any $i \in [n]$ with the other agents' policy $\pi_i : \mathcal{S} \times [H] \mapsto \Delta(\mathcal{A}_i)$ fixed. We shall prove this lemma by induction.

- **The base case.** Consider the base case $h = H$. Conditioned on the other agents' policy $\pi_i : \mathcal{S} \times [H] \mapsto \Delta(\mathcal{A}_{-i})$, the maximum of the robust value function of the $i$-th agent can be expressed by

$$\forall s \in \mathcal{S}: \quad V_{i,H}^{\star,\pi_{-i},\sigma_i}(s) = \max_{\pi'_i:\mathcal{S}\times[H]\mapsto\Delta(\mathcal{A}_i)} V_{i,H}^{\pi'_i\times\pi_{-i},\sigma_i}(s)$$

$$= \max_{\pi'_i:\mathcal{S}\times[H]\mapsto\Delta(\mathcal{A}_i)} \mathbb{E}_{a_i\sim\pi'_{i,H}(s)}\left[\mathbb{E}_{\boldsymbol{a}_{-i}\sim\pi_{-i,H}(s)}[r_{i,H}(s,\boldsymbol{a})]\right]$$

$$= \max_{\pi'_{i,H}(s)\sim\Delta(\mathcal{A}_i)} \mathbb{E}_{a_i\sim\pi'_{i,H}(s)}\left[\mathbb{E}_{\boldsymbol{a}_{-i}\sim\pi_{-i,H}(s)}[r_{i,H}(s,\boldsymbol{a})]\right]. \quad (51)$$

Since the maximum of the continuous function $\mathbb{E}_{a_i\sim\pi'_{i,H}(s)}\left[\mathbb{E}_{\boldsymbol{a}_{-i}\sim\pi_{-i,H}(s)}[r_{i,H}(s,\boldsymbol{a})]\right]$ on a compact set $\Delta(\mathcal{A}_i)$ exists, by setting

$$\forall s \in \mathcal{S}: \quad \widetilde{\pi}_{i,H}(s) = \mathrm{argmax}_{\pi'_{i,H}(s)\sim\Delta(\mathcal{A}_i)}\mathbb{E}_{a_i\sim\pi'_{i,H}(s)}\left[\mathbb{E}_{\boldsymbol{a}_{-i}\sim\pi_{-i,H}(s)}[r_{i,H}(s,\boldsymbol{a})]\right],$$

$$(52)$$

we arrive at

$$\forall s \in \mathcal{S}: \quad V_{i,H}^{\widetilde{\pi}_i \times \pi_{-i}, \sigma_i}(s) = V_{i,H}^{\star, \pi_{-i}, \sigma_i}(s). \tag{53}$$

This complete the proof for the base case.

- **Induction.** Assuming that for $t = h+1, h+2, \cdots, H$, we have

$$\forall s \in \mathcal{S}: \quad V_{i,t}^{\widetilde{\pi}_i \times \pi_{-i}, \sigma_i}(s) = V_{i,t}^{\star, \pi_{-i}, \sigma_i}(s). \tag{54}$$

Then, we want to prove for the step $h$, where the maximum of the robust value function of the $i$-th agent can be expressed as: for all $s \in \mathcal{S}$,

$$V_{i,h}^{\star, \pi_{-i}, \sigma_i}(s)$$

$$= \max_{\pi_i': \mathcal{S} \times [H] \mapsto \Delta(\mathcal{A}_i)} V_{i,h}^{\pi_i' \times \pi_{-i}, \sigma_i}(s)$$

$$= \max_{\pi_i': \mathcal{S} \times [H] \mapsto \Delta(\mathcal{A}_i)} \mathbb{E}_{a_i \sim \pi_{i,h}'(s)} \left[ \mathbb{E}_{\boldsymbol{a}_{-i} \sim \pi_{-i,h}(s)} [r_{i,h}(s, \boldsymbol{a})] \right]$$

$$+ \mathbb{E}_{a_i \sim \pi_{i,h}(s)} \left[ \inf_{\mathcal{U}_\rho^{\sigma_i} \left( P_{h,s,a_i}^{\pi_{-i}} \right)} P V_{i,h+1}^{\star, \pi_{-i}, \sigma_i} \right]$$

$$\overset{(i)}{=} \max_{\pi_i': \mathcal{S} \times [H] \mapsto \Delta(\mathcal{A}_i)} \mathbb{E}_{a_i \sim \pi_{i,h}'(s)} \left[ \mathbb{E}_{\boldsymbol{a}_{-i} \sim \pi_{-i,h}(s)} [r_{i,h}(s, \boldsymbol{a})] \right]$$

$$+ \mathbb{E}_{a_i \sim \pi_{i,h}(s)} \left[ \inf_{\mathcal{U}_\rho^{\sigma_i} \left( P_{h,s,a_i}^{\pi_{-i}} \right)} P V_{i,h+1}^{\widetilde{\pi}_i \times \pi_{-i}, \sigma_i} \right]$$

$$= \max_{\pi_{i,h}'(s) \sim \Delta(\mathcal{A}_i)} \mathbb{E}_{a_i \sim \pi_{i,h}'(s)} \left[ \mathbb{E}_{\boldsymbol{a}_{-i} \sim \pi_{-i,h}(s)} [r_{i,h}(s, \boldsymbol{a})] \right]$$

$$+ \mathbb{E}_{a_i \sim \pi_{i,h}(s)} \left[ \inf_{\mathcal{U}_\rho^{\sigma_i} \left( P_{h,s,a_i}^{\pi_{-i}} \right)} P V_{i,h+1}^{\widetilde{\pi}_i \times \pi_{-i}, \sigma_i} \right]. \tag{55}$$

where (i) holds by the induction assumption in (54). Similarly to the base case, the maximum of the continuous function $\mathbb{E}_{a_i \sim \pi_{i,h}'(s)} \left[ \mathbb{E}_{\boldsymbol{a}_{-i} \sim \pi_{-i,h}(s)} [r_{i,h}(s, \boldsymbol{a})] \right] + \mathbb{E}_{a_i \sim \pi_{i,h}(s)} \left[ \inf_{\mathcal{U}_\rho^{\sigma_i} \left( P_{h,s,a_i}^{\pi_{-i}} \right)} P V_{i,h+1}^{\widetilde{\pi}_i \times \pi_{-i}, \sigma_i} \right]$ on a compact set $\Delta(\mathcal{A}_i)$ exists. So without conflict, for all $s \in \mathcal{S}$, we can set

$$\widetilde{\pi}_{i,h}(s)$$

$$= \operatorname{argmax}_{\pi_{i,h}'(s) \sim \Delta(\mathcal{A}_i)} \mathbb{E}_{a_i \sim \pi_{i,h}'(s)} \left[ \mathbb{E}_{\boldsymbol{a}_{-i} \sim \pi_{-i,h}(s)} [r_{i,h}(s, \boldsymbol{a})] \right]$$

$$+ \mathbb{E}_{a_i \sim \pi_{i,h}(s)} \left[ \inf_{\mathcal{U}_\rho^{\sigma_i} \left( P_{h,s,a_i}^{\pi_{-i}} \right)} P V_{i,h+1}^{\widetilde{\pi}_i \times \pi_{-i}, \sigma_i} \right], \tag{56}$$

since the function $\inf_{\mathcal{U}_\rho^{\sigma_i} \left( P_{h,s,a_i}^{\pi_{-i}} \right)} P V_{i,h+1}^{\widetilde{\pi}_i \times \pi_{-i}, \sigma_i}$ and especially $V_{i,h+1}^{\widetilde{\pi}_i \times \pi_{-i}, \sigma_i}$ are independent from the policy in the first $h$ steps ($\{\widetilde{\pi}_{i,t}(s)\}_{s \in \mathcal{S}, t \in [h]}$).
Consequently, (56) directly implies that

$$\forall s \in \mathcal{S}: \quad V_{i,h}^{\widetilde{\pi}_i \times \pi_{-i}, \sigma_i}(s) = V_{i,h}^{\star, \pi_{-i}, \sigma_i}(s). \tag{57}$$

Combining the results in base case and the induction, we complete the proof by showing that

$$\forall (h, s) \in [H] \times \mathcal{S}: \quad V_{i,h}^{\widetilde{\pi}_i \times \pi_{-i}, \sigma_i}(s) = V_{i,h}^{\star, \pi_{-i}, \sigma_i}(s). \tag{58}$$

### C.2.2 PROOF OF LEMMA 2

First, we define the distance between any two policy $\pi, \pi' \in X = \{\pi : \mathcal{S} \mapsto \prod_{i \in [n]} \Delta(A_i)\}$ as below:

$$d(\pi, \pi') := \max_{i \in [n]} \max_{(s, a_i) \in \mathcal{S} \times \mathcal{A}_i} |\pi_i(a_i \mid s) - \pi_i'(a_i \mid s)|. \tag{59}$$

To prove the continuity, given any $\epsilon > 0$, we want to show that there exists $\delta(\epsilon) > 0$ such that if

$$d(\pi, \pi') < \delta(\epsilon), \tag{60}$$

then

$$\left| f_{i,s}(\pi_i(s), \pi_{-i}(s); V_{i,h+1}) - f_{i,s}(\pi'_i(s), \pi'_{-i}(s); V_{i,h+1}) \right| < \epsilon \tag{61}$$

for any fixed $\{V_{i,h+1}\}_{i \in [n]}$ with $0 \leq V_{i,h+1} \leq H$ for all $i \in [n]$. Towards this, we observe that

$$\left| f_{i,s}(\pi_i(s), \pi_{-i}(s); V_{i,h+1}) - f_{i,s}(\pi'_i(s), \pi'_{-i}(s); V_{i,h+1}) \right|$$

$$= \left| \mathbb{E}_{\boldsymbol{a} \sim \pi(s)}[r_{i,h}(s, \boldsymbol{a})] + \mathbb{E}_{a_i \sim \pi_i(s)} \left[ \inf_{\mathcal{U}^{\sigma_i}\left(P^{\pi_{-i}}_{h,s,a_i}\right)} PV_{i,h+1} \right] \right.$$

$$\left. - \mathbb{E}_{\boldsymbol{a} \sim \pi'(s)}[r_{i,h}(s, \boldsymbol{a})] + \mathbb{E}_{a_i \sim \pi'_i(s)} \left[ \inf_{\mathcal{U}^{\sigma_i}\left(P^{\pi'_{-i}}_{h,s,a_i}\right)} PV_{i,h+1} \right] \right|$$

$$\leq \left| \mathbb{E}_{\boldsymbol{a} \sim \pi(s)}[r_{i,h}(s, \boldsymbol{a})] - \mathbb{E}_{\boldsymbol{a} \sim \pi'(s)}[r_{i,h}(s, \boldsymbol{a})] \right|$$

$$+ \left| \mathbb{E}_{a_i \sim \pi_i(s)} \left[ \inf_{\mathcal{U}^{\sigma_i}\left(P^{\pi_{-i}}_{h,s,a_i}\right)} PV_{i,h+1} \right] - \mathbb{E}_{a_i \sim \pi'_i(s)} \left[ \inf_{\mathcal{U}^{\sigma_i}\left(P^{\pi'_{-i}}_{h,s,a_i}\right)} PV_{i,h+1} \right] \right|. \tag{62}$$

The first term can be bounded by

$$\left| \mathbb{E}_{\boldsymbol{a} \sim \pi(s)}[r_{i,h}(s, \boldsymbol{a})] - \mathbb{E}_{\boldsymbol{a} \sim \pi'(s)}[r_{i,h}(s, \boldsymbol{a})] \right|$$

$$\leq \sum_{\boldsymbol{a} \in \mathcal{A}} \left| \prod_{i \in [n]} \pi_i(a_i \mid s) - \prod_{i \in [n]} \pi'_i(a_i \mid s) \right| \max_{(s, \boldsymbol{a}) \in \mathcal{S} \times \mathcal{A}} r_{i,h}(s, \boldsymbol{a})$$

$$\leq \sum_{\boldsymbol{a} \in \mathcal{A}} \left| \prod_{i \in [n]} \pi_i(a_i \mid s) - \prod_{i \in [n]} \pi'_i(a_i \mid s) \right|, \tag{63}$$

where the last inequality holds by the definition of reward function $\max_{(s, \boldsymbol{a}) \in \mathcal{S} \times \mathcal{A}} r_{i,h}(s, \boldsymbol{a}) \leq 1$ for all $(i, h) \in [n] \times [H]$. To continue, we first define the difference between $\delta_i(s, a_i) := \pi'_i(a_i \mid s) - \pi_i(a_i \mid s)$. Therefore, we have

$$\left| \prod_{i \in [n]} \pi_i(a_i \mid s) - \prod_{i \in [n]} \pi'_i(a_i \mid s) \right|$$

$$= \left| \prod_{i \in [n]} \pi_i(a_i \mid s) - \prod_{i \in [n]} (\pi_i(a_i \mid s) + \delta_i(s, a_i)) \right|$$

$$= \left| \sum_{|\mathcal{Y}| \geq 1, \mathcal{Y} \subseteq [n]} \left( \prod_{i \in \mathcal{Y}} \delta_i(s, a_i) \right) \cdot \left( \prod_{i \in \mathcal{Y}^c} \pi_i(a_i \mid s) \right) \right|$$

$$\leq \sum_{|\mathcal{Y}| \geq 1, \mathcal{Y} \subseteq [n]} \left| \left( \prod_{i \in \mathcal{Y}} \delta_i(s, a_i) \right) \cdot \left( \prod_{i \in \mathcal{Y}^c} \pi_i(a_i \mid s) \right) \right| \leq (2^n - 1)\delta(\epsilon), \tag{64}$$

where the last inequality holds by (60). Plugging (64) back to (63) indicates that

$$\left| \mathbb{E}_{\boldsymbol{a} \sim \pi(s)}[r_{i,h}(s, \boldsymbol{a})] - \mathbb{E}_{\boldsymbol{a} \sim \pi'(s)}[r_{i,h}(s, \boldsymbol{a})] \right| \leq \prod_{i \in [n]} A_i (2^n - 1)\delta(\epsilon). \tag{65}$$

For the second term in (62), we observe that

$$\left| \mathbb{E}_{a_i \sim \pi_i(s)} \left[ \inf_{P \in \mathcal{U}^{\sigma_i}\left(P^{\pi_{-i}}_{h,s,a_i}\right)} PV_{i,h+1} \right] - \mathbb{E}_{a_i \sim \pi'_i(s)} \left[ \inf_{P \in \mathcal{U}^{\sigma_i}\left(P^{\pi'_{-i}}_{h,s,a_i}\right)} PV_{i,h+1} \right] \right|$$

$$\leq \left| \mathbb{E}_{a_i \sim \pi_i(s)} \left[ \inf_{P \in \mathcal{U}^{\sigma_i}\left(P^{\pi_{-i}}_{h,s,a_i}\right)} PV_{i,h+1} \right] - \mathbb{E}_{a_i \sim \pi_i(s)} \left[ \inf_{P \in \mathcal{U}^{\sigma_i}\left(P^{\pi'_{-i}}_{h,s,a_i}\right)} PV_{i,h+1} \right] \right|$$

$$+ \left| \mathbb{E}_{a_i \sim \pi_i(s)} \left[ \inf_{P \in \mathcal{U}^{\sigma_i}\left(P^{\pi'_{-i}}_{h,s,a_i}\right)} PV_{i,h+1} \right] - \mathbb{E}_{a_i \sim \pi'_i(s)} \left[ \inf_{P \in \mathcal{U}^{\sigma_i}\left(P^{\pi'_{-i}}_{h,s,a_i}\right)} PV_{i,h+1} \right] \right|$$

$$\overset{(i)}{\leq} \mathbb{E}_{a_i \sim \pi_i(s)} \left[ \max_{\alpha \in [\min_s V_{i,h+1}(s), \max_s V_{i,h+1}(s)]} \left| \mathbb{E}_{\pi_{-i}(\boldsymbol{a}_{-i} \mid s)} \left[ P^0_{h,s,(a_i,\boldsymbol{a}_{-i})} \right] [V_{i,h+1}]_\alpha \right. \right.$$

$$\left. \left. - \mathbb{E}_{\pi'_{-i}(\boldsymbol{a}_{-i} \mid s)} \left[ P^0_{h,s,(a_i,\boldsymbol{a}_{-i})} \right] [V_{i,h+1}]_\alpha \right| \right] + \sum_{a_i \in \mathcal{A}_i} \left| \pi'_i(a_i \mid s) - \pi_i(a_i \mid s) \right| \inf_{P \in \mathcal{U}^{\sigma_i}\left(P^{\pi'_{-i}}_{h,s,a_i}\right)} PV_{i,h+1}$$

$$\overset{(ii)}{\leq} \sum_{\boldsymbol{a}_{-i} \in \mathcal{A}_i} \left| \prod_{j \neq i} \pi_j(a_j \mid s) - \prod_{j \neq i} \pi'_j(a_j \mid s) \right| H + H A_i \Delta(\epsilon)$$

$$\overset{(iii)}{\leq} H \prod_{j \neq i, j \in [n]} A_j (2^{n-1} - 1) \delta(\epsilon) + H A_i \Delta(\epsilon) \leq 2H \prod_{i \in [n]} A_i (2^n - 1) \cdot \delta(\epsilon), \tag{66}$$

where the first inequality holds by the triangle inequality, and (i) follows from applying the dual form of TV distance

$$\inf_{\mathcal{P} \in U^{\sigma_i}(P)} \mathcal{P} V = \max_{\alpha \in [\min_s V(s), \max_s V(s)]} \left\{ P[V]_\alpha - \sigma_i \left( \alpha - \min_{s'} [V]_\alpha(s') \right) \right\}, \tag{67}$$

and the maximum operator is 1-Lipschitz, (ii) arises from the fact that $\|V_{i,h+1}\|_\infty \leq H$, and (iii) can be verified by following the same pipeline of (64). Combining (65) and (66), one has

$$\left| f_{i,s}(\pi_i(s), \pi_{-i}(s); V_{i,h+1}) - f_{i,s}(\pi'_i(s), \pi'_{-i}(s); V_{i,h+1}) \right| \leq 3H \prod_{i \in [n]} A_i (2^n - 1) \cdot \delta(\varepsilon). \tag{68}$$

Consequently, letting $\delta_1(\epsilon) = \frac{\min\{\epsilon, 1\}}{3H \prod_{i \in [n]} A_i (2^n - 1)}$, we have when $d(\pi, \pi') < \delta_1(\epsilon)$,

$$\left| f_{i,s}(\pi_i(s), \pi_{-i}(s); V_{i,h+1}) - f_{i,s}(\pi'_i(s), \pi'_{-i}(s); V_{i,h+1}) \right| < \epsilon.$$

### C.2.3 PROOF OF LEMMA 3

Without loss of generality, we consider any $i \in [n]$. Consider $x_{-i} : \mathcal{S} \mapsto \prod_{j \neq i, j \in [n]} \Delta(\mathcal{A}_j)$ and $y_{-i} : \mathcal{S} \mapsto \prod_{j \neq i, j \in [n]} \Delta(\mathcal{A}_j)$. Before continuing, for all $s \in \mathcal{S}$, we denote

$$u^\star_{i,s} \coloneqq \operatorname{argmax}_{\pi'_i \in \Delta(\mathcal{S})} f_{i,s}(\pi'_i(s), x_{-i}(s); V_{i,h+1}),$$
$$v^\star_{i,s} \coloneqq \operatorname{argmax}_{\pi'_i \in \Delta(\mathcal{S})} f_{i,s}(\pi'_i(s), y_{-i}(s); V_{i,h+1}). \tag{69}$$

Then we have for any $s \in \mathcal{S}$,

$$g_{i,s}(x_{-i}(s), V_{i,h+1}) - g_{i,s}(y_{-i}(s), V_{i,h+1})$$
$$= \max_{\pi'_i \in \Delta(\mathcal{S})} f_{i,s}(\pi'_i(s), x_{-i}(s); V_{i,h+1}) - \max_{\pi'_i \in \Delta(\mathcal{S})} f_{i,s}(\pi'_i(s), y_{-i}(s); V_{i,h+1})$$
$$= f_{i,s}(u^\star_{i,s}, x_{-i}(s); V_{i,h+1}) - f_{i,s}(v^\star_{i,s}, y_{-i}(s); V_{i,h+1})$$
$$\leq f_{i,s}(u^\star_{i,s}, x_{-i}(s); V_{i,h+1}) - f_{i,s}(u^\star_{i,s}, y_{-i}(s); V_{i,h+1}) \quad \to 0 \quad \text{as} \quad y_{-i}(s) \to x_{-i}(s), \tag{70}$$

where the last line holds by Lemma (2) which shows that the function $f_{i,s}$ is continuous. Similarly, one has

$$g_{i,s}(x_{-i}(s), V_{i,h+1}) - g_{i,s}(y_{-i}(s), V_{i,h+1})$$
$$\geq f_{i,s}(v^\star_{i,s}, x_{-i}(s); V_{i,h+1}) - f_{i,s}(u^\star_{i,s}, y_{-i}(s); V_{i,h+1}) \quad \to 0 \quad \text{as} \quad y_{-i}(s) \to x_{-i}(s). \tag{71}$$

We complete the proof by showing that

$$|g_{i,s}(x_{-i}(s), V_{i,h+1}) - g_{i,s}(y_{-i}(s), V_{i,h+1})| \to 0 \quad \text{as} \quad y_{-i}(s) \to x_{-i}(s). \tag{72}$$

## D PROOF OF THEOREM 2

We will present the proof of Theorem 2 by first outlining the proof structure, followed by a step-by-step explanation of the key components. Auxiliary proofs will be provided at the end of this section.

### D.1 PROOF PIPELINE

To proof Theorem 2, recall the goal is to show that

$$\forall (i,s) \in [n] \times \mathcal{S}: \quad \mathbb{E}_{\pi \sim \widehat{\xi}}\left[V_{i,1}^{\star,\pi_{-i},\sigma_i}(s)\right] - \mathbb{E}_{\pi \sim \widehat{\xi}}\left[V_{i,1}^{\pi,\sigma_i}(s)\right] \leq \varepsilon, \tag{73}$$

where $\widehat{\xi} = \{\widehat{\xi}_h\}_{h \in [H]}$ is the output distribution over the set of policies $\{\pi_h^k = (\pi_{1,h}^k \times \cdots \times \pi_{n,h}^k)\}_{k \in [K], h \in [H]}$ from Algorithm 2. Namely, $\pi \sim \widehat{\xi}$ means

$$\forall h \in [H]: \quad \pi_h \sim \widehat{\xi}_h, \quad \text{where} \quad \widehat{\xi}_h(\pi_h^k) = \alpha_k^K. \tag{74}$$

We first introduce the best-response policy for player $i$:

$$\tilde{\pi}_i^\star = [\tilde{\pi}_{i,h}^\star]_{h \in [H]} := \arg\max_{\pi_i': \mathcal{S} \times [H] \to \Delta(\mathcal{A}_i)} \mathbb{E}_{\pi \sim \widehat{\xi}}\left[V_{i,1}^{\pi_i',\pi_{-i}}\right].$$

Recall that value function $\mathbb{E}_{\pi \sim \widehat{\xi}}\left[V_{i,h}^{\pi,\sigma_i}\right]$ satisfies the following Bellman equation for all $(i,s,h) \in [n] \times \mathcal{S} \times [H]$:

$$\mathbb{E}_{\pi \sim \widehat{\xi}}\left[V_{i,H+1}^{\pi,\sigma_i}(s)\right] = 0,$$

$$\mathbb{E}_{\pi \sim \widehat{\xi}}\left[V_{i,h}^{\pi,\sigma_i}(s)\right]$$

$$= \mathbb{E}_{\pi \sim \widehat{\xi}}\left\{\sum_{\mathbf{a} \in \mathcal{A}} \pi_h(\mathbf{a} \mid s) r_{i,h}(s,\mathbf{a}) + \mathbb{E}_{a_i \sim \pi_{i,h}}\left[\inf_{\mathcal{P} \in \mathcal{U}_i^{\sigma_i}\left(P_{h,s,a_i}^{\pi_{-i}}\right)} \mathcal{P} \mathbb{E}_{\pi \sim \widehat{\xi}}\left[V_{i,h+1}^{\pi,\sigma_i}\right]\right]\right\},$$

$$= \sum_{k=1}^{K} \sum_{\mathbf{a} \in \mathcal{A}} \alpha_k^K \pi_h^k(\mathbf{a} \mid s) r_{i,h}(s,\mathbf{a}) + \sum_{k=1}^{K} \alpha_k^K \mathbb{E}_{a_i \sim \pi_{i,h}^k}\left[\inf_{\mathcal{P} \in \mathcal{U}_i^{\sigma_i}\left(P_{h,s,a_i}^{\pi_{-i}^k}\right)} \mathcal{P} \mathbb{E}_{\pi \sim \widehat{\xi}}\left[V_{i,h+1}^{\pi,\sigma_i}\right]\right],$$

where $P_{h,s,a_i}^{\pi_{-i}^k}$ is defined as:

$$P_{h,s,a_i}^{\pi_{-i}^k} = \mathbb{E}_{\mathbf{a}_{-i} \sim \pi_{-i,h}^k(\cdot \mid s)}\left[P_{h,s,(a_i,\mathbf{a}_{-i})}^0\right] = \sum_{\mathbf{a}_{-i} \in \mathcal{A}_{-i}} \pi_{-i,h}^k(\mathbf{a}_{-i} \mid s)\left[P_{h,s,(a_i,\mathbf{a}_{-i})}^0\right].$$

We decompose the error in the value functions as follows:

$$\mathbb{E}_{\pi \sim \widehat{\xi}}\left[V_{i,h}^{\star,\pi_{-i}}\right] - \mathbb{E}_{\pi \sim \widehat{\xi}}\left[V_{i,h}^{\pi}\right]$$

$$\leq \underbrace{\mathbb{E}_{\pi \sim \widehat{\xi}}\left[V_{i,h}^{\star,\pi_{-i}}\right] - \mathbb{E}_{\pi \sim \widehat{\xi}}\left[\overline{V}_{i,h}^{\tilde{\pi}_i^\star,\pi_{-i}}\right]}_{A} + \underbrace{\mathbb{E}_{\pi \sim \widehat{\xi}}\left[\overline{V}_{i,h}^{\star,\pi_{-i}}\right] - \mathbb{E}_{\pi \sim \widehat{\xi}}\left[\overline{V}_{i,h}^{\pi}\right]}_{B} \tag{75}$$

$$+ \underbrace{\mathbb{E}_{\pi \sim \widehat{\xi}}\left[\overline{V}_{i,h}^{\pi}\right] - \mathbb{E}_{\pi \sim \widehat{\xi}}\left[V_{i,h}^{\pi}\right]}_{C}.$$

We define the following auxiliary value functions for all $s \in \mathcal{S}$:

$$\mathbb{E}_{\pi \sim \widehat{\xi}}\left[\overline{V}_{i,h}^{\pi}(s)\right]$$

$$= \sum_{k=1}^{K} \alpha_k^K \mathbb{E}_{a_i \sim \pi_{i,h}^k(s)}\left[r_{i,h}^k(s,a_i)\right] + \sum_{k=1}^{K} \alpha_k^K \mathbb{E}_{a_i \sim \pi_{i,h}^k(s)}\left[\inf_{\mathcal{P} \in \mathcal{U}^{\sigma_i}\left(P_{i,h,s,a_i}^k\right)} \mathcal{P} \mathbb{E}_{\pi \sim \widehat{\xi}}\left[\overline{V}_{i,h+1}^{\pi}\right]\right], \tag{76a}$$

$$\mathbb{E}_{\pi \sim \widehat{\xi}}\left[\overline{V}_{i,h}^{\tilde{\pi}_i^\star,\pi_{-i}}(s)\right]$$

$$= \sum_{k=1}^{K} \alpha_k^K \mathbb{E}_{a_i \sim \tilde{\pi}_{i,h}^\star(s)}\left[r_{i,h}^k(s,a_i)\right] + \sum_{k=1}^{K} \alpha_k^K \mathbb{E}_{a_i \sim \tilde{\pi}_{i,h}^\star(s)}\left[\inf_{\mathcal{P} \in \mathcal{U}^{\sigma_i}\left(P_{i,h,s,a_i}^k\right)} \mathcal{P} \mathbb{E}_{\pi \sim \widehat{\xi}}\left[\overline{V}_{i,h+1}^{\tilde{\pi}_i^\star,\pi_{-i}}\right]\right], \tag{76b}$$

$$\mathbb{E}_{\pi\sim\widehat{\xi}}\left[\overline{V}_{i,h}^{\star,\pi_{-i}}(s)\right] = \max_{a_i\in\mathcal{A}_i}\sum_{k=1}^{K}\alpha_k^K\left[r_{i,h}^k(s,a_i) + \left(\inf_{\mathcal{P}\in\mathcal{U}^{\sigma_i}\left(P_{i,h,s,a_i}^k\right)}\mathcal{P}\mathbb{E}_{\pi\sim\widehat{\xi}}\left[\overline{V}_{i,h+1}^{\star,\pi_{-i}}\right]\right)\right],$$

$$(76c)$$

where for all $s\in\mathcal{S}$, we also have

$$\mathbb{E}_{\pi\sim\widehat{\xi}}\left[\overline{V}_{i,H+1}^{\pi}(s)\right] = \mathbb{E}_{\pi\sim\widehat{\xi}}\left[\overline{V}_{i,H+1}^{\tilde{\pi}_i^{\star},\pi_{-i}}(s)\right] = \mathbb{E}_{\pi\sim\widehat{\xi}}\left[\overline{V}_{i,H+1}^{\star,\pi_{-i}}(s)\right] = 0$$

Here, we use the fact that $\mathbb{E}_{\pi\sim\widehat{\xi}}\left[\overline{V}_{i,h}^{\star,\pi_{-i}}\right] \geq \mathbb{E}_{\pi\sim\widehat{\xi}}\left[\overline{V}_{i,h}^{\tilde{\pi}_i^{\star},\pi_{-i}}(s)\right]$. Using the error decomposition in (75), we will now individually bound the three terms, $A$, $B$, and $C$, in the following sections.

## D.2 CONTROLLING B: ADVERSARIAL ONLINE LEARNING

### D.2.1 STEP 1: SHOWING THAT $\widehat{V}_{i,h}$ IS AN ENTRY-WISE UPPER BOUND ON $\mathbb{E}_{\pi\sim\widehat{\xi}}\left[\overline{V}_{i,h}^{\star,\pi_{-i}}\right]$

The following lemma demonstrates that the value estimate $\widehat{V}_{i,h}$ for the $i^{\text{th}}$ player serves as an optimistic estimate of the auxiliary value $\mathbb{E}_{\pi\sim\widehat{\xi}}\left[\overline{V}_{i,h}^{\star,\pi_{-i}}\right]$, as defined in (76).

**Lemma 4.** *With probability at least $1-\delta$, it holds that*

$$\widehat{V}_{i,h} \geq \mathbb{E}_{\pi\sim\widehat{\xi}}\left[\overline{V}_{i,h}^{\star,\pi_{-i}}\right], \qquad \text{for all } (i,h)\in[n]\times[H].$$

*Proof.* See Appendix D.4.1 □

The following lemma demonstrates that the value estimate $\widehat{V}_{i,h}$ for the $i^{\text{th}}$ player serves as an optimistic estimate of the auxiliary value $\mathbb{E}_{\pi\sim\widehat{\xi}}\left[\overline{V}_{i,h}^{\star,\pi_{-i}}\right]$, as defined in (76).

**Lemma 5.** *For value vector $\widehat{V}_{i,h}$ and $\mathbb{E}_{\pi\sim\widehat{\xi}}\left[\overline{V}_{i,h}^{\pi}\right]$, it holds that*

$$\widehat{V}_{i,h} \geq \mathbb{E}_{\pi\sim\widehat{\xi}}\left[\overline{V}_{i,h}^{\pi}\right], \qquad \text{for all } (i,h)\in[n]\times[H].$$

*Proof.* See Appendix D.4.2 □

### D.2.2 STEP 2: CONSTRUCTING RECURSION

To begin with, according to the definition of $\widehat{V}_{i,h}(s)$ and $\mathbb{E}_{\pi\sim\widehat{\xi}}\left[\overline{V}_{i,h}^{\pi}(s)\right]$, we have

$$\widehat{V}_{i,h}(s) - \mathbb{E}_{\pi\sim\widehat{\xi}}\left[\overline{V}_{i,h}^{\pi}(s)\right]$$

$$= \min\left\{\sum_{k=1}^{K}\alpha_k^K\mathbb{E}_{a_i\sim\pi_{i,h}^k}\left[r_{i,h}^k(s,a_i) + \inf_{\mathcal{P}\in\mathcal{U}^{\sigma_i}\left(P_{i,h,s,a_i}^k\right)}\mathcal{P}\widehat{V}_{i,h+1}\right] + \beta_{i,h}(s), H-h+1\right\}$$

$$- \sum_{k=1}^{K}\alpha_k^K\mathbb{E}_{a_i\sim\pi_{i,h}^k}\left[r_{i,h}^k(s,a_i) + \inf_{\mathcal{P}\in\mathcal{U}^{\sigma_i}\left(P_{i,h,s,a_i}^k\right)}\mathcal{P}\mathbb{E}_{\pi\sim\widehat{\xi}}\left[\overline{V}_{i,h+1}^{\pi}\right]\right]$$

$$\leq \sum_{k=1}^{K}\alpha_k^K\mathbb{E}_{a_i\sim\pi_{i,h}^k}\left[r_{i,h}^k(s,a_i) + \inf_{\mathcal{P}\in\mathcal{U}^{\sigma_i}\left(P_{i,h,s,a_i}^k\right)}\mathcal{P}\widehat{V}_{i,h+1}\right] + \beta_{i,h}(s)$$

$$- \sum_{k=1}^{K}\alpha_k^K\mathbb{E}_{a_i\sim\pi_{i,h}^k}\left[r_{i,h}^k(s,a_i) + \inf_{\mathcal{P}\in\mathcal{U}^{\sigma_i}\left(P_{i,h,s,a_i}^k\right)}\mathcal{P}\mathbb{E}_{\pi\sim\widehat{\xi}}\left[\overline{V}_{i,h+1}^{\pi}\right]\right]$$

$$= \sum_{k=1}^{K} \alpha_k^K \mathbb{E}_{a_i \sim \pi_{i,h}^k} \left[ \inf_{\mathcal{P} \in \mathcal{U}^{\sigma_i}\left(P_{i,h,s,a_i}^k\right)} \mathcal{P} \widehat{V}_{i,h+1} \right] + \beta_{i,h}(s)$$

$$- \sum_{k=1}^{K} \alpha_k^K \mathbb{E}_{a_i \sim \pi_{i,h}^k} \left[ \inf_{\mathcal{P} \in \mathcal{U}^{\sigma_i}\left(P_{i,h,s,a_i}^k\right)} \mathcal{P} \mathbb{E}_{\pi \sim \widehat{\xi}} \left[ \overline{V}_{i,h+1}^{\pi} \right] \right] \qquad (77)$$

To simplify the notations, we define transition kernel associated estimated value function similarly as (23). For all $k \in [K]$, we define matrix notations $\widehat{P}_{i,h}^{\pi^k, \widehat{V}}$ and $\widehat{P}_{i,h}^{\widehat{\pi}^k, \overline{V}}$ as:

$$\widehat{P}_{i,h}^{\pi^k, \widehat{V}} := \widehat{P}_{i,h}^{\pi_{-i}^k, \widehat{V}_{i,h+1}},$$

$$\widehat{P}_{i,h,s,a_i}^{\pi^k, \widehat{V}} := \widehat{P}_{i,h,s,a_i}^{\pi_{-i}^k, \widehat{V}_{i,h+1}} = \text{argmin}_{\mathcal{P} \in \mathcal{U}_\rho^{\sigma_i}\left(\widehat{P}_{i,h,s,a_i}^{\pi_{-i}^k}\right)} \mathcal{P} \widehat{V}_{i,h+1},$$

$$\widehat{P}_{i,h}^{\pi^k, \overline{V}} := \widehat{P}_{i,h}^{\pi_{-i}^k, \mathbb{E}_{\pi \sim \widehat{\xi}}\left[\overline{V}_{i,h+1}^\pi\right]}$$

$$\widehat{P}_{i,h,s,a_i}^{\pi^k, \overline{V}} := \widehat{P}_{i,h,s,a_i}^{\pi_{-i}^k, \mathbb{E}_{\pi \sim \widehat{\xi}}\left[\overline{V}_{i,h+1}^\pi\right]} = \text{argmin}_{\mathcal{P} \in \mathcal{U}_\rho^{\sigma_i}\left(\widehat{P}_{i,h,s,a_i}^{\pi_{-i}^k}\right)} \mathcal{P} \mathbb{E}_{\pi \sim \widehat{\xi}} \left[ \overline{V}_{i,h+1}^{\pi} \right].$$

Additionally, we define square matrices $\underline{\widehat{P}}_{i,h}^{\pi^k, \overline{V}} \in \mathbb{R}^{S \times S}$ and $\underline{\widehat{P}}_{i,h}^{\pi^k, \widehat{V}} \in \mathbb{R}^{S \times S}$ as: $\underline{\widehat{P}}_{i,h}^{\pi^k, \overline{V}} := \Pi_h^{\pi_i^k} \widehat{P}_{i,h}^{\pi_{-i}^k, \overline{V}}$ and $\underline{\widehat{P}}_{i,h}^{\pi^k, \widehat{V}} := \Pi_h^{\pi_i^k} \widehat{P}_{i,h}^{\pi_{-i}^k, \widehat{V}}$. We rewrite the result of (77) in a vector form, we can obtain that

$$\widehat{V}_{i,h} - \mathbb{E}_{\pi \sim \widehat{\xi}} \left[ \overline{V}_{i,h}^{\pi} \right]$$

$$\leq \sum_{k=1}^{K} \alpha_k^K \Pi_h^{\pi_i} \left[ \inf_{\mathcal{P} \in \mathcal{U}^{\sigma_i}\left(\widehat{P}_{i,h,s,a_i}^{\pi_{-i}^k}\right)} \mathcal{P} \widehat{V}_{i,h+1} \right] + \beta_{i,h} - \sum_{k=1}^{K} \alpha_k^K \Pi_h^{\pi_i} \left[ \inf_{\mathcal{P} \in \mathcal{U}^{\sigma_i}\left(\widehat{P}_{i,h,s,a_i}^{\pi_{-i}^k}\right)} \mathcal{P} \overline{V}_{i,h+1}^{\pi} \right]$$

$$= \sum_{k=1}^{K} \alpha_k^K \underline{\widehat{P}}_{i,h}^{\pi^k, \widehat{V}} \widehat{V}_{i,h+1} + \beta_{i,h} - \sum_{k=1}^{K} \alpha_k^K \underline{\widehat{P}}_{i,h}^{\pi^k, \overline{V}} \mathbb{E}_{\pi \sim \widehat{\xi}} \left[ \overline{V}_{i,h+1}^{\pi} \right]$$

$$\leq \sum_{k=1}^{K} \alpha_k^K \underline{\widehat{P}}_{i,h}^{\pi^k, \overline{V}} \left( \widehat{V}_{i,h+1} - \mathbb{E}_{\pi \sim \widehat{\xi}} \left[ \overline{V}_{i,h+1}^{\pi} \right] \right) + \beta_{i,h}.$$

To continue, we first introduce an lemma of the upper bound for bonus vector $\beta_{i,h}$.

**Lemma 6.** *The bonus vector $\beta_{i,h}$ is bounded by the following inequality:*

$$\beta_{i,h} \leq 3c_{\mathsf{b}} \sqrt{\frac{\log^3\left(\frac{KS \sum_{i=1}^{n} A_i}{\delta}\right)}{KH}} \left( H \cdot 1 + \sum_{k=1}^{K} \alpha_k^K \mathsf{Var}_{\underline{\widehat{P}}_{i,h}^{\pi^k, \widehat{v}}} \widehat{V}_{i,h+1} \right)$$

*Proof.* See Appendix D.4.3 □

To proceed, we introduce some notations for convenience. Let $e_s$ denote the $S$-dimensional standard basis vector, with support on the $s$-th element. Additionally, we define:

$$b_h^h = e_s \quad \text{and} \quad b_h^j = e_s^\top \left[ \prod_{r=h}^{j-1} \left( \sum_{k=1}^{K} \alpha_k^K \underline{\widehat{P}}_{i,r}^{\pi^k, \overline{V}} \right) \right], \quad \forall j = h+1, \dots, H. \qquad (78)$$

Armed with above notations and fact, for any $s \in \mathcal{S}$, we have

$$\widehat{V}_{i,h}(s) - \mathbb{E}_{\pi \sim \widehat{\xi}} \left[ \overline{V}_{i,h}^{\pi}(s) \right] = \left\langle e_s, \widehat{V}_{i,h} - \mathbb{E}_{\pi \sim \widehat{\xi}} \left[ \overline{V}_{i,h}^{\pi} \right] \right\rangle = \sum_{j=h}^{H} \left\langle b_h^j, \beta_{i,j} \right\rangle$$

$$\leq \sum_{j=h}^{H} \left\langle b_h^j, 3c_{\mathsf{b}} H \sqrt{\frac{\log^3(\frac{KS\sum_{i=1}^n A_i}{\delta})}{KH}} 1 \right\rangle$$

$$+ \sum_{j=h}^{H} \sum_{k=1}^{K} \alpha_k^K \left\langle b_h^j, 3c_{\mathsf{b}} \sqrt{\frac{\log^3(\frac{KS\sum_{i=1}^n A_i}{\delta})}{KH}} \mathsf{Var}_{\widehat{\underline{P}}_{i,j}^{\pi^k,\widehat{v}}} \widehat{V}_{i,j+1} \right\rangle$$

$$= 3c_{\mathsf{b}} \sqrt{\frac{H^3 \log^3(\frac{KS\sum_{i=1}^n A_i}{\delta})}{K}} + 3c_{\mathsf{b}} \sqrt{\frac{\log^3(\frac{KS\sum_{i=1}^n A_i}{\delta})}{KH}} \sum_{j=h}^{H} \sum_{k=1}^{K} \alpha_k^K \left\langle b_h^j, \mathsf{Var}_{\widehat{\underline{P}}_{i,j}^{\pi^k,\widehat{v}}} \widehat{V}_{i,j+1} \right\rangle.$$
$$(79)$$

With elementary inequality $\sqrt{\mathsf{Var}_P(V+V')} \leq \sqrt{\mathsf{Var}_P(V)} + \sqrt{\mathsf{Var}_P(V')}$ for any transition kernel $P \in \mathbb{R}^S$ and vector $V, V' \in \mathbb{R}^S$, we further decompose (79) as

$$\widehat{V}_{i,h}(s) - \mathbb{E}_{\pi\sim\widehat{\xi}} \left[ \overline{V}_{i,h}^\pi(s) \right]$$

$$\leq 3c_{\mathsf{b}} \sqrt{\frac{H^3 \log^3(\frac{KS\sum_{i=1}^n A_i}{\delta})}{K}} + 3c_{\mathsf{b}} \sqrt{\frac{\log^3(\frac{KS\sum_{i=1}^n A_i}{\delta})}{KH}} \sum_{j=h}^{H} \sum_{k=1}^{K} \alpha_k^K \left\langle b_h^j, \mathsf{Var}_{\widehat{\underline{P}}_{i,j}^{\pi^k,\widehat{v}}} \widehat{V}_{i,j+1} \right\rangle$$

$$\leq 3c_{\mathsf{b}} \sqrt{\frac{\log^3(\frac{KS\sum_{i=1}^n A_i}{\delta})}{KH}} \sum_{k=1}^{K} \alpha_k^K \sum_{j=h}^{H} \left\langle b_h^j, \mathsf{Var}_{\widehat{\underline{P}}_{i,j}^{\pi^k,\widehat{v}}} \left( \widehat{V}_{i,j+1} - \mathbb{E}_{\pi\sim\widehat{\xi}} \left[ \overline{V}_{i,j+1}^\pi \right] \right) \right\rangle$$

$$+ 3c_{\mathsf{b}} \sqrt{\frac{\log^3(\frac{KS\sum_{i=1}^n A_i}{\delta})}{KH}} \left[ \sum_{k=1}^{K} \alpha_k^K \sum_{j=h}^{H} \left\langle b_h^j, \mathsf{Var}_{\widehat{\underline{P}}_{i,j}^{\pi^k,\widehat{v}}} \left( \mathbb{E}_{\pi\sim\widehat{\xi}} \left[ \overline{V}_{i,j+1}^\pi \right] \right) \right\rangle + H^2 \right]$$

$$\leq \mathcal{D}_1 + \mathcal{D}_2 + \mathcal{D}_3 + 3c_{\mathsf{b}} \sqrt{\frac{H^3 \log^3(\frac{KS\sum_{i=1}^n A_i}{\delta})}{K}},$$

where we define the three terms $\mathcal{D}_1, \mathcal{D}_2, \mathcal{D}_3$ as:

$$\mathcal{D}_1 = 3c_{\mathsf{b}} \sqrt{\frac{\log^3(\frac{KS\sum_{i=1}^n A_i}{\delta})}{KH}} \sum_{j=h}^{H} \sum_{k=1}^{K} \alpha_k^K \left\langle b_h^j, \mathsf{Var}_{\widehat{\underline{P}}_{i,j}^{\pi^k,\widehat{v}}} \left( \widehat{V}_{i,j+1} - \mathbb{E}_{\pi\sim\widehat{\xi}} \left[ \overline{V}_{i,j+1}^\pi \right] \right) \right\rangle$$

$$\mathcal{D}_2 = 3c_{\mathsf{b}} \sqrt{\frac{\log^3(\frac{KS\sum_{i=1}^n A_i}{\delta})}{KH}} \sum_{j=h}^{H} \sum_{k=1}^{K} \alpha_k^K \left\langle b_h^j, \mathsf{Var}_{\widehat{\underline{P}}_{i,j}^{\pi^k,\widehat{v}}} \left( \mathbb{E}_{\pi\sim\widehat{\xi}} \left[ \overline{V}_{i,j+1}^\pi \right] \right) \right\rangle$$

$$- 3c_{\mathsf{b}} \sqrt{\frac{\log^3(\frac{KS\sum_{i=1}^n A_i}{\delta})}{KH}} \sum_{j=h}^{H} \sum_{k=1}^{K} \alpha_k^K \left\langle b_h^j, \mathsf{Var}_{\widehat{\underline{P}}_{i,j}^{\pi^k,\overline{v}}} \left( \mathbb{E}_{\pi\sim\widehat{\xi}} \left[ \overline{V}_{i,j+1}^\pi \right] \right) \right\rangle$$

$$\mathcal{D}_3 = 3c_{\mathsf{b}} \sqrt{\frac{\log^3(\frac{KS\sum_{i=1}^n A_i}{\delta})}{KH}} \sum_{j=h}^{H} \sum_{k=1}^{K} \alpha_k^K \left\langle b_h^j, \mathsf{Var}_{\widehat{\underline{P}}_{i,j}^{\pi^k,\overline{v}}} \left( \mathbb{E}_{\pi\sim\widehat{\xi}} \left[ \overline{V}_{i,j+1}^\pi \right] \right) \right\rangle \qquad (80)$$

We now control the three terms $\mathcal{D}_1, \mathcal{D}_2, \mathcal{D}_3$ separately.

**Controlling $\mathcal{D}_1$.** We can directly obtain the following upper bound on $\mathcal{D}_1$:

$$\mathcal{D}_1 = 3c_{\mathsf{b}} \sqrt{\frac{\log^3(\frac{KS\sum_{i=1}^n A_i}{\delta})}{KH}} \sum_{j=h}^{H} \sum_{k=1}^{K} \alpha_k^K \left\langle b_h^j, \mathsf{Var}_{\widehat{\underline{P}}_{i,h}^{\pi^k,\widehat{v}}} \left( \widehat{V}_{i,j+1} - \mathbb{E}_{\pi\sim\widehat{\xi}} \left[ \overline{V}_{i,j+1}^\pi \right] \right) \right\rangle$$

$$\leq 3c_{\mathsf{b}} \sqrt{\frac{\log^3(\frac{KS\sum_{i=1}^n A_i}{\delta})}{KH}} \sum_{j=h}^{H} \sum_{k=1}^{K} \alpha_k^K \left\langle b_h^j, \left\| \mathsf{Var}_{\widehat{\underline{P}}_{i,h}^{\pi^k,\widehat{v}}} \left( \widehat{V}_{i,j+1} - \mathbb{E}_{\pi\sim\widehat{\xi}} \left[ \overline{V}_{i,j+1}^\pi \right] \right) \right\|_\infty \cdot 1 \right\rangle$$

$$\leq 3c_{\mathsf{b}} \sqrt{\frac{\log^3(\frac{KS\sum_{i=1}^n A_i}{\delta})}{KH}} \sum_{j=h}^{H} \sum_{k=1}^{K} \alpha_k^K \left\langle b_h^j, \left\| \widehat{V}_{i,j+1} - \mathbb{E}_{\pi\sim\widehat{\xi}} \left[ \overline{V}_{i,j+1}^\pi \right] \right\|_\infty^2 \cdot 1 \right\rangle$$

$$\overset{(i)}{\leq} 3c_{\mathsf{b}} \sqrt{\frac{H \log^3(\frac{KS \sum_{i=1}^n A_i}{\delta})}{K}} \sum_{j=h}^H \sum_{k=1}^K \alpha_k^K \left\langle b_h^j, \left\| \widehat{V}_{i,j+1} - \mathbb{E}_{\pi \sim \widehat{\xi}} \left[ \overline{V}_{i,j+1}^\pi \right] \right\|_\infty \cdot 1 \right\rangle$$

$$\leq 3c_{\mathsf{b}} \sqrt{\frac{H^3 \log^3(\frac{KS \sum_{i=1}^n A_i}{\delta})}{K}} \max_{h \leq j \leq H} \left\| \widehat{V}_{i,j+1} - \mathbb{E}_{\pi \sim \widehat{\xi}} \left[ \overline{V}_{i,j+1}^\pi \right] \right\|_\infty \tag{81}$$

where (i) follows from the elementary upper bound $\left\| \widehat{V}_{i,j+1} \right\|_\infty \leq H$, $\left\| \mathbb{E}_{\pi \sim \widehat{\xi}} \left[ \overline{V}_{i,j+1}^\pi \right] \right\|_\infty \leq H$ for all $h \leq j \leq H$.

Before deriving the upper bounds for the terms $\mathcal{D}_2$ and $\mathcal{D}_3$, we first introduce the following auxiliary lemmas, which will be instrumental in the subsequent derivation.

**Lemma 7.** *For all $(i, h) \in [n] \times [H]$, the estimated robust value function $\mathbb{E}_{\pi \sim \widehat{\xi}} \left[ \overline{V}_{i,h}^\pi \right]$ satisfies the following inequality:*

$$\max_{s \in \mathcal{S}} \mathbb{E}_{\pi \sim \widehat{\xi}} \left[ \overline{V}_{i,h}^\pi(s) \right] - \min_{s \in \mathcal{S}} \mathbb{E}_{\pi \sim \widehat{\xi}} \left[ \overline{V}_{i,h}^\pi(s) \right] \leq \min \left\{ \frac{1}{\sigma_i}, H - h + 1 \right\}.$$

*Proof.* See Appendix D.4.4. $\qquad\square$

With Lemma 7, we have the following lemma on variance base on different transition probability in the same uncertainty set, and we leave the proof to Appendix D.4.5.

**Lemma 8.** *For a transition kernel $P' \in \mathbb{R}^S$ and any $\widetilde{P} \in \mathbb{R}^S$ such that $\widetilde{P} \in \mathcal{U}^{\sigma_i}(P')$, the following bound holds for all $(i, h, ) \in [n] \times [H]$:*

$$\left| \mathsf{Var}_{P'} \left( \mathbb{E}_{\pi \sim \widehat{\xi}} \left[ \overline{V}_{i,h}^\pi \right] \right) - \mathsf{Var}_{\widetilde{P}} \left( \mathbb{E}_{\pi \sim \widehat{\xi}} \left[ \overline{V}_{i,h}^\pi \right] \right) \right| \leq \min \left\{ \frac{1}{\sigma_i}, H - h + 1 \right\}. \tag{82a}$$

**Controlling $\mathcal{D}_2$.** We can directly apply Lemma 8 and arrive at

$$\left| \mathsf{Var}_{\widehat{\underline{P}}_{i,h}^{\pi^k,\widehat{V}}} \left( \mathbb{E}_{\pi \sim \widehat{\xi}} \left[ \overline{V}_{i,h+1}^\pi \right] \right) - \mathsf{Var}_{\widehat{\underline{P}}_{i,h}^{\pi^k,\overline{V}}} \left( \mathbb{E}_{\pi \sim \widehat{\xi}} \left[ \overline{V}_{i,h+1}^\pi \right] \right) \right|$$

$$\leq \left| \mathsf{Var}_{\widehat{\underline{P}}_{i,h}^{\pi^k,\widehat{V}}} \left( \mathbb{E}_{\pi \sim \widehat{\xi}} \left[ \overline{V}_{i,h+1}^\pi \right] \right) - \mathsf{Var}_{\widehat{\underline{P}}_{i,h}^{\pi^k}} \left( \mathbb{E}_{\pi \sim \widehat{\xi}} \left[ \overline{V}_{i,h+1}^\pi \right] \right) \right|$$

$$+ \left| \mathsf{Var}_{\widehat{\underline{P}}_{i,h}^{\pi^k}} \left( \mathbb{E}_{\pi \sim \widehat{\xi}} \left[ \overline{V}_{i,h+1}^\pi \right] \right) - \mathsf{Var}_{\widehat{\underline{P}}_{i,h}^{\pi^k,\overline{V}}} \left( \mathbb{E}_{\pi \sim \widehat{\xi}} \left[ \overline{V}_{i,h+1}^\pi \right] \right) \right|$$

$$\leq 2 \min \left\{ \frac{1}{\sigma_i}, H \right\}.$$

We insert (83) back to the expression of $\mathcal{D}_2$, and we can obtain that

$$\mathcal{D}_2 = 3c_{\mathsf{b}} \sqrt{\frac{\log^3(\frac{KS \sum_{i=1}^n A_i}{\delta})}{KH}} \sum_{j=h}^H \sum_{k=1}^K \alpha_k^K \left\langle b_h^j, \mathsf{Var}_{\widehat{\underline{P}}_{i,j}^{\pi^k,\widehat{V}}} \left( \mathbb{E}_{\pi \sim \widehat{\xi}} \left[ \overline{V}_{i,j+1}^\pi \right] \right) \right\rangle$$

$$- 3c_{\mathsf{b}} \sqrt{\frac{\log^3(\frac{KS \sum_{i=1}^n A_i}{\delta})}{KH}} \sum_{j=h}^H \sum_{k=1}^K \alpha_k^K \left\langle b_h^j, \mathsf{Var}_{\widehat{\underline{P}}_{i,j}^{\pi^k,\overline{V}}} \left( \mathbb{E}_{\pi \sim \widehat{\xi}} \left[ \overline{V}_{i,j+1}^\pi \right] \right) \right\rangle$$

$$\leq 3c_{\mathsf{b}} \sqrt{\frac{\log^3(\frac{KS \sum_{i=1}^n A_i}{\delta})}{KH}} \sum_{j=h}^H \left\langle b_h^j, 2 \min \left\{ \frac{1}{\sigma_i}, H \right\} 1 \right\rangle$$

$$= 6c_{\mathsf{b}} \sqrt{\frac{H \log^3(\frac{KS \sum_{i=1}^n A_i}{\delta})}{K}} \min \left\{ \frac{1}{\sigma_i}, H \right\}. \tag{83}$$

**Controlling $\mathcal{D}_3$.** We first apply Lemma 12, and we can directly deduce that

$$\mathcal{D}_3 = 3c_{\mathsf{b}}\sqrt{\frac{\log^3(\frac{KS\sum_{i=1}^n A_i}{\delta})}{KH}}\sum_{j=h}^{H}\sum_{k=1}^{K}\alpha_k^K\left\langle b_h^j, \mathsf{Var}_{\underline{\widehat{P}}_{i,j}^{\pi^k,\overline{V}}}\left(\mathbb{E}_{\pi\sim\widehat{\xi}}\left[\overline{V}_{i,j+1}^{\pi}\right]\right)\right\rangle$$

$$\leq 3c_{\mathsf{b}}\sqrt{\frac{\log^3(\frac{KS\sum_{i=1}^n A_i}{\delta})}{KH}}\sum_{j=h}^{H}\left\langle b_h^j, \mathsf{Var}_{\sum_{k=1}^{K}\alpha_k^K\widehat{P}_{i,j}^{\pi^k,\overline{V}}}\left(\mathbb{E}_{\pi\sim\widehat{\xi}}\left[\overline{V}_{i,j+1}^{\pi}\right]\right)\right\rangle$$

We now introduce the following lemma on $\sum_{j=h}^{H}\left\langle b_h^j, \mathsf{Var}_{\sum_{k=1}^{K}\alpha_k^K\widehat{P}_{i,j}^{\pi^k,\overline{V}}}\mathbb{E}_{\pi\sim\widehat{\xi}}\left[\overline{V}_{i,j+1}^{\pi}\right]\right\rangle$, which is an empirical-transition version of Lemma 16.

**Lemma 9.** *Let $\delta \in (0,1)$. With probability at least $1 - \delta$, the following condition holds for all $(h,i) \in [H] \times [n]$:*

$$\sum_{j=h}^{H}\left\langle b_h^j, \mathsf{Var}_{\sum_{k=1}^{K}\alpha_k^K\widehat{P}_{i,j}^{\pi^k,\overline{V}}}\left(\mathbb{E}_{\pi\sim\widehat{\xi}}\left[\overline{V}_{i,j+1}^{\pi}\right]\right)\right\rangle$$

$$\leq 3H\left(\max_{s\in\mathcal{S}}\mathbb{E}_{\pi\sim\widehat{\xi}}\left[\overline{V}_{i,h}^{\pi}(s)\right] - \min_{s\in\mathcal{S}}\mathbb{E}_{\pi\sim\widehat{\xi}}\left[\overline{V}_{i,h}^{\pi}(s)\right]\right). \tag{84}$$

*Proof.* See Appendix D.4.6. $\qquad\square$

Therefore, we can further achieve the following upper bound of $\mathcal{D}_3$ by applying Lemma 9:

$$\mathcal{D}_3 \leq 3c_{\mathsf{b}}\sqrt{\frac{\log^3(\frac{KS\sum_{i=1}^n A_i}{\delta})}{KH}}\sum_{j=h}^{H}\left\langle b_h^j, \mathsf{Var}_{\sum_{k=1}^{K}\alpha_k^K\widehat{P}_{i,j}^{\pi^k,\overline{V}}}\left(\mathbb{E}_{\pi\sim\widehat{\xi}}\left[\overline{V}_{i,j+1}^{\pi}\right]\right)\right\rangle$$

$$\leq 9c_{\mathsf{b}}\sqrt{\frac{H\log^3(\frac{KS\sum_{i=1}^n A_i}{\delta})}{K}}\left(\max_{s\in\mathcal{S}}\mathbb{E}_{\pi\sim\widehat{\xi}}\left[\overline{V}_{i,h}^{\pi}(s)\right] - \min_{s\in\mathcal{S}}\mathbb{E}_{\pi\sim\widehat{\xi}}\left[\overline{V}_{i,h}^{\pi}(s)\right]\right)$$

$$\overset{(i)}{\leq} 9c_{\mathsf{b}}\sqrt{\frac{H\log^3(\frac{KS\sum_{i=1}^n A_i}{\delta})}{K}}\min\left\{\frac{1}{\sigma_i}, H\right\} \tag{85}$$

where (i) holds due to Lemma 8.

### D.2.3 STEP 3: SUMMING UP THE RESULT

We combine the result of (81), (83), (85), yielding

$$\widehat{V}_{i,h} - \mathbb{E}_{\pi\sim\widehat{\xi}}\left[\overline{V}_{i,h}^{\pi}\right] \leq 3c_{\mathsf{b}}\sqrt{\frac{H^3\log^3(\frac{KS\sum_{i=1}^n A_i}{\delta})}{K}}1 + \mathcal{D}_1 + \mathcal{D}_2 + \mathcal{D}_3$$

$$\leq c_{\mathsf{b}}\sqrt{\frac{H\log^3(\frac{KS\sum_{i=1}^n A_i}{\delta})}{K}}\left(3H + 15\min\left\{\frac{1}{\sigma_i}, H\right\}\right)1$$

$$+ 3c_{\mathsf{b}}\sqrt{\frac{H^3\log^3(\frac{KS\sum_{i=1}^n A_i}{\delta})}{K}}\max_{h\leq j\leq H}\left\|\widehat{V}_{i,j+1} - \mathbb{E}_{\pi\sim\widehat{\xi}}\left[\overline{V}_{i,j+1}^{\pi}\right]\right\|_{\infty}1.$$

Moreover, Lemma 5 implies that $\widehat{V}_{i,h} - \mathbb{E}_{\pi\sim\widehat{\xi}}\left[\overline{V}_{i,h}^{\pi}\right] = \left|\widehat{V}_{i,h} - \mathbb{E}_{\pi\sim\widehat{\xi}}\left[\overline{V}_{i,h}^{\pi}\right]\right|$, which indicates that

$$\max_{h\in[H]}\left\|\widehat{V}_{i,h} - -\mathbb{E}_{\pi\sim\widehat{\xi}}\left[\overline{V}_{i,h}^{\pi}\right]\right\|_{\infty}$$

$$\leq c_{\mathsf{b}}\sqrt{\frac{H\log^3(\frac{KS\sum_{i=1}^n A_i}{\delta})}{K}}\left(3H + 15\min\left\{\frac{1}{\sigma_i}, H\right\}\right)$$

$$+ 3c_{\mathsf{b}} \sqrt{\frac{H^3 \log^3(\frac{KS \sum_{i=1}^n A_i}{\delta})}{K}} \max_{h \leq j \leq H} \left\| \widehat{V}_{i,j+1} - \mathbb{E}_{\pi \sim \widehat{\xi}}\left[ \overline{V}_{i,j+1}^\pi \right] \right\|_\infty$$

$$\overset{(i)}{\leq} 18c_{\mathsf{b}} \sqrt{\frac{H^3 \log^3(\frac{KS \sum_{i=1}^n A_i}{\delta})}{K}} + \frac{1}{2} \max_{h \in [H]} \left\| \widehat{V}_{i,h} - \mathbb{E}_{\pi \sim \widehat{\xi}}\left[ \overline{V}_{i,h}^\pi \right] \right\|_\infty$$

$$\leq 36c_{\mathsf{b}} \sqrt{\frac{H^3 \log^3(\frac{KS \sum_{i=1}^n A_i}{\delta})}{K}}$$

where (i) holds by taking $K \geq 12c_{\mathsf{b}}^2 H^3 \log^3(\frac{KS \sum_{i=1}^n A_i}{\delta})$, and involving the basic facts that $\widehat{V}_{i,H+1} = \mathbb{E}_{\pi \sim \widehat{\xi}}\left[ \overline{V}_{i,H+1}^\pi \right] = 0$. Eventually, we can achieve the following upper bound of term $B$:

$$\mathbb{E}_{\pi \sim \widehat{\xi}}\left[ \overline{V}_{i,h}^{\star,\pi_{-i}} \right] - \mathbb{E}_{\pi \sim \widehat{\xi}}\left[ \overline{V}_{i,h}^\pi \right] \leq 36c_{\mathsf{b}} \sqrt{\frac{H^3 \log^3(\frac{KS \sum_{i=1}^n A_i}{\delta})}{K}} 1. \tag{86}$$

### D.3 CONTROLLING TERMS A AND C

In this section, we derive an upper bound for the difference between the true value function and the estimated value function. We consider a more general case involving a given set of policies $\left\{\widehat{\pi}_h^k\right\}_{(h,k) \in [H] \times [K]}$, where either $\widehat{\pi}_h^k = \pi_h^k$ for all $(h,k) \in [H] \times [K]$, or $\widehat{\pi}_h^k = \tilde{\pi}_i^\star \times \pi_{-i,h}^k$ for all $(h,k) \in [H] \times [K]$. Additionally, we define a distribution over the set of policies $\zeta := \{\zeta_h\}_{h \in [H]}$, with $\zeta_h : [H] \mapsto \Delta(\mathcal{S} \mapsto \prod_{i \in [n]} \Delta(\mathcal{A}_i))$, where $\zeta_h\left(\widehat{\pi}_h^k\right) = \alpha_k^K$ for all $(h,k) \in [H] \times [K]$. Our objective is to derive an upper bound for $\left| \mathbb{E}_{\pi \sim \zeta}\left[ V_{i,h}^\pi(s) \right] - \mathbb{E}_{\pi \sim \zeta}\left[ \overline{V}_{i,h}^\pi(s) \right] \right|$, where for all $s \in \mathcal{S}$, $\mathbb{E}_{\pi \sim \zeta}\left[ \overline{V}_{i,h}^\pi(s) \right]$ is defined as

$$\mathbb{E}_{\pi \sim \zeta}\left[ \overline{V}_{i,h}^\pi(s) \right]$$

$$= \sum_{k=1}^K \alpha_k^K \mathbb{E}_{a_i \sim \widehat{\pi}_{i,h}^k(s)}[r_{i,h}^k(s,a_i)] + \sum_{k=1}^K \alpha_k^K \mathbb{E}_{a_i \sim \widehat{\pi}_{i,h}^k(s)} \left[ \inf_{\mathcal{P} \in \mathcal{U}^{\sigma_i}\left(\widehat{P}_{i,h,s,a_i}^{\widehat{\pi}_{-i}^k}\right)} \mathcal{P} \mathbb{E}_{\pi \sim \zeta}\left[ \overline{V}_{i,h+1}^\pi \right] \right],$$

with $\mathbb{E}_{\pi \sim \zeta}\left[ \overline{V}_{i,H+1}^\pi(s) \right] = 0$. Here, $r_{i,h}^k(s,a_i)$ represents the empirical estimation of $r_{i,h}^{\widehat{\pi}_{-i}^k}(s,a_i)$, and $\widehat{P}_{i,h,s,a_i}^{\widehat{\pi}_{-i}^k}$ denotes the empirical estimation of $P_{h,s,a_i}^{\widehat{\pi}_{-i}^k}$ for all $(h,s,a_i,k) \in [H] \times \mathcal{S} \times \mathcal{A}_i \times [K]$. For notational clarity, we define the empirical reward vector $\overline{r}_{i,h}^{\widehat{\pi}^k} \in \mathbb{R}^S$, such that $\overline{r}_{i,h}^{\widehat{\pi}^k}(s) = \mathbb{E}_{a_i \sim \widehat{\pi}_{i,h}^k(s)}[r_{i,h}^k(s,a_i)]$ for all $s \in \mathcal{S}$.

We first introduce the following two lemmas in terms of estimation error of transition model and reward function:

**Lemma 10.** *Let $\delta \in (0,1)$ and consider any $(h,i,k) \in [H] \times [n] \times [K]$. With a probability of at least $1 - \delta$, for any fixed value vector $V \in \mathbb{R}^S$, where $0 \leq V(s) \leq H$ for all $s \in \mathcal{S}$, the following inequality holds:*

$$\left| P_{i,h}^{\widehat{\pi}_{-i}^k, V} V - \widehat{P}_{i,h}^{\widehat{\pi}_{-i}^k, V} V \right|$$

$$\leq 2 \sum_{k=1}^K \alpha_k^K \sqrt{\frac{\log\left(\frac{18S \sum_{i=1}^n A_i NHK}{\delta}\right)}{N}} \sqrt{\mathsf{Var}_{P_h^{\widehat{\pi}_{-i}^k}}(V)} + \frac{\log\left(\frac{18S \sum_{i=1}^n A_i NHK}{\delta}\right)}{N} 1$$

$$\leq 3 \sqrt{\frac{H^2 \log\left(\frac{18S \sum_{i=1}^n A_i NKH}{\delta}\right)}{N}} 1,$$

*where $\mathsf{Var}_{P_h^{\widehat{\pi}_{-i}^k}}(\cdot)$ is as defined in (25).*

*Proof.* See Appendix D.4.7. □

**Lemma 11.** *There exists a constant $c_r$ such that for any fixed pair $(h, i) \in [H] \times [n]$, with probability at least $1 - \delta$, the following inequality holds:*

$$\left| \sum_{k=1}^{K} \alpha_k^K r_{i,h}^{\widehat{\pi}^k} - \sum_{k=1}^{K} \alpha_k^K \overline{r}_{i,h}^{\widehat{\pi}^k} \right| \leq c_r \sqrt{\frac{\log\left(\frac{KS}{\delta}\right)}{K}} \mathbf{1}.$$

*Proof.* See Appendix D.4.8. □

For clarity of presentation, we extend the definitions in (23) and introduce additional notations related to transitions associated with the estimated value function. With a slight abuse of notation, we define the matrix notations $\widehat{P}_{i,h}^{\widehat{\pi}^k, \widehat{V}}$ and $\widehat{P}_{i,h}^{\widehat{\pi}^k, \overline{V}}$ as follows for all $(i, h, k) \in [n] \times [H] \times [K]$:

$$P_{i,h}^{\widehat{\pi}^k, \overline{V}} := P_{i,h}^{\widehat{\pi}_{-i}^k, \mathbb{E}_{\pi \sim \zeta}\left[\overline{V}_{i,h+1}^{\pi}\right]},$$

$$P_{i,h,s,a_i}^{\widehat{\pi}^k, \overline{V}} := P_{i,h,s,a_i}^{\widehat{\pi}_{-i}^k, \mathbb{E}_{\pi \sim \zeta}\left[\overline{V}_{i,h+1}^{\pi}\right]} = \operatorname{argmin}_{\mathcal{P} \in \mathcal{U}_{\rho}^{\sigma_i}\left(P_{h,s,a_i}^{\widehat{\pi}_{-i}^k}\right)} \mathcal{P} \mathbb{E}_{\pi \sim \zeta}\left[\overline{V}_{i,h+1}^{\pi}\right],$$

$$\widehat{P}_{i,h}^{\widehat{\pi}^k, \overline{V}} := \widehat{P}_{i,h}^{\widehat{\pi}_{-i}^k, \mathbb{E}_{\pi \sim \zeta}\left[\overline{V}_{i,h+1}^{\pi}\right]},$$

$$\widehat{P}_{i,h,s,a_i}^{\widehat{\pi}^k, \overline{V}} := \widehat{P}_{i,h,s,a_i}^{\widehat{\pi}_{-i}^k, \mathbb{E}_{\pi \sim \zeta}\left[\overline{V}_{i,h+1}^{\pi}\right]} = \operatorname{argmin}_{\mathcal{P} \in \mathcal{U}_{\rho}^{\sigma_i}\left(\widehat{P}_{i,h,s,a_i}^{\widehat{\pi}_{-i}^k}\right)} \mathcal{P} \mathbb{E}_{\pi \sim \zeta}\left[\overline{V}_{i,h+1}^{\pi}\right].$$

Additionally, we define the square matrices $\widehat{\underline{P}}_{i,h}^{\widehat{\pi}^k, \overline{V}} \in \mathbb{R}^{S \times S}$ and $\underline{P}_{i,h}^{\widehat{\pi}^k, \overline{V}} \in \mathbb{R}^{S \times S}$ as:

$$\widehat{\underline{P}}_{i,h}^{\widehat{\pi}^k, \overline{V}} := \Pi_h^{\widehat{\pi}_i^k} \widehat{P}_{i,h}^{\widehat{\pi}_{-i}^k, \overline{V}}, \quad \underline{P}_{i,h}^{\widehat{\pi}^k, \overline{V}} := \Pi_h^{\widehat{\pi}_i^k} P_{i,h}^{\widehat{\pi}_{-i}^k, \overline{V}}.$$

At any time step $h \in [H]$, we have

$$\mathbb{E}_{\pi \sim \zeta}\left[V_{i,h}^{\pi}\right] - \mathbb{E}_{\pi \sim \zeta}\left[\overline{V}_{i,h}^{\pi}\right]$$

$$\overset{(i)}{=} \sum_{k=1}^{K} \alpha_k^K r_{i,h}^{\widehat{\pi}^k} + \sum_{k=1}^{K} \alpha_k^K \Pi_h^{\widehat{\pi}_{i,h}^k} \left[ \inf_{\mathcal{P} \in \mathcal{U}^{\sigma_i}\left(P_{h,s,a_i}^{\widehat{\pi}_{-i}^k}\right)} \mathcal{P} \mathbb{E}_{\pi \sim \zeta}\left[V_{i,h+1}^{\pi}\right] \right]$$

$$- \sum_{k=1}^{K} \alpha_k^K \overline{r}_{i,h}^{\widehat{\pi}^k} - \sum_{k=1}^{K} \alpha_k^K \Pi_h^{\widehat{\pi}_{i,h}^k} \left[ \inf_{\mathcal{P} \in \mathcal{U}^{\sigma_i}\left(\widehat{P}_{h,s,a_i}^{\widehat{\pi}_{-i}^k}\right)} \mathcal{P} \mathbb{E}_{\pi \sim \zeta}\left[\overline{V}_{i,h+1}^{\pi}\right] \right]$$

$$\overset{(ii)}{=} \sum_{k=1}^{K} \alpha_k^K r_{i,h}^{\widehat{\pi}^k} + \sum_{k=1}^{K} \alpha_k^K \underline{P}_{i,h}^{\widehat{\pi}^k, V} \mathbb{E}_{\pi \sim \zeta}\left[V_{i,h+1}^{\pi}\right] - \sum_{k=1}^{K} \alpha_k^K \overline{r}_{i,h}^{\widehat{\pi}^k} - \sum_{k=1}^{K} \alpha_k^K \widehat{\underline{P}}_{i,h}^{\widehat{\pi}^k, \overline{V}} \mathbb{E}_{\pi \sim \zeta}\left[\overline{V}_{i,h+1}^{\pi}\right]$$

$$(87)$$

where (i) holds by the robust Bellman equation in (26) with matrix notation in (23), (ii) arises from the definition in (76). Moreover, through simple observation, we directly have $\underline{P}_{i,h}^{\widehat{\pi}^k, V} \mathbb{E}_{\pi \sim \zeta}\left[V_{i,h+1}^{\pi}\right] \leq \underline{P}_{i,h}^{\widehat{\pi}^k, \overline{V}} \mathbb{E}_{\pi \sim \zeta}\left[V_{i,h+1}^{\pi}\right]$ for all $(h, k) \in [H] \times [K]$. Thus, we further control (87) as

$$\mathbb{E}_{\pi \sim \zeta}\left[V_{i,h}^{\pi}\right] - \mathbb{E}_{\pi \sim \zeta}\left[\overline{V}_{i,h}^{\pi}\right]$$

$$= \sum_{k=1}^{K} \alpha_k^K r_{i,h}^{\widehat{\pi}^k} + \sum_{k=1}^{K} \alpha_k^K \underline{P}_{i,h}^{\widehat{\pi}^k, V} \mathbb{E}_{\pi \sim \zeta}\left[V_{i,h+1}^{\pi}\right] - \sum_{k=1}^{K} \alpha_k^K \overline{r}_{i,h}^{\widehat{\pi}^k} - \sum_{k=1}^{K} \alpha_k^K \widehat{\underline{P}}_{i,h}^{\widehat{\pi}^k, \overline{V}} \mathbb{E}_{\pi \sim \zeta}\left[\overline{V}_{i,h+1}^{\pi}\right]$$

$$= \sum_{k=1}^{K} \alpha_k^K \left[ \left( r_{i,h}^{\widehat{\pi}^k} - \overline{r}_{i,h}^{\widehat{\pi}^k} \right) + \left( \underline{P}_{i,h}^{\widehat{\pi}^k, V} \mathbb{E}_{\pi \sim \zeta}[V_{i,h+1}^{\pi}] - \underline{P}_{i,h}^{\widehat{\pi}^k, \overline{V}} \mathbb{E}_{\pi \sim \zeta}\left[\overline{V}_{i,h+1}^{\pi}\right] \right) \right]$$

$$+ \left( \underline{P}_{i,h}^{\widehat{\pi}^k, \overline{V}} \mathbb{E}_{\pi \sim \zeta} \left[ \overline{V}_{i,h+1}^{\pi} \right] - \underline{\widehat{P}}_{i,h}^{\widehat{\pi}^k, \overline{V}} \mathbb{E}_{\pi \sim \zeta} \left[ \overline{V}_{i,h+1}^{\pi} \right] \right) \Big]$$

$$\leq \sum_{k=1}^{K} \alpha_k^K \left( \underline{P}_{i,h}^{\widehat{\pi}^k, \overline{V}} \mathbb{E}_{\pi \sim \zeta} \left[ V_{i,h+1}^{\pi} \right] - \underline{P}_{i,h}^{\widehat{\pi}^k, \overline{V}} \mathbb{E}_{\pi \sim \zeta} \left[ \overline{V}_{i,h+1}^{\pi} \right] \right)$$

$$\underbrace{+ \sum_{k=1}^{K} \alpha_k^K \left[ \left| r_{i,h}^{\widehat{\pi}^k} - \overline{r}_{i,h}^{\widehat{\pi}^k} \right| + \left| \underline{P}_{i,h}^{\widehat{\pi}^k, \overline{V}} \mathbb{E}_{\pi \sim \zeta} \left[ \overline{V}_{i,h+1}^{\pi} \right] - \underline{\widehat{P}}_{i,h}^{\widehat{\pi}^k, \overline{V}} \mathbb{E}_{\pi \sim \zeta} \left[ \overline{V}_{i,h+1}^{\pi} \right] \right| \right]}_{:= a_{i,h}^{\zeta}}. \tag{88}$$

Applying (88) recursively leads to

$$\mathbb{E}_{\pi \sim \zeta} \left[ V_{i,h}^{\pi} \right] - \mathbb{E}_{\pi \sim \zeta} \left[ \overline{V}_{i,h}^{\pi} \right] \leq \sum_{j=h}^{H} \left[ \prod_{r=h}^{j-1} \left( \sum_{k=1}^{K} \alpha_k^K \underline{P}_{i,r}^{\widehat{\pi}^k, \overline{V}} \right) \right] a_{i,j}^{\zeta}, \tag{89}$$

where the inequality holds by adopting the following notations:

$$\left[ \prod_{r=h}^{h-1} \left( \sum_{k=1}^{K} \alpha_k^K \underline{P}_{i,r}^{\widehat{\pi}^k, \overline{V}} \right) \right] = I,$$

$$\left[ \prod_{r=h}^{j-1} \left( \sum_{k=1}^{K} \alpha_k^K \underline{P}_{i,r}^{\widehat{\pi}^k, \overline{V}} \right) \right] = \left( \sum_{k=1}^{K} \alpha_k^K \underline{P}_{i,h}^{\widehat{\pi}^k, \overline{V}} \right) \cdot \left( \sum_{k=1}^{K} \alpha_k^K \underline{P}_{i,h+1}^{\widehat{\pi}^k, \overline{V}} \right) \cdots \left( \sum_{k=1}^{K} \alpha_k^K \underline{P}_{i,j-1}^{\widehat{\pi}^k, \overline{V}} \right).$$

Next, similar to (88), we can achieve that

$$\mathbb{E}_{\pi \sim \zeta} \left[ \overline{V}_{i,h}^{\pi} \right] - \mathbb{E}_{\pi \sim \zeta} \left[ V_{i,h}^{\pi} \right]$$

$$\stackrel{\text{(i)}}{=} \sum_{k=1}^{K} \alpha_k^K \overline{r}_{i,h}^{\widehat{\pi}^k} + \sum_{k=1}^{K} \alpha_k^K \underline{\widehat{P}}_{i,h}^{\widehat{\pi}^k, \overline{V}} \mathbb{E}_{\pi \sim \zeta} \left[ \overline{V}_{i,h+1}^{\pi} \right] - \sum_{k=1}^{K} \alpha_k^K r_{i,h}^{\widehat{\pi}^k} - \sum_{k=1}^{K} \alpha_k^K \underline{P}_{i,h}^{\widehat{\pi}^k, V} \mathbb{E}_{\pi \sim \zeta} \left[ V_{i,h+1}^{\pi} \right]$$

$$= \sum_{k=1}^{K} \alpha_k^K \left[ \left( \overline{r}_{i,h}^{\widehat{\pi}^k} - r_{i,h}^{\widehat{\pi}^k} \right) + \left( \underline{\widehat{P}}_{i,h}^{\widehat{\pi}^k, \overline{V}} \mathbb{E}_{\pi \sim \zeta} \left[ \overline{V}_{i,h+1}^{\pi} \right] - \underline{P}_{i,h}^{\widehat{\pi}^k, \overline{V}} \mathbb{E}_{\pi \sim \zeta} \left[ \overline{V}_{i,h+1}^{\pi} \right] \right) \right.$$

$$\left. + \left( \underline{P}_{i,h}^{\widehat{\pi}^k, \overline{V}} \mathbb{E}_{\pi \sim \zeta} \left[ \overline{V}_{i,h+1}^{\pi} \right] - \underline{P}_{i,h}^{\widehat{\pi}^k, V} \mathbb{E}_{\pi \sim \zeta} \left[ V_{i,h+1}^{\pi} \right] \right) \right]$$

$$\stackrel{\text{(ii)}}{\leq} \sum_{k=1}^{K} \alpha_k^K \left( \underline{P}_{i,h}^{\widehat{\pi}^k, V} \mathbb{E}_{\pi \sim \zeta} \left[ \overline{V}_{i,h+1}^{\pi} \right] - \underline{P}_{i,h}^{\widehat{\pi}^k, V} \mathbb{E}_{\pi \sim \zeta} \left[ V_{i,h+1}^{\pi} \right] \right)$$

$$+ \sum_{k=1}^{K} \alpha_k^K \left[ \left| r_{i,h}^{\widehat{\pi}^k} - \overline{r}_{i,h}^{\widehat{\pi}^k} \right| + \left| \underline{P}_{i,h}^{\widehat{\pi}^k, \overline{V}} \mathbb{E}_{\pi \sim \zeta} \left[ \overline{V}_{i,h+1}^{\pi} \right] - \underline{\widehat{P}}_{i,h}^{\widehat{\pi}^k, \overline{V}} \mathbb{E}_{\pi \sim \zeta} \left[ \overline{V}_{i,h+1}^{\pi} \right] \right| \right],$$

where (i) holds due to robust Bellman equation, and (ii) holds due to the direct observation that $\underline{P}_{i,h}^{\widehat{\pi}^k, \overline{V}} \mathbb{E}_{\pi \sim \zeta} \left[ \overline{V}_{i,h+1}^{\pi} \right] \leq \underline{P}_{i,h}^{\widehat{\pi}^k, V} \mathbb{E}_{\pi \sim \zeta} \left[ \overline{V}_{i,h+1}^{\pi} \right]$. Then following the routine of achieving (89), we can obtain that

$$\mathbb{E}_{\pi \sim \zeta} \left[ \overline{V}_{i,h}^{\pi} \right] - \mathbb{E}_{\pi \sim \zeta} \left[ V_{i,h}^{\pi} \right] \leq \sum_{j=h}^{H} \left[ \prod_{r=h}^{j-1} \left( \sum_{k=1}^{K} \alpha_k^K \underline{P}_{i,r}^{\widehat{\pi}^k, V} \right) \right] a_{i,j}^{\zeta}. \tag{90}$$

Summing up (90) and (89), one has

$$\left| \mathbb{E}_{\pi \sim \zeta} \left[ \overline{V}_{i,h}^{\pi} \right] - \mathbb{E}_{\pi \sim \zeta} \left[ V_{i,h}^{\pi} \right] \right|$$

$$\leq \max \left\{ \mathbb{E}_{\pi \sim \zeta} \left[ V_{i,h}^{\pi} \right] - \mathbb{E}_{\pi \sim \zeta} \left[ \overline{V}_{i,h}^{\pi} \right], \mathbb{E}_{\pi \sim \zeta} \left[ \overline{V}_{i,h}^{\pi} \right] - \mathbb{E}_{\pi \sim \zeta} \left[ V_{i,h}^{\pi} \right] \right\}$$

$$\leq \max \left\{ \sum_{j=h}^{H} \left[ \prod_{r=h}^{j-1} \left( \sum_{k=1}^{K} \alpha_k^K \underline{P}_{i,r}^{\widehat{\pi}^k, \overline{V}} \right) \right] a_{i,j}^{\zeta}, \sum_{j=h}^{H} \left[ \prod_{r=h}^{j-1} \left( \sum_{k=1}^{K} \alpha_k^K \underline{P}_{i,r}^{\widehat{\pi}^k, V} \right) \right] a_{i,j}^{\zeta} \right\}, \tag{91}$$

where the max operator is taken entry-wise for vectors. To continue, we apply Lemma 10 and Lemma 11, and we can obtain the following upper bound on $a_{i,j}^\zeta$ for all $(i,j) \in [n] \times [H]$:

$$
\begin{aligned}
a_{i,h}^\zeta &= \sum_{k=1}^{K} \alpha_k^K \left[ \left| r_{i,h}^{\widehat{\pi}^k} - \overline{r}_{i,h}^{\widehat{\pi}^k} \right| + \left| \underline{P}_{i,h}^{\widehat{\pi}^k, \overline{V}} \mathbb{E}_{\pi \sim \zeta} \left[ \overline{V}_{i,h+1}^{\pi} \right] - \underline{\widehat{P}}_{i,h}^{\widehat{\pi}^k, \overline{V}} \mathbb{E}_{\pi \sim \zeta} \left[ \overline{V}_{i,h+1}^{\pi} \right] \right| \right] \\
&\leq 2 \sum_{k=1}^{K} \alpha_k^K \sqrt{\frac{\log \left( \frac{18S \sum_{i=1}^{n} A_i K N n H}{\delta} \right)}{N}} \sqrt{\mathsf{Var}_{\underline{P}_h^{\widehat{\pi}^k}} \left( \mathbb{E}_{\pi \sim \zeta} \left[ \overline{V}_{i,h+1}^{\pi} \right] \right)} \\
&\quad + \frac{\log \left( \frac{18S \sum_{i=1}^{n} A_i K N n H}{\delta} \right)}{N} 1 + c_r \sqrt{\frac{\log(\frac{KSnH}{\delta})}{K}} 1,
\end{aligned}
$$

holds with probability at least $1 - \delta$.

### D.3.1 CONTROLLING THE FIRST TERM IN (91)

To simplify notation, let us introduce some additional symbols. Recall that $e_s$ represents the standard basis vector in $S$-dimensional space associated with the $s$-th component. We define

$$
d_h^h = e_s \quad \text{and} \quad d_h^j = e_s^\top \left[ \prod_{r=h}^{j-1} \left( \sum_{k=1}^{K} \alpha_k^K \underline{P}_{i,r}^{\widehat{\pi}^k, \overline{V}} \right) \right] \quad \text{for } j = h+1, \ldots, H. \tag{92}
$$

With these notations in place, for any $s \in \mathcal{S}$, we consider

$$
\mathbb{E}_{\pi \sim \zeta} \left[ V_{i,h}^{\pi}(s) \right] - \mathbb{E}_{\pi \sim \zeta} \left[ \overline{V}_{i,h}^{\pi}(s) \right] = \left\langle e_s, \mathbb{E}_{\pi \sim \zeta} \left[ V_{i,h}^{\pi} \right] - \mathbb{E}_{\pi \sim \zeta} \left[ \overline{V}_{i,h}^{\pi} \right] \right\rangle = \sum_{j=h}^{H} \left\langle d_h^j, a_{i,j}^\zeta \right\rangle.
$$

Applying Lemma 10, we obtain

$$
\begin{aligned}
& \mathbb{E}_{\pi \sim \zeta} \left[ V_{i,h}^{\pi}(s) \right] - \mathbb{E}_{\pi \sim \zeta} \left[ \overline{V}_{i,h}^{\pi}(s) \right] \\
& \leq \sum_{j=h}^{H} \left\langle d_h^j, 2 \sum_{k=1}^{K} \alpha_k^K \sqrt{\frac{\log \left( \frac{18S \sum_{i=1}^{n} A_i K N H}{\delta} \right)}{N}} \sqrt{\mathsf{Var}_{\underline{P}_j^{\widehat{\pi}^k}} \left( \mathbb{E}_{\pi \sim \zeta} \left[ \overline{V}_{i,j+1}^{\pi} \right] \right)} \right\rangle \\
& \quad + \frac{\log \left( \frac{18S \sum_{i=1}^{n} A_i K N H}{\delta} \right)}{N} + c_r \sqrt{\frac{\log \left( \frac{KSnH}{\delta} \right)}{K}} \\
& \leq \frac{H \log \left( \frac{18S \sum_{i=1}^{n} A_i K N H}{\delta} \right)}{N} + c_r \sqrt{\frac{H^2 \log \left( \frac{KSnH}{\delta} \right)}{K}} \\
& \quad + \sum_{j=h}^{H} \left\langle d_h^j, 2 \sum_{k=1}^{K} \alpha_k^K \sqrt{\frac{\log \left( \frac{18S \sum_{i=1}^{n} A_i K N H}{\delta} \right)}{N}} \sqrt{\mathsf{Var}_{\underline{P}_j^{\widehat{\pi}^k}} \left( \mathbb{E}_{\pi \sim \zeta} \left[ \overline{V}_{i,j+1}^{\pi} \right] \right)} \right\rangle. \tag{93}
\end{aligned}
$$

By applying the triangle inequality, we can further decompose the term of interest as follows:

$$
\begin{aligned}
& \mathbb{E}_{\pi \sim \zeta} \left[ V_{i,h}^{\pi}(s) \right] - \mathbb{E}_{\pi \sim \zeta} \left[ \overline{V}_{i,h}^{\pi}(s) \right] \\
& \leq \frac{H \log \left( \frac{18S \sum_{i=1}^{n} A_i K N H}{\delta} \right)}{N} + c_r \sqrt{\frac{H^2 \log \left( \frac{KSnH}{\delta} \right)}{K}} + \mathcal{B}_1 + \mathcal{B}_2, \tag{94}
\end{aligned}
$$

where we define term $\mathcal{B}_1$ and $\mathcal{B}_2$ as:

$$
\mathcal{B}_1 = \sum_{j=h}^{H} \left\langle d_h^j, 2 \sum_{k=1}^{K} \alpha_k^K \sqrt{\frac{\log \left( \frac{18S \sum_{i=1}^{n} A_i K N H}{\delta} \right)}{N}} \sqrt{\mathsf{Var}_{\underline{P}_{i,j}^{\widehat{\pi}^k, \overline{V}}} \left( \mathbb{E}_{\pi \sim \zeta} \left[ \overline{V}_{i,j+1}^{\pi} \right] \right)} \right\rangle
$$

$$\mathcal{B}_2 = \sum_{j=h}^{H} 2 \sum_{k=1}^{K} \alpha_k^K \left\langle d_h^j, \sqrt{\left| \mathsf{Var}_{\underline{P}_{i,j}^{\widehat{\pi}^k}, \overline{V}} \left( \mathbb{E}_{\pi \sim \zeta} \left[ \overline{V}_{i,j+1}^{\pi} \right] \right) - \mathsf{Var}_{\underline{P}_{j}^{\widehat{\pi}^k}} \left( \mathbb{E}_{\pi \sim \zeta} \left[ \overline{V}_{i,j+1}^{\pi} \right] \right) \right|} \right\rangle$$

$$\cdot \sqrt{\frac{\log \left( 18S \sum_{i=1}^{n} A_i K N H / \delta \right)}{N}}$$

We then analyze the bounds for the terms $\mathcal{B}_1$ and $\mathcal{B}_2$ separately.

**Controlling $\mathcal{B}_1$.** First, we introduce the following lemma and corresponding inequality to establish control over the term $\sum_{k=1}^{K} \alpha_k^K \sqrt{\mathsf{Var}_{\underline{P}_{i,j}^{\widehat{\pi}^k}, \overline{V}} \left( \mathbb{E}_{\pi \sim \zeta} \left[ \overline{V}_{i,j+1}^{\pi} \right] \right)}$:

**Lemma 12.** *For any transition kernels $P_1, \ldots, P_m \in \mathbb{R}^S$, and any weight $a_1, \ldots, a_m \in [0, 1]$ with $a_1 + \ldots + a_m = 1$, one has*

$$\sum_{i=1}^{m} a_i \sqrt{\mathsf{Var}_{P_i}(V)} \le \sqrt{\mathsf{Var}_{\sum_{i=1}^{m} a_i P_i}(V)},$$

*where $V$ denote any fixed value vector $V \in \mathbb{R}^S$ with $0 \le V(s) \le H$ for all $s \in \mathcal{S}$.*

*Proof.* Initially, since $f(x) = \sqrt{x}$ is a concave function, we have

$$\sum_{i=1}^{m} a_i \sqrt{\mathsf{Var}_{P_i}(V)} = \sqrt{\sum_{i=1}^{m} a_i \mathsf{Var}_{P_i}(V)}.$$

Moreover, according to the definition of variance in (24), we obtain that

$$\sum_{i=1}^{m} a_i \mathsf{Var}_{P_i}(V) = \sum_{i=1}^{m} a_i \left( \mathbb{E}_{P_i} \left( V \circ V \right) - \left( \mathbb{E}_{P_i} V \circ \mathbb{E}_{P_i} V \right) \right)$$

$$\le \sum_{i=1}^{m} a_i \mathbb{E}_{P_i} \left( V \circ V \right) - \left( \sum_{i=1}^{m} a_i \mathbb{E}_{P_i} V \right) \circ \left( \sum_{i=1}^{m} a_i \mathbb{E}_{P_i} V \right),$$

where the last inequality holds due to the elementary fact that $f(x) = x^2$ is a convex function. Therefore, we have proven the result of the lemma. $\square$

With Lemma 12, we can further control $\mathcal{B}_1$ with

$$\mathcal{B}_1 = \sum_{j=h}^{H} \left\langle d_h^j, 2 \sum_{k=1}^{K} \alpha_k^K \sqrt{\frac{\log \left( \frac{18S \sum_{i=1}^{n} A_i K N n H}{\delta} \right)}{N}} \sqrt{\mathsf{Var}_{\underline{P}_{i,j}^{\widehat{\pi}^k}, \overline{V}} \left( \mathbb{E}_{\pi \sim \zeta} \left[ \overline{V}_{i,j+1}^{\pi} \right] \right)} \right\rangle$$

$$\le 2 \sqrt{\frac{\log \left( \frac{18S \sum_{i=1}^{n} A_i K N n H}{\delta} \right)}{N}} \sum_{j=h}^{H} \left\langle d_h^j, \sqrt{\mathsf{Var}_{\sum_{k=1}^{K} \alpha_k^K \underline{P}_{i,j}^{\widehat{\pi}^k}, \overline{V}} \left( \mathbb{E}_{\pi \sim \zeta} \left[ \overline{V}_{i,j+1}^{\pi} \right] \right)} \right\rangle$$

$$\le 2 \sqrt{\frac{\log \left( \frac{18S \sum_{i=1}^{n} A_i K N n H}{\delta} \right)}{N}} \sqrt{H \sum_{j=h}^{H} \left\langle d_h^j, \mathsf{Var}_{\sum_{k=1}^{K} \alpha_k^K \underline{P}_{i,j}^{\widehat{\pi}^k}, \overline{V}} \left( \mathbb{E}_{\pi \sim \zeta} \left[ \overline{V}_{i,j+1}^{\pi} \right] \right) \right\rangle}. \quad (95)$$

The last inequality holds due to Cauchy-Schwartz inequality. To further achieve the upper bound of $\mathcal{B}_1$, we introduce the following lemma of $\sum_{j=h}^{H} \left\langle d_h^j, \mathsf{Var}_{\sum_{k=1}^{K} \alpha_k^K \underline{P}_{i,j}^{\widehat{\pi}^k}, \overline{V}} \left( \mathbb{E}_{\pi \sim \zeta} \left[ \overline{V}_{i,j+1}^{\pi} \right] \right) \right\rangle$:

**Lemma 13.** *Consider any $\delta \in (0, 1)$. With probability at least $1 - \delta$, one has for all $(h, i) \in [H] \times [n]$:*

$$\sum_{j=h}^{H} \left\langle d_h^j, \mathsf{Var}_{\sum_{k=1}^{K} \alpha_k^K \underline{P}_{i,j}^{\widehat{\pi}^k}, \overline{V}} \left( \mathbb{E}_{\pi \sim \zeta} \left[ \overline{V}_{i,j+1}^{\pi} \right] \right) \right\rangle$$

$$\leq 3H \left( \max_{s \in \mathcal{S}} \mathbb{E}_{\pi \sim \zeta} \left[ \overline{V}_{i,h+1}^{\pi}(s) \right] - \min_{s \in \mathcal{S}} \mathbb{E}_{\pi \sim \zeta} \left[ \overline{V}_{i,h+1}^{\pi}(s) \right] \right) \left( 1 + 2H \sqrt{\frac{\log(\frac{18S \sum_{i=1}^{n} A_i nKNH}{\delta})}{N}} \right).$$

$$(96)$$

*Proof.* See Appendix D.4.9. $\qquad\square$

**Lemma 14.** *For all $(i,h) \in [n] \times [H]$, the estimated robust value function $\mathbb{E}_{\pi \sim \zeta} \left[ \overline{V}_{i,h}^{\pi} \right]$ satisfies the following inequality:*

$$\max_{s \in \mathcal{S}} \mathbb{E}_{\pi \sim \zeta} \left[ \overline{V}_{i,h}^{\pi}(s) \right] - \min_{s \in \mathcal{S}} \mathbb{E}_{\pi \sim \zeta} \left[ \overline{V}_{i,h}^{\pi}(s) \right] \leq \min \left\{ \frac{1}{\sigma_i}, H - h + 1 \right\}.$$

*Proof.* The proof of Lemma 14 closely parallels that of Lemma 7. Therefore, we omit the details here for brevity and clarity. $\qquad\square$

Apply Lemma 13 to (95), we arrive at

$$\mathcal{B}_1 \leq 2 \sqrt{\frac{\log \left( \frac{18S \sum_{i=1}^{n} A_i KNnH}{\delta} \right)}{N}} \sqrt{H \sum_{j=h}^{H} \left\langle d_h^j, \mathsf{Var}_{\sum_{k=1}^{K} \alpha_k^K \underline{P}_{i,j}^{\hat{\pi}^k}, \overline{V}} \left( \mathbb{E}_{\pi \sim \zeta} \left[ \overline{V}_{i,j+1}^{\pi} \right] \right) \right\rangle}$$

$$\leq \sqrt{3H^2 \left( \max_{s \in \mathcal{S}} \mathbb{E}_{\pi \sim \zeta} \left[ \overline{V}_{i,h+1}^{\pi}(s) \right] - \min_{s \in \mathcal{S}} \mathbb{E}_{\pi \sim \zeta} \left[ \overline{V}_{i,h+1}^{\pi}(s) \right] \right)}$$

$$\cdot 2 \sqrt{\frac{\log \left( \frac{18S \sum_{i=1}^{n} A_i KNnH}{\delta} \right)}{N} \cdot \left( 1 + 2H \sqrt{\frac{\log(\frac{18S \sum_{i=1}^{n} A_i nKNH}{\delta})}{N}} \right)}$$

$$\overset{(i)}{\leq} 2 \sqrt{3H^2 \min \left\{ \frac{1}{\sigma_i}, H - h + 1 \right\} \left( 1 + 2H \sqrt{\frac{\log(\frac{18S \sum_{i=1}^{n} A_i nKNH}{\delta})}{N}} \right)}$$

$$\cdot \sqrt{\frac{\log \left( \frac{18S \sum_{i=1}^{n} A_i nKNnH}{\delta} \right)}{N}}$$

$$\leq 6 \sqrt{\frac{H^2 \min\{1/\sigma_i, H\} \log \left( \frac{18S \sum_{i=1}^{n} A_i KNnH}{\delta} \right)}{N}},$$

$$(97)$$

where (i) holds by applying Lemma 7 and Lemma 14, and the final inequality follows by taking $N \geq 4H^2 \log \left( \frac{18S \sum_{i=1}^{n} A_i KnNH}{\delta} \right)$.

**Controlling $\mathcal{B}_2$.** Initially, with similar analysis as Lemma 8, we have the following lemma:

**Lemma 15.** *For transition kernel $P' \in \mathbb{R}^S$ and any $\widetilde{P} \in \mathbb{R}^S$ such that $\widetilde{P} \in \mathcal{U}^{\sigma_i}(P')$, the following bounds are established for all $(i,h) \in [n] \times [H]$:*

$$\left| \mathsf{Var}_{P'} \left( \mathbb{E}_{\pi \sim \zeta} \left[ \overline{V}_{i,h}^{\pi} \right] \right) - \mathsf{Var}_{\widetilde{P}} \left( \mathbb{E}_{\pi \sim \zeta} \left[ \overline{V}_{i,h}^{\pi} \right] \right) \right| \leq \min \left\{ \frac{1}{\sigma_i}, H - h + 1 \right\}.$$

With Lemma 15, we observe that

$$\left| \mathsf{Var}_{\underline{P}_j^{\hat{\pi}^k}} \left( \mathbb{E}_{\pi \sim \zeta} \left[ \overline{V}_{i,j+1}^{\pi} \right] \right) - \mathsf{Var}_{\underline{P}_{i,j}^{\hat{\pi}^k}, \overline{V}} \left( \mathbb{E}_{\pi \sim \zeta} \left[ \overline{V}_{i,j+1}^{\pi} \right] \right) \right|$$

$$\overset{(i)}{=} \left| \Pi_j^{\pi_i} \left( \mathsf{Var}_{P_j^{\hat{\pi}^k_{-i}}} \left( \mathbb{E}_{\pi \sim \zeta} \left[ \overline{V}_{i,j+1}^{\pi} \right] \right) - \mathsf{Var}_{P_{i,j}^{\hat{\pi}^k}, \overline{V}} \left( \mathbb{E}_{\pi \sim \zeta} \left[ \overline{V}_{i,j+1}^{\pi} \right] \right) \right) \right|$$

$$\overset{(ii)}{\leq} \left\| \mathsf{Var}_{P_j^{\widehat{\pi}^k_{-i}}} \left( \mathbb{E}_{\pi \sim \zeta} \left[ \overline{V}_{i,j+1}^{\pi} \right] \right) - \mathsf{Var}_{P_{i,j}^{\widehat{\pi}^k}, \overline{V}} \left( \mathbb{E}_{\pi \sim \zeta} \left[ \overline{V}_{i,j+1}^{\pi} \right] \right) \right\|_{\infty} 1$$

$$\leq \min \left\{ \frac{1}{\sigma_i}, H - h + 1 \right\} 1, \tag{98}$$

where (i) and (ii) follows from the definition of matrix notations $\Pi_j^{\pi}$ (cf B.1) and $\underline{P}_j^{\widehat{\pi}^k}, \underline{P}_{i,j}^{\widehat{\pi}^k, \overline{V}}$ (cf B.1), and the last inequality holds by applying Lemma 15 with $P' = P_{j,s,a_i}^{\pi^k_{-i}}, \widetilde{P} = P_{i,j,s,a_i}^{\widehat{\pi}^k, \overline{V}}$ for all $(s, a_i) \in \mathcal{S} \times \mathcal{A}_i$.

Plugging back (98) to (94), it can be verified that

$$\mathcal{B}_2 = \sum_{j=h}^{H} 2 \sum_{k=1}^{K} \alpha_k^K \left\langle d_h^j, \sqrt{\left| \mathsf{Var}_{\underline{P}_{i,j}^{\widehat{\pi}^k}, \overline{V}} \left( \mathbb{E}_{\pi \sim \zeta} \left[ \overline{V}_{i,j+1}^{\pi} \right] \right) - \mathsf{Var}_{\underline{P}_j^{\widehat{\pi}^k}} \left( \mathbb{E}_{\pi \sim \zeta} \left[ \overline{V}_{i,j+1}^{\pi} \right] \right) \right|} \right\rangle$$

$$\cdot \sqrt{\frac{\log \left( \frac{18 S \sum_{i=1}^{n} A_i K N n H}{\delta} \right)}{N}}$$

$$\leq \sum_{j=h}^{H} 2 \sum_{k=1}^{K} \alpha_k^K \sqrt{\frac{\log \left( \frac{18 S \sum_{i=1}^{n} A_i K N n H}{\delta} \right)}{N}} \left\langle d_h^j, \sqrt{\min \left\{ \frac{1}{\sigma_i}, H \right\} 1} \right\rangle$$

$$\leq 2 \sqrt{\frac{H^2 \min \left\{ \frac{1}{\sigma_i}, H \right\} \log \left( \frac{18 S \sum_{i=1}^{n} A_i K N n H}{\delta} \right)}{N}} \tag{99}$$

Consequently, combining (97) and (99), (94) can be bounded by

$$\mathbb{E}_{\pi \sim \zeta} \left[ V_{i,h}^{\pi}(s) \right] - \mathbb{E}_{\pi \sim \zeta} \left[ \overline{V}_{i,h}^{\pi}(s) \right]$$

$$\leq \frac{H \log \left( \frac{18 S \sum_{i=1}^{n} A_i K N n H}{\delta} \right)}{N} + c_r \sqrt{\frac{H^2 \log(\frac{K S n H}{\delta})}{K}} + \mathcal{B}_1 + \mathcal{B}_2$$

$$\leq \frac{H \log \left( \frac{18 S \sum_{i=1}^{n} A_i K N n H}{\delta} \right)}{N} + c_r \sqrt{\frac{H^2 \log(\frac{K S n H}{\delta})}{K}} \tag{100}$$

$$+ 8 \sqrt{\frac{H^2 \min \left\{ \frac{1}{\sigma_i}, H \right\} \log \left( \frac{18 S \sum_{i=1}^{n} A_i K N n H}{\delta} \right)}{N}}$$

$$\leq c_r \sqrt{\frac{H^2 \log(\frac{K S n H}{\delta})}{K}} + 12 \sqrt{\frac{H^2 \min \left\{ \frac{1}{\sigma_i}, H \right\} \log \left( \frac{18 S \sum_{i=1}^{n} A_i K N n H}{\delta} \right)}{N}}, \tag{101}$$

where the last inequality holds by taking $N \geq 4 H^2 \log \left( \frac{18 S \sum_{i=1}^{n} A_i K N n H}{\delta} \right)$.

### D.3.2 CONTROLLING THE SECOND TERM IN (91)

To do so, similar to (92), we define

$$w_h^h = e_s \quad \text{and} \quad w_h^j = e_s^{\top} \left[ \prod_{r=h}^{j-1} \left( \sum_{k=1}^{K} \alpha_k^K \underline{P}_{i,r}^{\widehat{\pi}^k, V} \right) \right] \quad \forall j = h+1, \cdots, H. \tag{102}$$

With the above notations in mind, following the routine of (93) gives: for any $s \in \mathcal{S}$,

$$\mathbb{E}_{\pi \sim \zeta} \left[ V_{i,h}^{\pi}(s) \right] - \mathbb{E}_{\pi \sim \zeta} \left[ \overline{V}_{i,h}^{\pi}(s) \right]$$

$$\leq \sum_{j=h}^{H} \left\langle w_h^j, 2 \sum_{k=1}^{K} \alpha_k^K \sqrt{\frac{\log \left( \frac{18 S \sum_{i=1}^{n} A_i K N n H}{\delta} \right)}{N}} \sqrt{\mathsf{Var}_{\underline{P}_j^{\widehat{\pi}^k}} \left( \mathbb{E}_{\pi \sim \zeta} \left[ \overline{V}_{i,j+1}^{\pi} \right] \right)} \right.$$

$$+ \frac{\log\left(\frac{18S\sum_{i=1}^n A_i KNnH}{\delta}\right)}{N}1 + c_r\sqrt{\frac{\log(\frac{KSnH}{\delta})}{K}}1 \Bigg\rangle$$

$$\leq \frac{H\log\left(\frac{18S\sum_{i=1}^n A_i KNnH}{\delta}\right)}{N} + c_r\sqrt{\frac{H^2\log(\frac{KSnH}{\delta})}{K}}$$

$$+ \sum_{j=h}^H \left\langle w_h^j, 2\sum_{k=1}^K \alpha_k^K\sqrt{\frac{\log\left(\frac{18S\sum_{i=1}^n A_i KNnH}{\delta}\right)}{N}}\sqrt{\mathsf{Var}_{\underline{P}_j^{\hat\pi^k}}\left(\mathbb{E}_{\pi\sim\zeta}\left[\overline{V}_{i,j+1}^\pi\right]\right)}\right\rangle. \quad (103)$$

Furthermore, following the routine established in (94), we can decompose the expression as follows:

$$\mathbb{E}_{\pi\sim\zeta}\left[V_{i,h}^\pi(s)\right] - \mathbb{E}_{\pi\sim\zeta}\left[\overline{V}_{i,h}^\pi(s)\right]$$

$$\leq \frac{H\log\left(\frac{18S\sum_{i=1}^n A_i KNnH}{\delta}\right)}{N} + c_r\sqrt{\frac{H^2\log(\frac{KSnH}{\delta})}{K}}$$

$$+ \sum_{j=h}^H \left\langle w_h^j, 2\sum_{k=1}^K \alpha_k^K\sqrt{\frac{\log\left(\frac{18S\sum_{i=1}^n A_i KNnH}{\delta}\right)}{N}}\sqrt{\mathsf{Var}_{\underline{P}_j^{\hat\pi^k}}\left(\mathbb{E}_{\pi\sim\zeta}\left[\overline{V}_{i,j+1}^\pi\right]\right)}\right\rangle$$

$$\overset{(i)}{\leq} \frac{H\log\left(\frac{18S\sum_{i=1}^n A_i KNnH}{\delta}\right)}{N} + c_r\sqrt{\frac{H^2\log(\frac{KSnH}{\delta})}{K}}$$

$$+ \mathcal{B}_3 + \sqrt{\frac{\log\left(\frac{18S\sum_{i=1}^n A_i KNnH}{\delta}\right)}{N}}\cdot\mathcal{B}_4 + \sqrt{\frac{\log\left(\frac{18S\sum_{i=1}^n A_i KNnH}{\delta}\right)}{N}}\cdot\mathcal{B}_5, \quad (104)$$

where (i) holds due to the triangle inequality and the fundamental inequality $\sqrt{\mathsf{Var}_P(V+V')} \leq \sqrt{\mathsf{Var}_P(V)} + \sqrt{\mathsf{Var}_P(V')}$ for any transition kernel $P \in \mathbb{R}^S$ and vectors $V, V' \in \mathbb{R}^S$, and we define the three terms $\mathcal{B}_3$, $\mathcal{B}_4$ and $\mathcal{B}_5$ as

$$\mathcal{B}_3 = \sum_{j=h}^H \left\langle w_h^j, 2\sum_{k=1}^K \alpha_k^K\sqrt{\frac{\log\left(\frac{18S\sum_{i=1}^n A_i KNnH}{\delta}\right)}{N}}\sqrt{\mathsf{Var}_{\underline{P}_{i,j}^{\hat\pi^k,V}}\left(\mathbb{E}_{\pi\sim\zeta}\left[V_{i,j+1}^\pi\right]\right)}\right\rangle$$

$$\mathcal{B}_4 = \sum_{j=h}^H \left\langle w_h^j, 2\sum_{k=1}^K \alpha_k^K\sqrt{\left|\mathsf{Var}_{\underline{P}_j^{\hat\pi^k}}\left(\mathbb{E}_{\pi\sim\zeta}\left[V_{i,j+1}^\pi\right]\right) - \mathsf{Var}_{\underline{P}_{i,j}^{\hat\pi^k,V}}\left(\mathbb{E}_{\pi\sim\zeta}\left[V_{i,j+1}^\pi\right]\right)\right|}\right\rangle$$

$$\mathcal{B}_5 = \sum_{j=h}^H \left\langle w_h^j, 2\sum_{k=1}^K \alpha_k^K\sqrt{\mathsf{Var}_{\underline{P}_j^{\hat\pi^k}}\left(\mathbb{E}_{\pi\sim\zeta}\left[\overline{V}_{i,j+1}^\pi\right] - \mathbb{E}_{\pi\sim\zeta}\left[V_{i,j+1}^\pi\right]\right)}\right\rangle.$$

Next, we will control the three main terms $\mathcal{B}_3, \mathcal{B}_4, \mathcal{B}_5$ in (104) separately as outlined below:

**Controlling $\mathcal{B}_3$.** Initially, we apply Lemma 12, and we can obtain the following upper bound of $\mathcal{B}_3$:

$$\mathcal{B}_3 = \sum_{j=h}^H \left\langle w_h^j, 2\sum_{k=1}^K \alpha_k^K\sqrt{\frac{\log\left(\frac{18S\sum_{i=1}^n A_i KNnH}{\delta}\right)}{N}}\sqrt{\mathsf{Var}_{\underline{P}_{i,j}^{\hat\pi^k,V}}\left(\mathbb{E}_{\pi\sim\zeta}\left[V_{i,j+1}^\pi\right]\right)}\right\rangle$$

$$\leq 2\sqrt{\frac{\log\left(\frac{18S\sum_{i=1}^n A_i KNnH}{\delta}\right)}{N}}\sum_{j=h}^H \left\langle w_h^j, \sqrt{\mathsf{Var}_{\sum_{k=1}^K \alpha_k^K \underline{P}_{i,j}^{\hat\pi^k,V}}\left(\mathbb{E}_{\pi\sim\zeta}\left[V_{i,j+1}^\pi\right]\right)}\right\rangle.$$

We further apply Cauchy-Schwartz inequality, and we can obtain that

$$\mathcal{B}_3 \leq 2\sqrt{\frac{\log\left(\frac{18S\sum_{i=1}^n A_i KNnH}{\delta}\right)}{N}}\sum_{j=h}^H \left\langle w_h^j, \sqrt{\mathsf{Var}_{\sum_{k=1}^K \alpha_k^K \underline{P}_{i,j}^{\hat\pi^k,V}}\left(\mathbb{E}_{\pi\sim\zeta}\left[V_{i,j+1}^\pi\right]\right)}\right\rangle$$

$$\leq 2\sqrt{\frac{\log\left(\frac{18S\sum_{i=1}^{n}A_iKNnH}{\delta}\right)}{N}}\sqrt{H\sum_{j=h}^{H}\left\langle w_h^j,\mathsf{Var}_{\sum_{k=1}^{K}\alpha_k^K\underline{P}_{i,j}^{\widehat{\pi}^k},V}\left(\mathbb{E}_{\pi\sim\zeta}\left[V_{i,j+1}^{\pi}\right]\right)\right\rangle}$$

In addition, to further bound the term of interest, we introduce the following lemma and inequalities for $\sum_{j=h}^{H}\left\langle w_h^j,\mathsf{Var}_{\sum_{k=1}^{K}\alpha_k^K\underline{P}_{i,j}^{\widehat{\pi}^k},V}\left(\mathbb{E}_{\pi\sim\zeta}\left[V_{i,j+1}^{\pi}\right]\right)\right\rangle$.

**Lemma 16.** *For any joint policy $\pi$, we have for all $(h,i)\in[H]\times[n]$:*

$$\sum_{j=h}^{H}\left\langle w_h^j,\mathsf{Var}_{\sum_{k=1}^{K}\alpha_k^K\underline{P}_{i,j}^{\widehat{\pi}^k},V}\left(\mathbb{E}_{\pi\sim\zeta}\left[V_{i,j+1}^{\pi}\right]\right)\right\rangle$$

$$\leq 3H\left(\max_{s\in\mathcal{S}}\mathbb{E}_{\pi\sim\zeta}\left[V_{i,h}^{\pi}(s)\right]-\min_{s\in\mathcal{S}}\mathbb{E}_{\pi\sim\zeta}\left[V_{i,h}^{\pi}(s)\right]\right). \tag{105}$$

*Proof.* See Appendix D.4.10. $\qquad\square$

**Lemma 17.** *For all $(i,h)\in[n]\times[H]$, the estimated robust value function $\mathbb{E}_{\pi\sim\zeta}\left[V_{i,h}^{\pi}\right]$ satisfies the following inequality:*

$$\max_{s\in\mathcal{S}}\mathbb{E}_{\pi\sim\zeta}\left[V_{i,h}^{\pi}(s)\right]-\min_{s\in\mathcal{S}}\mathbb{E}_{\pi\sim\zeta}\left[V_{i,h}^{\pi}(s)\right]\leq\min\left\{\frac{1}{\sigma_i},H-h+1\right\}.$$

*Proof.* The proof of Lemma 17 closely parallels that of Lemma 7. Therefore, we omit the details here for brevity and clarity. $\qquad\square$

Then applying Lemma 16 and Lemma 17 yields

$$\mathcal{B}_3\leq 2\sqrt{\frac{\log\left(\frac{18S\sum_{i=1}^{n}A_iKNnH}{\delta}\right)}{N}}\sqrt{H\sum_{j=h}^{H}\left\langle w_h^j,\mathsf{Var}_{\sum_{k=1}^{K}\alpha_k^K\underline{P}_{i,j}^{\widehat{\pi}^k},V}\left(\mathbb{E}_{\pi\sim\zeta}\left[V_{i,j+1}^{\pi}\right]\right)\right\rangle}$$

$$\leq 2\sqrt{\frac{\log\left(\frac{18S\sum_{i=1}^{n}A_iKNnH}{\delta}\right)}{N}}\sqrt{3H^2\left(\max_{s\in\mathcal{S}}\mathbb{E}_{\pi\sim\zeta}\left[V_{i,h}^{\pi,\sigma_i}(s)\right]-\min_{s\in\mathcal{S}}\mathbb{E}_{\pi\sim\zeta}\left[V_{i,h}^{\pi,\sigma_i}(s)\right]\right)}$$

$$\leq 4\sqrt{\frac{H^2\min\{1/\sigma_i,H\}\log\left(\frac{18S\sum_{i=1}^{n}A_iKNnH}{\delta}\right)}{N}}, \tag{106}$$

where the last inequality follows from Lemma 17.

**Controlling $\mathcal{B}_4$ and $\mathcal{B}_5$** With similar analysis as Lemma 8, we have the following lemma:

**Lemma 18.** *For any joint policy $\pi$, transition kernel $P'\in\mathbb{R}^S$, and any $\widetilde{P}\in\mathbb{R}^S$ such that $\widetilde{P}\in\mathcal{U}^{\sigma_i}\left(P'\right)$, the following bounds are established for all $(i,h)\in[n]\times[H]$:*

$$\left|\mathsf{Var}_{P'}\left(\mathbb{E}_{\pi\sim\zeta}\left[V_{i,h}^{\pi}\right]\right)-\mathsf{Var}_{\widetilde{P}}\left(\mathbb{E}_{\pi\sim\zeta}\left[V_{i,h}^{\pi}\right]\right)\right|\leq\min\left\{\frac{1}{\sigma_i},H-h+1\right\}.$$

We apply Lemma 18, and we can directly obtain the following upper bound of $\mathcal{B}_4$:

$$\mathcal{B}_4=\sum_{j=h}^{H}\left\langle w_h^j,2\sum_{k=1}^{K}\alpha_k^K\sqrt{\left|\mathsf{Var}_{\underline{P}_j^{\widehat{\pi}^k}}\left(\mathbb{E}_{\pi\sim\zeta}\left[V_{i,j+1}^{\pi}\right]\right)-\mathsf{Var}_{\underline{P}_{i,j}^{\widehat{\pi}^k},V}\left(\mathbb{E}_{\pi\sim\zeta}\left[V_{i,j+1}^{\pi}\right]\right)\right|}\right\rangle$$

$$\leq\sum_{j=h}^{H}\left\langle w_h^j,2\sum_{k=1}^{K}\alpha_k^K\sqrt{\min\left\{\frac{1}{\sigma_i},H\right\}}1\right\rangle$$

$$= s\sqrt{H^2 \min\{H, \frac{1}{\sigma_i}\}} \tag{107}$$

Then the remainder of the proof shall focus on $\mathcal{B}_5$. Recalling the definition in (104), one has

$$\mathcal{B}_5 = \sum_{j=h}^{H} \left\langle w_h^j, 2 \sum_{k=1}^{K} \alpha_k^K \sqrt{\mathsf{Var}_{\underline{P}_j^{\hat{\pi}^k}} \left( \mathbb{E}_{\pi \sim \zeta} \left[ \overline{V}_{i,j+1}^{\pi} \right] - \mathbb{E}_{\pi \sim \zeta} \left[ V_{i,j+1}^{\pi} \right] \right)} \right\rangle$$

$$\leq \sum_{j=h}^{H} \left\langle w_h^j, 2 \sum_{k=1}^{K} \alpha_k^K \sqrt{\left\| \mathsf{Var}_{\underline{P}_j^{\hat{\pi}^k}} \left( \mathbb{E}_{\pi \sim \zeta} \left[ \overline{V}_{i,j+1}^{\pi} \right] - \mathbb{E}_{\pi \sim \zeta} \left[ V_{i,j+1}^{\pi} \right] \right) \right\|_{\infty}} 1 \right\rangle$$

$$\leq 2 \sum_{j=h}^{H} \left\langle w_h^j, \left\| \mathbb{E}_{\pi \sim \zeta} \left[ \overline{V}_{i,j+1}^{\pi} \right] - \mathbb{E}_{\pi \sim \zeta} \left[ V_{i,j+1}^{\pi} \right] \right\|_{\infty} 1 \right\rangle$$

$$\leq 2 \max_{h \leq j \leq H} \left\| \mathbb{E}_{\pi \sim \zeta} \left[ \overline{V}_{i,j+1}^{\pi} \right] - \mathbb{E}_{\pi \sim \zeta} \left[ V_{i,j+1}^{\pi} \right] \right\|_{\infty}. \tag{108}$$

Summing up (106), (107), and (108) and inserting back to (104), we conclude

$$\mathbb{E}_{\pi \sim \zeta} \left[ V_{i,h}^{\pi}(s) \right] - \mathbb{E}_{\pi \sim \zeta} \left[ \overline{V}_{i,h}^{\pi}(s) \right]$$

$$\leq \frac{H \log \left( \frac{18S \sum_{i=1}^{n} A_i KNnH}{\delta} \right)}{N} + c_r \sqrt{\frac{H^2 \log \left( \frac{KSnH}{\delta} \right)}{K}}$$

$$+ \mathcal{B}_3 + \sqrt{\frac{\log \left( \frac{18S \sum_{i=1}^{n} A_i KNnH}{\delta} \right)}{N}} \cdot \mathcal{B}_4 + \sqrt{\frac{\log \left( \frac{18S \sum_{i=1}^{n} A_i KNnH}{\delta} \right)}{N}} \cdot \mathcal{B}_5$$

$$\leq \frac{H \log \left( \frac{18S \sum_{i=1}^{n} A_i KNnH}{\delta} \right)}{N} + c_r \sqrt{\frac{H^2 \log \left( \frac{KSnH}{\delta} \right)}{K}}$$

$$+ 6 \sqrt{\frac{H^2 \min\{1/\sigma_i, H\} \log \left( \frac{18S \sum_{i=1}^{n} A_i KNnH}{\delta} \right)}{N}}$$

$$+ 2 \sqrt{\frac{H^2 \log \left( \frac{18S \sum_{i=1}^{n} A_i KNnH}{\delta} \right)}{N}} \max_{h \leq j \leq H} \left\| \mathbb{E}_{\pi \sim \zeta} \left[ \overline{V}_{i,j+1}^{\pi} \right] - \mathbb{E}_{\pi \sim \zeta} \left[ V_{i,j+1}^{\pi} \right] \right\|_{\infty} \tag{109}$$

### D.3.3 SUMMING UP THE RESULTS: UPPER BOUND FOR TERM A AND C

Inserting (101) and (109) back into (91), we observe that

$$\left| \mathbb{E}_{\pi \sim \zeta} \left[ V_{i,h}^{\pi} \right] - \mathbb{E}_{\pi \sim \zeta} \left[ \overline{V}_{i,h}^{\pi} \right] \right|$$

$$\leq \max \left\{ \mathbb{E}_{\pi \sim \zeta} \left[ V_{i,h}^{\pi} \right] - \mathbb{E}_{\pi \sim \zeta} \left[ \overline{V}_{i,h}^{\pi} \right], \mathbb{E}_{\pi \sim \zeta} \left[ \overline{V}_{i,h}^{\pi} \right] - \mathbb{E}_{\pi \sim \zeta} \left[ V_{i,h}^{\pi} \right] \right\}$$

$$\leq \max \left\{ c_r \sqrt{\frac{H^2 \log \left( \frac{KSnH}{\delta} \right)}{K}} 1 + 12 \sqrt{\frac{H^2 \min \left\{ \frac{1}{\sigma_i}, H \right\} \log \left( \frac{18S \sum_{i=1}^{n} A_i KNnH}{\delta} \right)}{N}} 1 \right.$$

$$, c_r \sqrt{\frac{H^2 \log \left( \frac{KSnH}{\delta} \right)}{K}} 1 + \frac{H \log \left( \frac{18S \sum_{i=1}^{n} A_i KNnH}{\delta} \right)}{N} 1$$

$$+ 2 \sqrt{\frac{H^2 \log \left( \frac{18S \sum_{i=1}^{n} A_i KNnH}{\delta} \right)}{N}} \max_{h \leq j \leq H} \left\| \mathbb{E}_{\pi \sim \zeta} \left[ \overline{V}_{i,j+1}^{\pi} \right] - \mathbb{E}_{\pi \sim \zeta} \left[ V_{i,j+1}^{\pi} \right] \right\|_{\infty} 1$$

$$\left. + 6 \sqrt{\frac{H^2 \min\{ \frac{1}{\sigma_i}, H\} \log \left( \frac{18S \sum_{i=1}^{n} A_i KNnH}{\delta} \right)}{N}} 1 \right\},$$

which indicates that

$$\max_{h\in[H]} \left\| \mathbb{E}_{\pi\sim\zeta}\left[V_{i,h}^{\pi}\right] - \mathbb{E}_{\pi\sim\zeta}\left[\overline{V}_{i,h}^{\pi}\right] \right\|_{\infty}$$

$$\leq c_r\sqrt{\frac{H^2\log(\frac{KSnH}{\delta})}{K}} + 12\sqrt{\frac{H^2\min\left\{\frac{1}{\sigma_i},H\right\}\log\left(\frac{18S\sum_{i=1}^n A_i KNnH}{\delta}\right)}{N}}$$

$$+ \frac{H\log\left(\frac{18S\sum_{i=1}^n A_i KNnH}{\delta}\right)}{N}$$

$$+ 2\sqrt{\frac{H^2\log\left(\frac{18S\sum_{i=1}^n A_i KNnH}{\delta}\right)}{N}} \max_{h\in[H]}\left\| \mathbb{E}_{\pi\sim\zeta}\left[\overline{V}_{i,h+1}^{\pi}\right] - \mathbb{E}_{\pi\sim\zeta}\left[V_{i,h+1}^{\pi}\right] \right\|_{\infty}$$

$$\overset{(i)}{\leq} c_r\sqrt{\frac{H^2\log(\frac{KSnH}{\delta})}{K}} + 12\sqrt{\frac{H^2\min\left\{\frac{1}{\sigma_i},H\right\}\log\left(\frac{18S\sum_{i=1}^n A_i KNnH}{\delta}\right)}{N}}$$

$$+ \frac{H\log\left(\frac{18S\sum_{i=1}^n A_i KNnH}{\delta}\right)}{N} + \frac{1}{2}\max_{h\in[H]}\left\| \mathbb{E}_{\pi\sim\zeta}\left[\overline{V}_{i,h}^{\pi}\right] - \mathbb{E}_{\pi\sim\zeta}\left[V_{i,h}^{\pi}\right] \right\|_{\infty}$$

$$\leq 2c_r\sqrt{\frac{H^2\log(\frac{KSnH}{\delta})}{K}} + 24\sqrt{\frac{H^2\min\left\{\frac{1}{\sigma_i},H\right\}\log\left(\frac{18S\sum_{i=1}^n A_i KNnH}{\delta}\right)}{N}}$$

$$+ \frac{2H\log\left(\frac{18S\sum_{i=1}^n A_i KNnH}{\delta}\right)}{N} \tag{110}$$

where (i) holds when $N \geq 4H^2\log\left(\frac{18S\sum_{i=1}^n A_i NnH}{\delta}\right)$, and invoking the basic fact that $\mathbb{E}_{\pi\sim\zeta}\left[\overline{V}_{i,H+1}^{\pi}\right] = \mathbb{E}_{\pi\sim\zeta}\left[V_{i,H+1}^{\pi}\right] = 0$. With (110), we can achieve the following upper bound on term $A$ and term $C$:

$$\mathbb{E}_{\pi\sim\widehat{\xi}}\left[V_{i,h}^{\star,\pi_{-i}}\right] - \mathbb{E}_{\pi\sim\widehat{\xi}}\left[\overline{V}_{i,h}^{\widetilde{\pi}_i^{\star},\pi_{-i}}\right] \leq 24\sqrt{\frac{H^2\min\left\{\frac{1}{\sigma_i},H\right\}\log\left(\frac{18S\sum_{i=1}^n A_i KNnH}{\delta}\right)}{N}}1$$

$$+ 2c_r\sqrt{\frac{H^2\log(\frac{KSnH}{\delta})}{K}}1 + + \frac{2H\log\left(\frac{18S\sum_{i=1}^n A_i KNnH}{\delta}\right)}{N}1 \tag{111}$$

$$\mathbb{E}_{\pi\sim\widehat{\xi}}\left[\overline{V}_{i,h}^{\pi}\right] - \mathbb{E}_{\pi\sim\widehat{\xi}}\left[V_{i,h}^{\pi}\right] \leq 24\sqrt{\frac{H^2\min\left\{\frac{1}{\sigma_i},H\right\}\log\left(\frac{18S\sum_{i=1}^n A_i KNnH}{\delta}\right)}{N}}1$$

$$+ 2c_r\sqrt{\frac{H^2\log(\frac{KSnH}{\delta})}{K}}1 + + \frac{2H\log\left(\frac{18S\sum_{i=1}^n A_i KNnH}{\delta}\right)}{N}1 \tag{112}$$

### D.3.4 SUMMING UP THE RESULTS

Summing up the results in (111), (86), (112), we can achieve the upper bound of our target:

$$\mathbb{E}_{\pi\sim\widehat{\xi}}\left[V_{i,1}^{\star,\pi_{-i}}\right] - \mathbb{E}_{\pi\sim\widehat{\xi}}\left[V_{i,1}^{\pi}\right]$$

$$\leq 36c_{\mathsf{b}}\sqrt{\frac{H^3\log^3(\frac{KS\sum_{i=1}^n A_i}{\delta})}{K}}1 + 4c_r\sqrt{\frac{H^2\log(\frac{KSnH}{\delta})}{K}}1$$

$$+ \frac{4H\log\left(\frac{18S\sum_{i=1}^n A_i KNnH}{\delta}\right)}{N}1 + 48\sqrt{\frac{H^2\min\left\{\frac{1}{\sigma_i},H\right\}\log\left(\frac{18S\sum_{i=1}^n A_i KNnH}{\delta}\right)}{N}}1.$$

Therefore, there exists a constant $C$, such that when $N$ and $K$ satisfies:

$$N \geq CH^2 \min \left\{ \frac{1}{\min_{1 \leq i \leq n} \sigma_i}, H \right\} \log \left( \frac{18S \sum_{i=1}^n A_i KNnH}{\delta} \right) \frac{1}{\epsilon^2},$$

$$K \geq CH^3 \log^3 \left( \frac{KS \sum_{i=1}^n A_i H}{\delta} \right) \frac{1}{\epsilon^2}$$

we achieve $\max_{i \in [n]} \mathbb{E}_{\pi \sim \widehat{\xi}} \left[ V_{i,1}^{\star, \pi_{-i}} \right] - \mathbb{E}_{\pi \sim \widehat{\xi}} \left[ V_{i,1}^\pi \right] \leq \epsilon \cdot 1$ with probability at least $1 - \delta$. Therefore, the total number of samples we need is at least

$$N_{\mathsf{all}} = HS \sum_{i=1}^n KN = \tilde{\mathcal{O}} \left( \frac{S \max_{1 \leq i \leq n} A_i H^6}{\epsilon^4} \min \left\{ \frac{1}{\min_{1 \leq i \leq n} \sigma_i}, H \right\} \right).$$

Thus, we finish the proof of Theorem 2.

### D.4 PROOF OF AUXILIARY LEMMAS

#### D.4.1 PROOF OF LEMMA 4

Before proving Lemma 4, we first introduce the following lemma regarding the properties of the learning rate.

**Lemma 19** (Li et al. (2023, Lemma 1)). *For any $k \geq 1$, one has*

$$\alpha_1 = 1, \qquad \sum_{i=1}^k \alpha_i^k = 1, \qquad \max_{1 \leq i \leq k} \alpha_i^k \leq \frac{2c_\alpha \log K}{k}. \tag{113a}$$

*In addition, if $k \geq c_\alpha \log K + 1$ and $c_\alpha \geq 24$, then one has*

$$\max_{1 \leq i \leq k/2} \alpha_i^k \leq \frac{1}{K^6}. \tag{113b}$$

We will now prove the lemma with induction argument. Initially, the base step $H + 1$ trivially holds true, since we have

$$\widehat{V}_{i,H+1} = \mathbb{E}_{\pi \sim \widehat{\xi}} \left[ \overline{V}_{i,H+1}^{\star, \pi_{-i}} \right] = 0.$$

Next, we assume that the lemma holds for step $h + 1$, namely

$$\widehat{V}_{i,h+1} \geq \mathbb{E}_{\pi \sim \widehat{\xi}} \left[ \overline{V}_{i,h+1}^{\star, \pi_{-i}} \right]$$

and attempt to justify the validity of Lemma 4 for step $h$. Let $l_k$ denote $l_k = -q_{i,h}^k(s, \cdot), \forall k \geq 1$, then the update rule of Algorithm 2 can be viewed as the FTRL algorithm applied to the loss vectors $\{l_k\}_{k \in [K]}$. According to the definition of $\{\eta_k\}_{k \in [K]}$ and $\{\alpha_k\}_{k \in [K]}$, we have

$$\left( \frac{\eta_k}{\eta_{k+1}} \right)^2 = \frac{\alpha_k}{\alpha_{k-1}} = \frac{k - 2 + c_\alpha \log K}{k - 1 + c_\alpha \log K} \geq \frac{k - 1}{k - 1 + c_\alpha \log K} = 1 - \alpha_k > (1 - \alpha_k)^2. \tag{114}$$

This property (114) permits us to invoke Theorem 3 to obtain

$$\max_{a_i \in \mathcal{A}_i} \sum_{k=1}^K \alpha_k^K q_{i,h}^k(s, a_i) - \sum_{k=1}^K \alpha_k^K \left\langle \pi_{i,h}^k, q_{i,h}^k(s, \cdot) \right\rangle$$

$$= \max_{a_i \in \mathcal{A}_i} \left\{ \sum_{k=1}^K \alpha_k^K \left\langle \pi_{i,h}^k(s), l_k \right\rangle - \sum_{k=1}^K \alpha_k^K l_k(a_i) \right\}$$

$$\leq \frac{5}{3} \sum_{k=2}^K \alpha_k^K \frac{\eta_k \alpha_k}{1 - \alpha_k} \mathsf{Var}_{\pi_{i,h}^k(s)} \left( q_{i,h}^k(s, \cdot) \right) + \frac{\log A_i}{\eta_{K+1}} + \tau_{i,h}$$

where $\tau_{i,h}$ is defined as

$$\tau_{i,h}$$

$$:= \frac{5}{3}\alpha_1^K\eta_2\|q_{i,h}^1\|_\infty^2 + \left\{3\sum_{k=2}^K \alpha_k^K \frac{\eta_k^2\alpha_k^2}{(1-\alpha_k)^2}\|q_{i,h}^k\|_\infty^3 \mathbf{1}\left(\frac{\eta_k\alpha_k}{1-\alpha_k}\|q_{i,h}^k\|_\infty > \frac{1}{3}\right)\right\} + 3\alpha_1^K\eta_2^2\|q_{i,h}^1\|_\infty^3.$$

According to the definition of $\{\alpha_k\}_{k=1}^K$ and $\{\eta_k\}_{k=1}^K$, we have the following fact:

$$1 - \alpha_k = 1 - \frac{c_\alpha \log K}{k - 1 + c_\alpha \log K} \geq \begin{cases} 1 - \frac{c_\alpha \log K}{1 + c_\alpha \log K} = \frac{1}{1 + c_\alpha \log K} \geq \frac{1}{2c_\alpha \log K}, & \text{if } k \geq 2, \\ 1 - \frac{c_\alpha \log K}{K/2 + c_\alpha \log K} = \frac{K}{K + 2c_\alpha \log K} \geq \frac{1}{2}, & \text{if } k \geq K/2 + 1, \end{cases} \tag{115a}$$

$$\eta_k\alpha_k = \sqrt{\frac{\log K}{\alpha_{k-1}H}} \cdot \alpha_k \leq \sqrt{\frac{\log K}{\alpha_k H}} \cdot \alpha_k = \sqrt{\frac{\alpha_k \log K}{H}} \leq \sqrt{\frac{2c_\alpha \log^2 K}{kH}}. \tag{115b}$$

Therefore, we can re-control $\max_{a_i \in \mathcal{A}_i} \sum_{k=1}^K \alpha_k^K q_{i,h}^k(s, a_i) - \sum_{k=1}^K \alpha_k^K \langle \pi_{i,h}^k, q_{i,h}^k(s, \cdot)\rangle$ with

$$\max_{a_i \in \mathcal{A}_i} \sum_{k=1}^K \alpha_k^K q_{i,h}^k(s, a_i) - \sum_{k=1}^K \alpha_k^K \langle \pi_{i,h}^k, q_{i,h}^k(s, \cdot)\rangle \tag{116}$$

$$\leq \frac{5}{3}\sum_{k=2}^K \alpha_k^K \frac{\eta_k\alpha_k}{1 - \alpha_k} \mathsf{Var}_{\pi_{i,h}^k(s)}\left(q_{i,h}^k(s, \cdot)\right) + \frac{\log A_i}{\eta_{K+1}} + \tau_{i,h}$$

$$\overset{(i)}{\leq} \underbrace{\frac{5}{3}\sum_{k=2}^{K/2} \frac{(2c_\alpha)^{1.5}\log^2 K}{\sqrt{kH}}\alpha_k^K \mathsf{Var}_{\pi_{i,h}^k(s)}\left(q_{i,h}^k(s, \cdot)\right)}_{\mathcal{C}_1}$$

$$+ \frac{20}{3}\sum_{k=K/2+1}^K \alpha_k^K \sqrt{\frac{c_\alpha \log^2 K}{KH}} \mathsf{Var}_{\pi_{i,h}^k(s)}\left(q_{i,h}^k(s, \cdot)\right) + \underbrace{\frac{\log A_i}{\eta_{K+1}}}_{\mathcal{C}_2} + \underbrace{\tau_{i,h}}_{\mathcal{C}_3}, \tag{117}$$

Now we separately control the four terms $\mathcal{C}_1, \mathcal{C}_2, \mathcal{C}_3$ in (117).

- For term $\mathcal{C}_1$, we have

$$\sum_{k=2}^{K/2} \frac{\alpha_k^K \log^2 K}{\sqrt{kH}} \mathsf{Var}_{\pi_{i,h}^k(s)}\left(q_{i,h}^k(s, \cdot)\right) \leq \sum_{k=2}^{K/2} \frac{\log^2 K}{K^6\sqrt{kH}} \mathsf{Var}_{\pi_{i,h}^k(s)}\left(q_{i,h}^k(s, \cdot)\right)$$

$$\leq \sum_{k=2}^{K/2} \frac{\log^2 K}{K^6\sqrt{kH}}\|q_{i,h}^k(s, \cdot)\|_\infty^2 \leq \frac{H^{3/2}\log^2 K}{K^6}\sum_{k=2}^{K/2}\frac{1}{\sqrt{k}}$$

$$\leq \frac{2H^{3/2}\log^2 K}{K^6} \cdot \sqrt{K/2} \leq \frac{2H^{3/2}\log^2 K}{K^5}, \tag{118}$$

where the third inequality holds due to the elementary bound $\|q_{i,h}^k(s, \cdot)\|_\infty \leq H$.

- For term $\mathcal{C}_2$, we have

$$\frac{\log A_i}{\eta_{K+1}} = \log A_i \sqrt{\frac{\alpha_K H}{\log K}} \leq \sqrt{\frac{2c_\alpha H \log^2 A_i}{K}}, \tag{119}$$

where the first equality holds due to the definition of $\eta_{K+1}$.

- For term $\mathcal{C}_3$, we initially have

$$\frac{\eta_k\alpha_k}{1 - \alpha_k}\|q_{i,h}^k\|_\infty \leq \frac{\sqrt{\frac{2c_\alpha \log^2 K}{kH}}}{\frac{1}{2c_\alpha \log K}} \cdot H = \sqrt{\frac{8c_\alpha^3 H \log^4 K}{k}}. \tag{120}$$

Clearly, the right-hand side of (120) is upper bounded by $1/3$ for all $k$ obeying $k \geq c_9 H^2 \log^4 \frac{K}{\delta}$ for some large enough constant $c_9 > 0$. Consequently, one can derive

$$\tau_{i,h}$$

$$= \left\{ 3 \sum_{k=2}^{K} \alpha_k^K \frac{\eta_k^2 \alpha_k^2}{(1-\alpha_k)^2} \|q_{i,h}^k\|_\infty^3 \mathbf{1} \left( \frac{\eta_k \alpha_k}{1-\alpha_k} \|q_{i,h}^k\|_\infty > \frac{1}{3} \right) \right\}$$

$$+ \frac{5}{3} \alpha_1^K \eta_2 \|q_{i,h}^1\|_\infty^2 + 3\alpha_1^K \eta_2^2 \|q_{i,h}^1\|_\infty^3$$

$$\leq \frac{5}{3K^6} \sqrt{\frac{\log K}{H}} \|q_{i,h}^1\|_\infty^2 + \frac{(2c_\alpha \log K)^2}{K^6} \left\{ 3 \sum_{k=2}^{c_9 H^2 \log^4 \frac{K}{\delta}} \eta_k^2 \alpha_k^2 \|q_{i,h}^k\|_\infty^3 \right\} + \frac{3}{K^6} \frac{\log K}{H} \|q_{i,h}^1\|_\infty^3$$

$$\leq \frac{24c_\alpha^3 \log^4 K}{K^6 H} \left\{ \sum_{k=1}^{K} \frac{1}{k} H^3 \right\}$$

$$\leq \frac{24c_\alpha^3 H^3 \log^5 K}{K^6} \leq \frac{1}{K^4}, \tag{121}$$

where the second line comes from (115) and the fact that $K/2 > c_9 H \log^4 \frac{K}{\delta}$.

Combining previous three items, we can obtain that

$$\max_{a_i \in \mathcal{A}_i} \sum_{k=1}^{K} \alpha_k^K q_{i,h}^k(s, a_i) - \sum_{k=1}^{K} \alpha_k^K \langle \pi_{i,h}^k, q_{i,h}^k(s, \cdot) \rangle$$

$$\leq \frac{5(2c_\alpha)^{1.5}}{3} \cdot \frac{2H^{3/2} \log^2 K}{K^5} + \frac{20}{3} \sqrt{\frac{c_\alpha \log^2 K}{KH}} \sum_{k=K/2+1}^{K} \alpha_k^K \mathsf{Var}_{\pi_{i,h}^k(s)} \left( q_{i,h}^k(s, \cdot) \right)$$

$$+ \sqrt{\frac{2c_\alpha H \log^2 A_i}{K}} + \frac{1}{K^4}$$

$$\leq 10 \sqrt{\frac{c_\alpha \log^3(KA_i)}{KH}} \sum_{k=1}^{K} \alpha_k^K \mathsf{Var}_{\pi_{i,h}^k(s)} \left( q_{i,h}^k(s, \cdot) \right) + 2 \sqrt{\frac{c_\alpha H \log^3(KA_i)}{K}}, \tag{122}$$

According to the definition of $q_{i,h}^k(s, a_i)$ in the update rule of Algorithm 2, we have

$$\max_{a_i \in \mathcal{A}_i} \sum_{k=1}^{K} \alpha_k^K \left[ r_{i,h}^k(s, a_i) + \inf_{\mathcal{P} \in \mathcal{U}^{\sigma_i}(P_{i,h,s,a_i}^k)} \mathcal{P} \widehat{V}_{i,h+1} \right]$$

$$- \sum_{k=1}^{K} \alpha_k^K \mathbb{E}_{a_i \sim \pi_{i,h}^k(s)} \left[ r_{i,h}^k(s, a_i) + \inf_{\mathcal{P} \in \mathcal{U}^{\sigma_i}(P_{i,h,s,a_i}^k)} \mathcal{P} \widehat{V}_{i,h+1} \right]$$

$$= \max_{a_i \in \mathcal{A}_i} \sum_{k=1}^{K} \alpha_k^K q_{i,h}^k(s, a_i) - \sum_{k=1}^{K} \alpha_k^K \langle \pi_{i,h}^k, q_{i,h}^k(s, \cdot) \rangle$$

$$\leq 10 \sqrt{\frac{c_\alpha \log^3(KA_i)}{KH}} \sum_{k=1}^{K} \alpha_k^K \mathsf{Var}_{\pi_{i,h}^k(s)} \left( q_{i,h}^k(s, \cdot) \right) + 2 \sqrt{\frac{c_\alpha H \log^3(KA_i)}{K}} = \beta_{i,h}(s)$$

Moreover, according to the induction hypothesis, we have for all $s \in \mathcal{S}$

$$\mathbb{E}_{\pi \sim \widehat{\xi}} \left[ \overline{V}_{i,h}^{\star, \pi_{-i}}(s) \right] = \max_{a_i \in \mathcal{A}_i} \sum_{k=1}^{K} \alpha_k^K \left[ r_{i,h}^k(s, a_i) + \inf_{\mathcal{P} \in \mathcal{U}^{\sigma_i}(P_{i,h,s,a_i}^k)} \mathcal{P} \mathbb{E}_{\pi \sim \widehat{\xi}} \left[ \overline{V}_{i,h+1}^{\star, \pi_{-i}} \right] \right]$$

$$\leq \max_{a_i \in \mathcal{A}_i} \sum_{k=1}^{K} \alpha_k^K \left[ r_{i,h}^k(s, a_i) + \inf_{\mathcal{P} \in \mathcal{U}^{\sigma_i}(P_{i,h,s,a_i}^k)} \mathcal{P} \widehat{V}_{i,h+1} \right]$$

$$\leq \sum_{k=1}^{K} \alpha_k^K \mathbb{E}_{a_i \sim \pi_{i,h}^k(s)} \left[ r_{i,h}^k(s, a_i) + \inf_{\mathcal{P} \in \mathcal{U}^{\sigma_i}(P_{i,h,s,a_i}^k)} \mathcal{P} \widehat{V}_{i,h+1} \right] + \beta_{i,h}(s)$$

$$\leq \widehat{V}_{i,h}(s).$$

Thus, we finished the proof of the lemma.

### D.4.2 PROOF OF LEMMA 5

We will prove the lemma with induction argument. Initially, the base step $H+1$ trivially holds true, since we have

$$\widehat{V}_{i,H+1} = \mathbb{E}_{\pi\sim\widehat{\xi}}\left[\overline{V}_{i,H+1}^{\pi}\right] = 0.$$

Next, we assume that the lemma holds for step $h+1$, namely

$$\widehat{V}_{i,h+1} \geq \mathbb{E}_{\pi\sim\widehat{\xi}}\left[\overline{V}_{i,h+1}^{\pi}\right].$$

According to the definition of $\widehat{V}_{i,h}$ and $\mathbb{E}_{\pi\sim\widehat{\xi}}\left[\overline{V}_{i,h}^{\pi}\right]$, we have

$$\mathbb{E}_{\pi\sim\widehat{\xi}}\left[\overline{V}_{i,h}^{\pi}(s)\right] = \sum_{k=1}^{K}\alpha_k^K\mathbb{E}_{a_i\sim\pi_{i,h}^k(s)}\left[r_{i,h}^k(s,a_i) + \inf_{\mathcal{P}\in\mathcal{U}^{\sigma_i}(P_{i,h,s,a_i}^k)}\mathcal{P}\mathbb{E}_{\pi\sim\widehat{\xi}}\left[\overline{V}_{i,h+1}^{\pi}\right]\right]$$

$$\leq \sum_{k=1}^{K}\alpha_k^K\mathbb{E}_{a_i\sim\pi_{i,h}^k(s)}\left[r_{i,h}^k(s,a_i) + \inf_{\mathcal{P}\in\mathcal{U}^{\sigma_i}(P_{i,h,s,a_i}^k)}\mathcal{P}\widehat{V}_{i,h+1}\right]$$

$$\leq \sum_{k=1}^{K}\alpha_k^K\mathbb{E}_{a_i\sim\pi_{i,h}^k(s)}\left[r_{i,h}^k(s,a_i) + \inf_{\mathcal{P}\in\mathcal{U}^{\sigma_i}(P_{i,h,s,a_i}^k)}\mathcal{P}\widehat{V}_{i,h+1}\right] + \beta_{i,h}(s).$$

Since, we also trivially have $\mathbb{E}_{\pi\sim\widehat{\xi}}\left[\overline{V}_{i,h}^{\pi}(s)\right] \leq H-h+1$, we can deduce that for all $s\in\mathcal{S}$

$$\mathbb{E}_{\pi\sim\widehat{\xi}}\left[\overline{V}_{i,h}^{\pi}(s)\right]$$

$$\leq \min\left\{\sum_{k=1}^{K}\alpha_k^K\mathbb{E}_{a_i\sim\pi_{i,h}^k(s)}\left[r_{i,h}^k(s,a_i) + \inf_{\mathcal{P}\in\mathcal{U}^{\sigma_i}(P_{i,h,s,a_i}^k)}\mathcal{P}\widehat{V}_{i,h+1}\right] + \beta_{i,h}(s), H-h+1\right\}$$

$$= \widehat{V}_{i,h}(s).$$

Thus, we finished the proof of the lemma.

### D.4.3 PROOF OF LEMMA 6

Recall that for all $s\in\mathcal{S}$, bonus term $\beta_{i,h}(s)$ is defined as

$$\beta_{i,h}(s) = c_{\mathsf{b}}\sqrt{\frac{\log^3\left(\frac{KS\sum_{i=1}^n A_i}{\delta}\right)}{KH}}\sum_{k=1}^{K}\alpha_k^K\left\{\mathsf{Var}_{\pi_{i,h}^k(\cdot|s)}\left(q_{i,h}^k(s,\cdot)\right) + H\right\}. \tag{123}$$

For any $k\in[K]$, we have the following inequality for $\mathsf{Var}_{\pi_{i,h}^k(\cdot|s)}\left(q_{i,h}^k(s,\cdot)\right)$:

$$\mathsf{Var}_{\pi_{i,h}^k(\cdot|s)}\left(q_{i,h}^k(s,\cdot)\right)$$

$$\leq 2\mathsf{Var}_{\pi_{i,h}^k(\cdot|s)}\left(r_{i,h}^k(s,\cdot)\right) + 2\mathsf{Var}_{\pi_{i,h}^k(\cdot|s)}\left(\sum_{s'}\widehat{P}_{i,h}^{\pi_{-i}^k,\widehat{V}}(s'\mid s,\cdot)\widehat{V}_{i,h+1}(s')\right)$$

$$\overset{(i)}{\leq} 2 + 2\sum_{a_i\in\mathcal{A}_i}\pi_{i,h}^k(a_i\mid s)\widehat{P}_{i,h}^{\pi_{-i}^k,\widehat{V}}(\cdot\mid s,a_i)\left(\widehat{V}_{i,h+1}\circ\widehat{V}_{i,h+1}\right)$$

$$- \left(\sum_{a_i\in\mathcal{A}_i}\pi_{i,h}^k(a_i\mid s)\widehat{P}_{i,h}^{\pi_{-i}^k,\widehat{V}}(\cdot\mid s,a_i)\widehat{V}_{i,h+1}\right)^2$$

$$= 2 + 2\left\langle e_s, \mathsf{Var}_{\widehat{\underline{P}}_{i,h}^{\pi^k,\widehat{V}}}\widehat{V}_{i,h+1}\right\rangle. \tag{124}$$

where $e_s$ denotes an $S$-dimensional standard basis supported on the $s$-th element, and (i) holds due to the elementary fact that $\left| r_{i,h}^k(s,a_i) \right| \leq 1$ and $\left| \widehat{P}_{i,h}^{\pi_{-i}^k, \widehat{V}}(s' \mid s, a_i) \right| \leq 1$ for all $s, s' \in \mathcal{S}, a_i \in \mathcal{A}_i$. We insert the result of (124) back to (123), and rewrite the result in vector form, we can achieve that

$$\beta_{i,h} \leq 3c_{\mathsf{b}} \sqrt{\frac{\log^3(\frac{KS \sum_{i=1}^n A_i}{\delta})}{KH}} \left( H \cdot 1 + \mathsf{Var}_{\underline{\widehat{P}}_{i,h}^{\pi^k, \widehat{v}}} \widehat{V}_{i,h+1} \right)$$

### D.4.4 Proof of Lemma 7

To prove Lemma 7, we start by analyzing the value function of policy $\pi$ under uncertainty set $\sigma_i$. We first establish bounds on $\min_{s\in\mathcal{S}} \mathbb{E}_{\pi\sim\widehat{\xi}}\left[ \overline{V}_{i,h}^\pi(s) \right]$:

$$\min_{s\in\mathcal{S}} \mathbb{E}_{\pi\sim\widehat{\xi}}\left[ \overline{V}_{i,h}^\pi(s) \right]$$

$$= \min_{s\in\mathcal{S}} \sum_{k=1}^K \alpha_k^K \left[ \mathbb{E}_{a_i\sim\pi_{i,h}^k(s)}\left[ r_{i,h}^k(s,a_i) \right] + \mathbb{E}_{a_i\sim\pi_{i,h}^k(s)}\left[ \inf_{\mathcal{P}\in\mathcal{U}^{\sigma_i}\left( P_{i,h,s,a_i}^k \right)} \mathcal{P} \mathbb{E}_{\pi\sim\widehat{\xi}}\left[ \overline{V}_{i,h+1}^\pi(s) \right] \right] \right]$$

$$\geq \min_{s\in\mathcal{S}} \mathbb{E}_{\pi\sim\widehat{\xi}}\left[ \overline{V}_{i,h+1}^\pi(s) \right].$$

This follows from the robust Bellman equation (26).

Next, we examine $\max_{s\in\mathcal{S}} \mathbb{E}_{\pi\sim\widehat{\xi}}\left[ \overline{V}_{i,h}^\pi(s) \right]$:

$$\max_{s\in\mathcal{S}} \mathbb{E}_{\pi\sim\widehat{\xi}}\left[ \overline{V}_{i,h}^\pi(s) \right]$$

$$= \max_{s\in\mathcal{S}} \sum_{k=1}^K \alpha_k^K \left[ \mathbb{E}_{a_i\sim\pi_{i,h}^k(s)}\left[ r_{i,h}^k(s,a_i) \right] + \mathbb{E}_{a_i\sim\pi_{i,h}^k(s)}\left[ \inf_{\mathcal{P}\in\mathcal{U}^{\sigma_i}\left( P_{i,h,s,a_i}^k \right)} \mathcal{P} \mathbb{E}_{\pi\sim\widehat{\xi}}\left[ \overline{V}_{i,h+1}^\pi(s) \right] \right] \right]$$

$$\leq 1 + \sum_{k=1}^K \alpha_k^K \max_{(s,a_i)\in\mathcal{S}\times\mathcal{A}_i} \left[ \inf_{\mathcal{P}\in\mathcal{U}^{\sigma_i}\left( P_{h,s,a_i}^{\pi_{-i}} \right)} \mathcal{P} \mathbb{E}_{\pi\sim\widehat{\xi}}\left[ \overline{V}_{i,h+1}^\pi(s) \right] \right]. \tag{125}$$

We now construct an auxiliary distribution vector $P'_{h,s,a_i} \in \mathbb{R}^S$ by strictly reducing some elements of $P_{h,s,a_i}^{\pi_{-i}}$ such that:

$$0 \leq P'_{h,s,a_i} \leq P_{i,h,s,a_i}^k \quad \text{and} \quad \sum_{s'\in\mathcal{S}} P_{i,h,s,a_i}^k(s') - P'_{h,s,a_i}(s') = \left\| P'_{h,s,a_i} - P_{i,h,s,a_i}^k \right\|_1 = \sigma_i. \tag{126}$$

Let $e_{s_{i,h}^\star}$ denote the standard basis vector supported on $s_{i,h}^\star$. We can show:

$$\frac{1}{2} \left\| P'_{h,s,a_i} + \sigma_i \left[ e_{s_{i,h}^\star} \right]^\top - P_{i,h,s,a_i}^k \right\|_1 \leq \frac{1}{2} \left\| P'_{h,s,a_i} - P_{i,h,s,a_i}^k \right\|_1 + \frac{1}{2} \left\| \sigma_i \left[ e_{s_{i,h}^\star} \right]^\top \right\|_1 \leq \sigma_i, \tag{127}$$

where the first inequality follows from the triangle inequality of the total variation distance.

From (127), we conclude that:

$$\inf_{\mathcal{P}\in\mathcal{U}^{\sigma_i}\left( P_{i,h,s,a_i}^k \right)} \mathcal{P} \mathbb{E}_{\pi\sim\widehat{\xi}}\left[ \overline{V}_{i,h+1}^\pi \right]$$

$$\leq \left( P'_{h,s,a_i} + \sigma_i \left[ e_{s_{i,h}^\star} \right]^\top \right) \mathbb{E}_{\pi\sim\widehat{\xi}}\left[ \overline{V}_{i,h+1}^\pi \right] \tag{128}$$

$$\leq \left\| P'_{h,s,a_i} \right\|_1 \left\| \mathbb{E}_{\pi\sim\widehat{\xi}}\left[ \overline{V}_{i,h+1}^\pi \right] \right\|_\infty + \sigma_i \mathbb{E}_{\pi\sim\widehat{\xi}}\left[ \overline{V}_{i,h+1}^\pi(s_{i,h+1}^\star) \right]$$

$$\leq (1-\sigma_i) \max_{s\in\mathcal{S}} \mathbb{E}_{\pi\sim\widehat{\xi}}\left[ \overline{V}_{i,h+1}^\pi(s) \right] + \sigma_i \min_{s\in\mathcal{S}} \mathbb{E}_{\pi\sim\widehat{\xi}}\left[ \overline{V}_{i,h+1}^\pi(s) \right].$$

Substituting (128) into (125) yields:

$$\max_{s\in\mathcal{S}} \mathbb{E}_{\pi\sim\widehat{\xi}}\left[\overline{V}_{i,h}^{\pi}(s)\right] \le 1 + (1-\sigma_i)\max_{s\in\mathcal{S}} \mathbb{E}_{\pi\sim\widehat{\xi}}\left[\overline{V}_{i,h+1}^{\pi}(s)\right] + \sigma_i\min_{s\in\mathcal{S}} \mathbb{E}_{\pi\sim\widehat{\xi}}\left[\overline{V}_{i,h+1}^{\pi}(s)\right]. \quad (129)$$

Combining (125) and (129) gives:

$$\max_{s\in\mathcal{S}} \mathbb{E}_{\pi\sim\widehat{\xi}}\left[\overline{V}_{i,h}^{\pi}(s)\right] - \min_{s\in\mathcal{S}} \mathbb{E}_{\pi\sim\widehat{\xi}}\left[\overline{V}_{i,h}^{\pi}(s)\right]$$

$$\le 1 + (1-\sigma_i)\left(\max_{s\in\mathcal{S}} \mathbb{E}_{\pi\sim\widehat{\xi}}\left[\overline{V}_{i,h+1}^{\pi}(s)\right] - \min_{s\in\mathcal{S}} \mathbb{E}_{\pi\sim\widehat{\xi}}\left[\overline{V}_{i,h+1}^{\pi}(s)\right]\right)$$

$$\le 1 + (1-\sigma_i)\left[1 + (1-\sigma_i)\left(\max_{s\in\mathcal{S}} \mathbb{E}_{\pi\sim\widehat{\xi}}\left[\overline{V}_{i,h+2}^{\pi}(s)\right] - \min_{s\in\mathcal{S}} \mathbb{E}_{\pi\sim\widehat{\xi}}\left[\overline{V}_{i,h+2}^{\pi}(s)\right]\right)\right]$$

$$\le \cdots \le \frac{1-(1-\sigma_i)^{H-h}}{\sigma_i} \le \frac{1}{\sigma_i}. \quad (130)$$

Combining this with the basic fact that $\max_{s\in\mathcal{S}} \mathbb{E}_{\pi\sim\widehat{\xi}}\left[\overline{V}_{i,h}^{\pi}(s)\right] - \min_{s\in\mathcal{S}} \mathbb{E}_{\pi\sim\widehat{\xi}}\left[\overline{V}_{i,h}^{\pi}(s)\right] \le H - h + 1$, we complete the proof.

### D.4.5 PROOF FOR LEMMA 8

We introduce the following notation for the value function at time $h$:

$$\forall h \in [H], \quad \overline{V}_{i,h}^{\text{span}} := \mathbb{E}_{\pi\sim\widehat{\xi}}\left[\overline{V}_{i,h}^{\pi}(s)\right] - \min_{s'\in\mathcal{S}} \mathbb{E}_{\pi\sim\widehat{\xi}}\left[\overline{V}_{i,h}^{\pi}(s')\right], \quad (131)$$

which normalizes the value function $\overline{V}_{i,h}^{\pi}$. This definition leads to the following bound:

$$\left\|\overline{V}_{i,h}^{\text{span}}\right\|_{\infty} \le \min\left\{\frac{1}{\sigma_i}, H-h+1\right\}, \quad (132)$$

a result derived using Lemma 7. With this notation established, we now consider any transition kernel $P' \in \mathbb{R}^S$ and any $\widetilde{P} \in \mathbb{R}^S$ such that $\widetilde{P} \in \mathcal{U}^{\sigma_i}(P')$. For all $(i,h) \in [n] \times [H]$, we analyze the variance difference between the value functions under these kernels:

$$\left|\text{Var}_{P'}\left(\mathbb{E}_{\pi\sim\widehat{\xi}}\left[\overline{V}_{i,h}^{\pi}\right]\right) - \text{Var}_{\widetilde{P}}\left(\mathbb{E}_{\pi\sim\widehat{\xi}}\left[\overline{V}_{i,h}^{\pi}\right]\right)\right|$$

$$= \left|\text{Var}_{P'}\left(\overline{V}_{i,h}^{\text{span}}\right) - \text{Var}_{\widetilde{P}}\left(\overline{V}_{i,h}^{\text{span}}\right)\right|$$

$$\le \left\|\widetilde{P} - P'\right\|_1 \left\|\overline{V}_{i,h}^{\text{span}}\right\|_{\infty}$$

$$\le \sigma_i\left(\min\left\{\frac{1}{\sigma_i}, H-h+1\right\}\right)^2 \le \min\left\{\frac{1}{\sigma_i}, H-h+1\right\}. \quad (133)$$

### D.4.6 PROOF OF LEMMA 9

Analogous to Appendix D.4.9, we introduce some auxiliary values and reward functions to control

$$\sum_{j=h}^{H}\left\langle b_h^j, \text{Var}_{\sum_{k=1}^{K}\alpha_k^K \widehat{P}_{i,j}^{\pi^k,\overline{V}}}\left(\mathbb{E}_{\pi\sim\widehat{\xi}}\left[\overline{V}_{i,j+1}^{\pi}\right]\right)\right\rangle$$

as below for any time step $h$ and agent $i$.

**Definition 3.** *For any time step $h \in [H]$ and the $i$-th agent, we denote $\overline{V}_h^{\min} := \min_{s\in\mathcal{S}} \mathbb{E}_{\pi\sim\widehat{\xi}}\left[\overline{V}_{i,h}^{\pi}(s)\right]$ as the minimum value of all the entries in vector $\mathbb{E}_{\pi\sim\widehat{\xi}}\left[\overline{V}_{i,h}^{\pi}\right]$. We further define $\overline{V}_h' := \mathbb{E}_{\pi\sim\widehat{\xi}}\left[\overline{V}_{i,h}^{\pi}\right] - \overline{V}_h^{\min}\mathbf{1}$ as the truncated value function. Eventually for reward function, we define $\overline{r}_{i,h}^{\min} = \sum_{k=1}^{K}\alpha_k^K \mathbb{E}_{a_i\sim\pi_{i,h}^k} r_{i,h}^k(\cdot,a_i) + \left(\overline{V}_{h+1}^{\min} - \overline{V}_h^{\min}\right)\mathbf{1}$ as the truncated reward function..*

Then applying the robust Bellman's consistency equation in (26) gives

$$
\begin{aligned}
\overline{V}_h' &= \mathbb{E}_{\pi \sim \widehat{\xi}}\left[\overline{V}_{i,h}^\pi\right] - \overline{V}_h^{\min}\mathbf{1} \\
&= \sum_{k=1}^K \alpha_k^K \mathbb{E}_{a_i \sim \pi_{i,h}^k} r_{i,h}^k(\cdot, a_i) + \sum_{k=1}^K \alpha_k^K \widehat{\underline{P}}_{i,h}^{\pi^k, \overline{V}} \mathbb{E}_{\pi \sim \widehat{\xi}}\left[\overline{V}_{i,h+1}^\pi\right] - \overline{V}_h^{\min}\mathbf{1} \\
&= \sum_{k=1}^K \alpha_k^K \mathbb{E}_{a_i \sim \pi_{i,h}^k} r_{i,h}^k(\cdot, a_i) + \left(\overline{V}_{h+1}^{\min} - \overline{V}_h^{\min}\right)\mathbf{1} + \sum_{k=1}^K \alpha_k^K \widehat{\underline{P}}_{i,h}^{\pi^k, \overline{V}} \overline{V}_{h+1}' \\
&= \overline{r}_{i,h}^{\min} + \sum_{k=1}^K \alpha_k^K \widehat{\underline{P}}_{i,h}^{\pi^k, \overline{V}} \overline{V}_{h+1}'.
\end{aligned}
\tag{134}
$$

The above fact leads to

$$
\begin{aligned}
&\mathsf{Var}_{\sum_{k=1}^K \alpha_k^K \widehat{\underline{P}}_{i,h}^{\pi^k, \overline{V}}}\left(\mathbb{E}_{\pi \sim \widehat{\xi}}\left[\overline{V}_{i,h+1}^\pi\right]\right) \\
&\stackrel{(i)}{=} \mathsf{Var}_{\sum_{k=1}^K \alpha_k^K \widehat{\underline{P}}_{i,h}^{\pi^k, \overline{V}}}\left(\overline{V}_{h+1}'\right) \\
&= \sum_{k=1}^K \alpha_k^K \widehat{\underline{P}}_{i,h}^{\pi^k, \overline{V}}\left(\overline{V}_{h+1}' \circ \overline{V}_{h+1}'\right) - \left(\sum_{k=1}^K \alpha_k^K \widehat{\underline{P}}_{i,h}^{\pi^k, \overline{V}} \overline{V}_{h+1}'\right) \circ \left(\sum_{k=1}^K \alpha_k^K \widehat{\underline{P}}_{i,h}^{\pi^k, \overline{V}} \overline{V}_{h+1}'\right)
\end{aligned}
$$

where (i) follows from the fact that $\mathsf{Var}_{\sum_{k=1}^K \alpha_k^K \widehat{\underline{P}}_{i,h}^{\pi^k, V}}(V - b\mathbf{1}) = \mathsf{Var}_{\sum_{k=1}^K \alpha_k^K \widehat{\underline{P}}_{i,h}^{\pi^k, V}}(V)$ for any value vector $V \in \mathbb{R}^S$ and scalar $b$. According to (134) and (26), we have

$$
\begin{aligned}
&\mathsf{Var}_{\sum_{k=1}^K \alpha_k^K \widehat{\underline{P}}_{i,h}^{\pi^k, \overline{V}}}\left(\mathbb{E}_{\pi \sim \widehat{\xi}}\left[\overline{V}_{i,h+1}^\pi\right]\right) \\
&= \sum_{k=1}^K \alpha_k^K \widehat{\underline{P}}_{i,h}^{\pi^k, \overline{V}}\left(\overline{V}_{h+1}' \circ \overline{V}_{h+1}'\right) - \left(\overline{V}_h' - \overline{r}_{i,h}^{\min}\right)^{\circ 2} \\
&= \sum_{k=1}^K \alpha_k^K \widehat{\underline{P}}_{i,h}^{\pi^k, \overline{V}}\left(\overline{V}_{h+1}' \circ \overline{V}_{h+1}'\right) - \overline{V}_h' \circ \overline{V}_h' + 2\overline{V}_h' \circ \overline{r}_{i,h}^{\min} - \overline{r}_{i,h}^{\min} \circ \overline{r}_{i,h}^{\min} \\
&\leq \sum_{k=1}^K \alpha_k^K \widehat{\underline{P}}_{i,h}^{\pi^k, \overline{V}}\left(\overline{V}_{h+1}' \circ \overline{V}_{h+1}'\right) - \overline{V}_h' \circ \overline{V}_h' + 2\|\overline{V}_h'\|_\infty \mathbf{1},
\end{aligned}
$$

where the last inequality arises from $\overline{r}_{i,h}^{\min} \leq \sum_{k=1}^K \alpha_k^K \mathbb{E}_{a_i \sim \pi^k} r_{i,h}^k(\cdot, a_i) \leq 1$ since $\overline{V}_{h+1}^{\min} - \overline{V}_h^{\min} \leq 0$ by definition.

Consequently, combining (145) and the definition of $b_h^j$ in (92), we arrive at

$$
\begin{aligned}
&\sum_{j=h}^H \left\langle b_h^j, \mathsf{Var}_{\sum_{k=1}^K \alpha_k^K \widehat{\underline{P}}_{i,h}^{\pi^k, V}}\left(\mathbb{E}_{\pi \sim \widehat{\xi}}\left[\overline{V}_{i,j+1}^\pi\right]\right)\right\rangle \\
&= \sum_{j=h}^H \left(b_h^j\right)^\top \left(\sum_{k=1}^K \alpha_k^K \widehat{\underline{P}}_{i,h}^{\pi^k, \overline{V}}\left(\overline{V}_{j+1}' \circ \overline{V}_{j+1}'\right) - \overline{V}_j' \circ \overline{V}_j' + 2\|\overline{V}_h'\|_\infty \mathbf{1}\right) \\
&\stackrel{(i)}{\leq} \sum_{j=h}^H \left[\left(b_h^j\right)^\top \left(\sum_{k=1}^K \alpha_k^K \widehat{\underline{P}}_{i,h}^{\pi^k, \overline{V}}\left(\overline{V}_{j+1}' \circ \overline{V}_{j+1}'\right) - \overline{V}_j' \circ \overline{V}_j'\right)\right] + 2H\left\|\overline{V}_h'\right\|_\infty
\end{aligned}
$$

where (i) and the last inequality hold by the fact $\left\|\overline{V}_h'\right\|_\infty \geq \left\|\overline{V}_{h+1}'\right\|_\infty \geq \cdots \geq \left\|\overline{V}_H'\right\|_\infty$. Further according to basic calculus, we have

$$
\sum_{j=h}^H \left\langle b_h^j, \mathsf{Var}_{\sum_{k=1}^K \alpha_k^K \widehat{\underline{P}}_{i,h}^{\pi^k, V}}\left(\mathbb{E}_{\pi \sim \widehat{\xi}}\left[\overline{V}_{i,j+1}^\pi\right]\right)\right\rangle
$$

$$= \sum_{j=h}^{H} \left[ \left( b_h^{j+1} \right)^\top \left( \overline{V}'_{j+1} \circ \overline{V}'_{j+1} \right) - (b_h^j)^\top \left( \overline{V}'_j \circ \overline{V}'_j \right) \right] + 2H \left\| \overline{V}'_h \right\|_\infty$$

$$\leq \left\| b_h^{H+1} \right\|_1 \left\| \overline{V}'_{H+1} \circ \overline{V}'_{H+1} \right\|_\infty + 2H \left\| \overline{V}'_h \right\|_\infty$$

$$\leq 3H \left\| \overline{V}'_h \right\|_\infty . \tag{135}$$

### D.4.7 Proof of Lemma 10

To prove the inequality involving $P_{i,h}^{\widehat{\pi}_{-i}^k, V} V$ and $\widehat{P}_{i,h}^{\widehat{\pi}_{-i}^k, V} V$, we start by analyzing the absolute difference between these terms:

$$\left| P_{i,h,s,a_i}^{\widehat{\pi}_{-i}^k, V} V - \widehat{P}_{i,h,s,a_i}^{\widehat{\pi}_{-i}^k, V} V \right|$$

$$= \left| \inf_{\mathcal{P} \in \mathcal{U}^{\sigma_i} \left( P_{h,s,a_i}^{\widehat{\pi}_{-i}^k} \right)} \mathcal{P}V - \inf_{\mathcal{P} \in \mathcal{U}^{\sigma_i} \left( \widehat{P}_{i,h,s,a_i}^{\widehat{\pi}_{-i}^k} \right)} \mathcal{P}V \right|$$

$$\stackrel{(i)}{=} \left| \mathbb{E}_{a_i \sim \widehat{\pi}_{i,h}^k} \max_{\alpha \in [\min_s V(s), \max_s V(s)]} \left[ P_{h,s,a_i}^{\widehat{\pi}_{-i}^k} [V]_\alpha - \sigma_i \left( \alpha - \min_{s'} [V]_\alpha(s') \right) \right] \right.$$

$$\left. - \mathbb{E}_{a_i \sim \widehat{\pi}_{i,h}^k} \max_{\alpha \in [\min_s V(s), \max_s V(s)]} \left[ \widehat{P}_{i,h,s,a_i}^{\widehat{\pi}_{-i}^k} [V]_\alpha - \sigma_i \left( \alpha - \min_{s'} [V]_\alpha(s') \right) \right] \right|$$

$$\leq \mathbb{E}_{a_i \sim \widehat{\pi}_{i,h}^k} \max_{\alpha \in [\min_s V(s), \max_s V(s)]} \left| P_{h,s,a_i}^{\widehat{\pi}_{-i}^k} [V]_\alpha - \widehat{P}_{i,h,s,a_i}^{\widehat{\pi}_{-i}^k} [V]_\alpha \right|, \tag{136}$$

where (i) follows from applying the robust Bellman equation (26), and the last inequality uses the fact that the maximum operator is 1-Lipschitz.

Next, we apply Bernstein's inequality to bound the difference between $P_{h,s,a_i}^{\widehat{\pi}_{-i}^k} [V]_\alpha$ and $\widehat{P}_{i,h,s,a_i}^{\widehat{\pi}_{-i}^k} [V]_\alpha$ for fixed $\alpha$, $k$, and $(s, a_i)$. With probability at least $1 - \delta$, we have:

$$\left| P_{h,s,a_i}^{\widehat{\pi}_{-i}^k} [V]_\alpha - \widehat{P}_{i,h,s,a_i}^{\widehat{\pi}_{-i}^k} [V]_\alpha \right| \leq \sqrt{\frac{2 \log \left( \frac{2}{\delta} \right)}{N}} \sqrt{\mathsf{Var}_{P_{h,s,a_i}^{\widehat{\pi}_{-i}^k}} ([V]_\alpha)} + \frac{2H \log \left( \frac{2}{\delta} \right)}{3N}. \tag{137}$$

To extend this bound to all $(s, a_i)$, we use a uniform bound over an $\varepsilon_1$-net for $\alpha$. The net size $|N_{\varepsilon_1}| \leq \frac{3H}{\varepsilon_1}$ allows us to apply the union bound:

$$\left| P_{h,s,a_i}^{\widehat{\pi}_{-i}^k} [V]_\alpha - \widehat{P}_{i,h,s,a_i}^{\widehat{\pi}_{-i}^k} [V]_\alpha \right|$$

$$\leq \max_{\alpha \in N_{\varepsilon_1}} \left| P_{h,s,a_i}^{\widehat{\pi}_{-i}^k} [V]_\alpha - \widehat{P}_{i,h,s,a_i}^{\widehat{\pi}_{-i}^k} [V]_\alpha \right| + \varepsilon_1$$

$$\leq \sqrt{\frac{2 \log \left( \frac{2S \sum_{i=1}^n A_i |N_{\varepsilon_1}| Kn}{\delta} \right)}{N}} \sqrt{\mathsf{Var}_{P_{h,s,a_i}^{\widehat{\pi}_{-i}^k}} (V)} + \frac{2H \log \left( \frac{2S \sum_{i=1}^n A_i |N_{\varepsilon_1}| Kn}{\delta} \right)}{3N} + \varepsilon_1$$

$$\leq \sqrt{\frac{2 \log \left( \frac{2S \sum_{i=1}^n A_i NKn}{\delta} \right)}{N}} \sqrt{\mathsf{Var}_{P_{h,s,a_i}^{\widehat{\pi}_{-i}^k}} (V)} + \frac{H \log \left( \frac{2S \sum_{i=1}^n A_i NKn}{\delta} \right)}{N}, \tag{138}$$

where the last steps use that $\varepsilon_1 = \frac{H \log \left( \frac{2S \sum_{i=1}^n A_i NKn}{\delta} \right)}{3N}$ and $|N_{\varepsilon_1}| \leq 9N$.

Inserting this back into (136) gives:

$$\left| P_{i,h}^{\widehat{\pi}_{-i}^k, V} V - \widehat{P}_{i,h}^{\widehat{\pi}_{-i}^k, V} V \right|$$

$$\leq \sqrt{\frac{2\log\left(\frac{2S\sum_{i=1}^{n}A_iNKn}{\delta}\right)}{N}}\sqrt{\mathsf{Var}_{P_h^{\widehat{\pi}_{-i}^k}}(V)} + \frac{H\log\left(\frac{2S\sum_{i=1}^{n}A_iNKn}{\delta}\right)}{N}1$$

$$\leq 3\sqrt{\frac{H^2\log\left(\frac{2S\sum_{i=1}^{n}A_iNKn}{\delta}\right)}{N}}1.$$

This completes the proof by showing that the bound holds uniformly over all $(s, a_i) \in \mathcal{S} \times \mathcal{A}_i$.

### D.4.8 PROOF OF LEMMA 11

Before proving Lemma 11, we first state a modified version of the Freedman inequality for martingales, which is crucial for our analysis.

**Theorem 5.** *Suppose $Y_n = \sum_{k=1}^{n} X_k \in \mathbb{R}$, where $\{X_k\}$ is a real-valued scalar sequence such that*

$$|X_k| \leq R \qquad and \qquad \mathbb{E}\left[X_k \mid \{X_j\}_{j<k}\right] = 0 \quad for\ all\ k \geq 1$$

*for some constant $R > 0$. Define*

$$W_n := \sum_{k=1}^{n} \mathbb{E}_{k-1}[X_k^2],$$

*where $\mathbb{E}_{k-1}$ denotes the conditional expectation given $\{X_j\}_{j<k}$. For any $\kappa > 0$, with probability at least $1 - \delta$, the following holds:*

$$|Y_n| \leq \sqrt{8W_n\log\frac{3n}{\delta}} + 5R\log\frac{3n}{\delta} \leq \kappa W_n + \left(\frac{2}{\kappa} + 5R\right)\log\frac{3n}{\delta}. \tag{139}$$

*Proof.* Suppose deterministically that $W_n \leq \sigma^2$ for some $\sigma^2$. According to Li et al. (2024), with probability at least $1 - \delta$, we have

$$|Y_n| \leq \sqrt{8\max\left\{W_n, \frac{\sigma^2}{2^K}\right\}\log\frac{2K}{\delta}} + \frac{4}{3}R\log\frac{2K}{\delta}.$$

for any positive integer $K \geq 1$. Utilizing the trivial bound $W_n \leq nR^2$, set $\sigma^2 = nR^2$ and $K = \log_2 n$. Then:

$$|Y_n| \leq \sqrt{8\max\left\{W_n, R^2\right\}\log\frac{4\log_2 n}{\delta}} + \frac{4}{3}R\log\frac{4\log_2 n}{\delta}$$

$$\leq \sqrt{8W_n\log\frac{3n}{\delta}} + \sqrt{8R^2\log\frac{3n}{\delta}} + \frac{4}{3}R\log\frac{3n}{\delta}$$

$$\leq \sqrt{8W_n\log\frac{3n}{\delta}} + 5R\log\frac{3n}{\delta},$$

where we used $4\log_2 n \leq 3n$ for any integer $n \geq 1$. This establishes the first inequality in (139). The second inequality follows from the elementary inequality $2ab \leq a^2 + b^2$. $\qquad\square$

To prove Lemma 11, we apply Lemma 5. Define

$$R := \max_{k\in[K]}\left|\alpha_k^K\left\langle\widehat{\pi}_{i,h}^k(s), r_{i,h}^k(s,\cdot)\right\rangle\right| \leq \left\{\max_{k\in[K]}\alpha_k^K\right\}\left\{\max_{k\in[K]}\|\widehat{\pi}_{i,h}^k(s)\|_1\|r_{i,h}^k\|_\infty\right\} \leq \frac{2c_\alpha\log K}{K},$$

where the first line uses Lemma 19. We further define

$$W_K = \sum_{k=1}^{K}(\alpha_k^K)^2\mathsf{Var}_{h,k-1}\left(\left\langle\widehat{\pi}_{i,h}^k(s), r_{i,h}^k(s,\cdot)\right\rangle\right)$$

$$\leq \left\{\max_{k\in[K]}\alpha_k^K\right\}\left\{\sum_{k=1}^{K}\alpha_k^K\mathsf{Var}_{h,k-1}\left(\left\langle\widehat{\pi}_{i,h}^k(s), r_{i,h}^k(s,\cdot)\right\rangle\right)\right\}$$

$$\leq \frac{2c_\alpha \log K}{K} \sum_{k=1}^{K} \alpha_k^K \mathsf{Var}_{h,k-1}(r_{i,h}^k(s)),$$

where we use variance operator $\mathsf{Var}_{h,k-1}[\cdot]$ to denote the variance conditional on what happens before the beginning of the $k$-th round of data collection for step $h$. Applying Freedman's inequality (Lemma 5) with $\kappa_1 = \sqrt{K \log(K/\delta)}$, we obtain

$$\left| \sum_{k=1}^{K} \alpha_k^K \mathbb{E}_{a_i \sim \widehat{\pi}_{i,h}^k} r_{i,h}^k(s, a_i) - \sum_{k=1}^{K} \alpha_k^K \mathbb{E}_{a_i \sim \widehat{\pi}_{i,h}^k} r_{i,h}^{\widehat{\pi}_{-i}^k}(s, a_i) \right|$$

$$\leq \kappa_1 W_K + \left( \frac{2}{\kappa_1} + 5R_1 \right) \log \left( \frac{3K}{\delta} \right)$$

$$\leq 2c_\alpha \sqrt{\frac{\log^3 \left( \frac{K}{\delta} \right)}{K} \sum_{k=1}^{K} \alpha_k^K \mathsf{Var}_{h,k-1} \left( r_{i,h}^k(s) \right)} + \left( 2\sqrt{\frac{1}{K \log \left( \frac{K}{\delta} \right)}} + \frac{10c_\alpha \log K}{K} \right) \log \left( \frac{3K}{\delta} \right)$$

$$\leq 2c_\alpha \sqrt{\frac{\log^3 \left( \frac{K}{\delta} \right)}{K} \sum_{k=1}^{K} \alpha_k^K \mathsf{Var}_{h,k-1} \left( r_{i,h}^k(s) \right)} + 4\sqrt{\frac{\log \left( \frac{3K}{\delta} \right)}{K}},$$

with probability at least $1 - \delta$. Taking a union bound over all $s \in \mathcal{S}$, there exists an absolute constant $c_r$ such that

$$\sum_{k=1}^{K} \alpha_k^K \left| r_{i,h}^{\widehat{\pi}^k} - \overline{r}_{i,h}^{\widehat{\pi}^k} \right| \leq c_r \sqrt{\frac{\log(KS/\delta)}{K}} 1.$$

### D.4.9   PROOF OF LEMMA 13

In this section, we want to take the accessible range of the robust value function $\mathbb{E}_{\pi \sim \zeta} \left[ \overline{V}_{i,j+1}^{\pi} \right]$ into consideration when controlling $\sum_{j=h}^{H} \left\langle d_h^j, \mathsf{Var}_{\sum_{k=1}^{K} \alpha_k^K \underline{P}_{i,j}^{\widehat{\pi}^k, \overline{V}}} \left( \mathbb{E}_{\pi \sim \zeta} \left[ \overline{V}_{i,j+1}^{\pi} \right] \right) \right\rangle$. Towards this, we introduce some auxiliary values and reward functions as below.

**Definition 4.** *For any time step* $h \in [H]$ *and the $i$-th agent, we denote* $\overline{V}_h^{\min} := \min_{s \in \mathcal{S}} \mathbb{E}_{\pi \sim \zeta} \left[ \overline{V}_{i,h}^{\pi}(s) \right]$ *as the minimum value of all the entries in vector* $\mathbb{E}_{\pi \sim \zeta} \left[ \overline{V}_{i,h}^{\pi} \right]$. *We further define* $\overline{V}_h' := \mathbb{E}_{\pi \sim \zeta} \left[ \overline{V}_{i,h}^{\pi} \right] - \overline{V}_h^{\min} 1$ *as the truncated value function. Eventually for reward function, we define* $\overline{r}_{i,h}^{\min} = \sum_{k=1}^{K} \alpha_k^K \mathbb{E}_{a_i \sim \widehat{\pi}_{i,h}^k} r_{i,h}^k(\cdot, a_i) + \left( \overline{V}_{h+1}^{\min} - \overline{V}_h^{\min} \right) 1$ *as the truncated reward function..*

With above notation, we introduce the following fact of $\overline{V}_h'$:

$$\overline{V}_h' = \mathbb{E}_{\pi \sim \zeta} \left[ \overline{V}_{i,h}^{\pi} \right] - \overline{V}_h^{\min} 1$$

$$\overset{\text{(i)}}{=} \sum_{k=1}^{K} \alpha_k^K \mathbb{E}_{a_i \sim \widehat{\pi}_{i,h}^k} r_{i,h}^k(\cdot, a_i) + \sum_{k=1}^{K} \alpha_k^K \underline{\widehat{P}}_{i,h}^{\widehat{\pi}^k, \overline{V}} \mathbb{E}_{\pi \sim \zeta} \left[ \overline{V}_{i,h+1}^{\pi} \right] - \overline{V}_h^{\min} 1$$

$$= \sum_{k=1}^{K} \alpha_k^K \mathbb{E}_{a_i \sim \widehat{\pi}_{i,h}^k} r_{i,h}^k(\cdot, a_i) + \sum_{k=1}^{K} \alpha_k^K \underline{P}_{i,h}^{\widehat{\pi}^k, \overline{V}} \mathbb{E}_{\pi \sim \zeta} \left[ \overline{V}_{i,h+1}^{\pi} \right]$$

$$+ \left( \sum_{k=1}^{K} \alpha_k^K \underline{\widehat{P}}_{i,h}^{\widehat{\pi}^k, \overline{V}} - \sum_{k=1}^{K} \alpha_k^K \underline{P}_{i,h}^{\widehat{\pi}^k, \overline{V}} \right) \mathbb{E}_{\pi \sim \zeta} \left[ \overline{V}_{i,h+1}^{\pi} \right] - \overline{V}_h^{\min} 1$$

$$= \sum_{k=1}^{K} \alpha_k^K \mathbb{E}_{a_i \sim \widehat{\pi}_{i,h}^k} r_{i,h}^k(\cdot, a_i) + \left( \overline{V}_{h+1}^{\min} - \overline{V}_h^{\min} \right) 1$$

$$+ \sum_{k=1}^{K} \alpha_k^K \underline{P}_{i,h}^{\widehat{\pi}^k, \overline{V}} \overline{V}'_{h+1} + \Big( \sum_{k=1}^{K} \alpha_k^K \widehat{\underline{P}}_{i,h}^{\widehat{\pi}^k, \overline{V}} - \sum_{k=1}^{K} \alpha_k^K \underline{P}_{i,h}^{\widehat{\pi}^k, \overline{V}} \Big) \mathbb{E}_{\pi \sim \zeta} \Big[ \overline{V}_{i,h+1}^{\pi} \Big]$$

$$\overset{(i)}{=} \overline{r}_{i,h}^{\min} + \sum_{k=1}^{K} \alpha_k^K \underline{P}_{i,h}^{\widehat{\pi}^k, \overline{V}} \overline{V}'_{h+1} + \Big( \sum_{k=1}^{K} \alpha_k^K \widehat{\underline{P}}_{i,h}^{\widehat{\pi}^k, \overline{V}} - \sum_{k=1}^{K} \alpha_k^K \underline{P}_{i,h}^{\widehat{\pi}^k, \overline{V}} \Big) \mathbb{E}_{\pi \sim \zeta} \Big[ \overline{V}_{i,h+1}^{\pi} \Big], \quad (140)$$

where (i) holds by the robust Bellman's consistency equation of $\mathbb{E}_{\pi \sim \zeta} \Big[ \overline{V}_{i,h}^{\pi} \Big]$. With the above fact in hand, we control $\mathrm{Var}_{\sum_{k=1}^{K} \alpha_k^K \underline{P}_{i,h}^{\widehat{\pi}^k, \overline{V}}} \Big( \mathbb{E}_{\pi \sim \zeta} \Big[ \overline{V}_{i,h+1}^{\pi} \Big] \Big)$ as follows:

$$\mathrm{Var}_{\sum_{k=1}^{K} \alpha_k^K \underline{P}_{i,h}^{\widehat{\pi}^k, \overline{V}}} \Big( \mathbb{E}_{\pi \sim \zeta} \Big[ \overline{V}_{i,h+1}^{\pi} \Big] \Big)$$

$$\overset{(i)}{=} \mathrm{Var}_{\sum_{k=1}^{K} \alpha_k^K \underline{P}_{i,h}^{\widehat{\pi}^k, \overline{V}}} \Big( \overline{V}'_{h+1} \Big)$$

$$= \sum_{k=1}^{K} \alpha_k^K \underline{P}_{i,h}^{\widehat{\pi}^k, \overline{V}} \Big( \overline{V}'_{h+1} \circ \overline{V}'_{h+1} \Big) - \Big( \sum_{k=1}^{K} \alpha_k^K \underline{P}_{i,h}^{\widehat{\pi}^k, \overline{V}} \overline{V}'_{h+1} \Big) \circ \Big( \sum_{k=1}^{K} \alpha_k^K \underline{P}_{i,h}^{\widehat{\pi}^k, \overline{V}} \overline{V}'_{h+1} \Big),$$

where (i) follows from the fact that $\mathrm{Var}_{\sum_{k=1}^{K} \alpha_k^K \underline{P}_{i,h}^{\widehat{\pi}^k, \overline{V}}}(V - b\mathbf{1}) = \mathrm{Var}_{\sum_{k=1}^{K} \alpha_k^K \underline{P}_{i,h}^{\widehat{\pi}^k, \overline{V}}}(V)$ for any value vector $V \in \mathbb{R}^S$ and scalar $b$, Additionally according to (140), we have

$$\mathrm{Var}_{\sum_{k=1}^{K} \alpha_k^K \underline{P}_{i,h}^{\widehat{\pi}^k, \overline{V}}} \Big( \mathbb{E}_{\pi \sim \zeta} \Big[ \overline{V}_{i,h+1}^{\pi} \Big] \Big)$$

$$= \sum_{k=1}^{K} \alpha_k^K \underline{P}_{i,h}^{\widehat{\pi}^k, \overline{V}} \Big( \overline{V}'_{h+1} \circ \overline{V}'_{h+1} \Big)$$

$$- \Big( \overline{V}'_h - \overline{r}_{i,h}^{\min} - \Big( \sum_{k=1}^{K} \alpha_k^K \widehat{\underline{P}}_{i,h}^{\widehat{\pi}^k, \overline{V}} - \sum_{k=1}^{K} \alpha_k^K \underline{P}_{i,h}^{\widehat{\pi}^k, \overline{V}} \Big) \mathbb{E}_{\pi \sim \zeta} \Big[ \overline{V}_{i,h+1}^{\pi} \Big] \Big)^{\circ 2}$$

$$= -\overline{V}'_h \circ \overline{V}'_h + 2\overline{V}'_h \circ \Big( \overline{r}_{i,h}^{\min} + \Big( \sum_{k=1}^{K} \alpha_k^K \widehat{\underline{P}}_{i,h}^{\widehat{\pi}^k, \overline{V}} - \sum_{k=1}^{K} \alpha_k^K \underline{P}_{i,h}^{\widehat{\pi}^k, \overline{V}} \Big) \mathbb{E}_{\pi \sim \zeta} \Big[ \overline{V}_{i,h+1}^{\pi} \Big] \Big)$$

$$- \Big( \overline{r}_{i,h}^{\min} + \Big( \sum_{k=1}^{K} \alpha_k^K \widehat{\underline{P}}_{i,h}^{\widehat{\pi}^k, \overline{V}} - \sum_{k=1}^{K} \alpha_k^K \underline{P}_{i,h}^{\widehat{\pi}^k, \overline{V}} \Big) \mathbb{E}_{\pi \sim \zeta} \Big[ \overline{V}_{i,h+1}^{\pi} \Big] \Big)^{\circ 2}$$

$$+ \sum_{k=1}^{K} \alpha_k^K \underline{P}_{i,h}^{\widehat{\pi}^k, \overline{V}} \Big( \overline{V}'_{h+1} \circ \overline{V}'_{h+1} \Big)$$

Furthermore, we have

$$\mathrm{Var}_{\sum_{k=1}^{K} \alpha_k^K \underline{P}_{i,h}^{\widehat{\pi}^k, \overline{V}}} \Big( \mathbb{E}_{\pi \sim \zeta} \Big[ \overline{V}_{i,h+1}^{\pi} \Big] \Big)$$

$$= \sum_{k=1}^{K} \alpha_k^K \underline{P}_{i,h}^{\widehat{\pi}^k, \overline{V}} \Big( \overline{V}'_{h+1} \circ \overline{V}'_{h+1} \Big)$$

$$+ 2\overline{V}'_h \circ \Big( \overline{r}_{i,h}^{\min} + \Big( \sum_{k=1}^{K} \alpha_k^K \widehat{\underline{P}}_{i,h}^{\widehat{\pi}^k, \overline{V}} - \sum_{k=1}^{K} \alpha_k^K \underline{P}_{i,h}^{\widehat{\pi}^k, \overline{V}} \Big) \mathbb{E}_{\pi \sim \zeta} \Big[ \overline{V}_{i,h+1}^{\pi} \Big] \Big)$$

$$- \Big( \overline{r}_{i,h}^{\min} + \Big( \sum_{k=1}^{K} \alpha_k^K \widehat{\underline{P}}_{i,h}^{\widehat{\pi}^k, \overline{V}} - \sum_{k=1}^{K} \alpha_k^K \underline{P}_{i,h}^{\widehat{\pi}^k, \overline{V}} \Big) \mathbb{E}_{\pi \sim \zeta} \Big[ \overline{V}_{i,h+1}^{\pi} \Big] \Big)^{\circ 2} - \overline{V}'_h \circ \overline{V}'_h$$

$$\overset{(i)}{\leq} \sum_{k=1}^{K} \alpha_k^K \underline{P}_{i,h}^{\widehat{\pi}^k, \overline{V}} \Big( \overline{V}'_{h+1} \circ \overline{V}'_{h+1} \Big) - \overline{V}'_h \circ \overline{V}'_h$$

$$+ 2\|\overline{V}'_h\|_\infty \Big( 1 + \Big| \Big( \sum_{k=1}^{K} \alpha_k^K \widehat{\underline{P}}_{i,h}^{\widehat{\pi}^k, \overline{V}} - \sum_{k=1}^{K} \alpha_k^K \underline{P}_{i,h}^{\widehat{\pi}^k, \overline{V}} \Big) \mathbb{E}_{\pi \sim \zeta} \Big[ \overline{V}_{i,h+1}^{\pi} \Big] \Big| \Big) \quad (141)$$

$$
\leq \sum_{k=1}^{K} \alpha_k^K \underline{P}_{i,h}^{\widehat{\pi}^k, \overline{V}} \left( \overline{V}'_{h+1} \circ \overline{V}'_{h+1} \right) - \overline{V}'_h \circ \overline{V}'_h + 2\|\overline{V}'_h\|_\infty 1
$$

$$
+ 6\|V'_h\|_\infty \sqrt{\frac{H^2 \log\left(\frac{18S \sum_{i=1}^n A_i nHNK}{\delta}\right)}{N}} 1, \tag{142}
$$

holds with probability at least $1 - \delta$, where (i) arises from $\underline{r}_{i,h}^{\min} \leq \sum_{k=1}^{K} \alpha_k^K \mathbb{E}_{a_i \sim \widehat{\pi}_{i,h}^k} r_{i,h}^k(\cdot, a_i) \leq 1$ since $V_{h+1}^{\min} - V_h^{\min} \leq 0$ by definition, and the last inequality holds by Lemma 10. Finally, combining (142) and the definition of $d_h^j$ in (92), the term of interest can be controlled as

$$
\sum_{j=h}^{H} \left\langle d_h^j, \mathsf{Var}_{\sum_{k=1}^K \alpha_k^K \underline{P}_{i,j}^{\widehat{\pi}^k, \overline{V}}} \left( \mathbb{E}_{\pi \sim \zeta} \left[ \overline{V}_{i,j+1}^\pi \right] \right) \right\rangle
$$

$$
= \sum_{j=h}^{H} (d_h^j)^\top \left( \sum_{k=1}^{K} \alpha_k^K \underline{P}_{i,j}^{\widehat{\pi}^k, \overline{V}} \left( \overline{V}'_{j+1} \circ \overline{V}'_{j+1} \right) - \overline{V}'_j \circ \overline{V}'_j + 2\|\overline{V}'_j\|_\infty 1 \right)
$$

$$
+ \sum_{j=h}^{H} (d_h^j)^\top \left( 6\|\overline{V}'_j\|_\infty \sqrt{\frac{H^2 \log\left(\frac{18S \sum_{i=1}^n A_i nHNK}{\delta}\right)}{N}} 1 \right)
$$

$$
\overset{(i)}{\leq} \sum_{j=h}^{H} \left[ (d_h^j)^\top \left( \sum_{k=1}^{K} \alpha_k^K \underline{P}_{i,j}^{\widehat{\pi}^k, \overline{V}} \left( \overline{V}'_{j+1} \circ \overline{V}'_{j+1} \right) - \overline{V}'_j \circ \overline{V}'_j \right) \right] + 2H\|\overline{V}'_h\|_\infty
$$

$$
+ 6H^2 \|\overline{V}'_h\|_\infty \sqrt{\frac{\log\left(\frac{18S \sum_{i=1}^n A_i nHNK}{\delta}\right)}{N}}
$$

where (i) holds by the fact $\|\overline{V}'_h\|_\infty \geq \|\overline{V}'_{h+1}\|_\infty \geq \cdots \geq \|\overline{V}'_H\|_\infty$. With further basic calculus, we can finally obtain that

$$
\sum_{j=h}^{H} \left\langle d_h^j, \mathsf{Var}_{\sum_{k=1}^K \alpha_k^K \underline{P}_{i,j}^{\widehat{\pi}^k, \overline{V}}} \left( \mathbb{E}_{\pi \sim \zeta} \left[ \overline{V}_{i,j+1}^\pi \right] \right) \right\rangle
$$

$$
= \sum_{j=h}^{H} \left[ (d_h^{j+1})^\top \left( \overline{V}'_{j+1} \circ \overline{V}'_{j+1} \right) - (d_h^j)^\top \left( \overline{V}'_j \circ \overline{V}'_j \right) \right] + 2H\|\overline{V}'_h\|_\infty
$$

$$
+ 6H^2 \|\overline{V}'_h\|_\infty \sqrt{\frac{\log\left(\frac{18S \sum_{i=1}^n A_i nHNK}{\delta}\right)}{N}}
$$

$$
\leq \left\| d_h^{H+1} \right\|_1 \left\| \overline{V}'_{H+1} \circ \overline{V}'_{H+1} \right\|_\infty + 2H\|\overline{V}'_h\|_\infty + 6H^2 \|\overline{V}'_h\|_\infty \sqrt{\frac{\log\left(\frac{18S \sum_{i=1}^n A_i nHNK}{\delta}\right)}{N}}
$$

$$
\leq 3H\|\overline{V}'_h\|_\infty + 6H^2 \|\overline{V}'_h\|_\infty \sqrt{\frac{\log\left(\frac{18S \sum_{i=1}^n A_i nHNK}{\delta}\right)}{N}}
$$

$$
= 3H\|\overline{V}'_h\|_\infty \left( 1 + 2H \sqrt{\frac{\log\left(\frac{18S \sum_{i=1}^n A_i nHNK}{\delta}\right)}{N}} \right),
$$

### D.4.10 Proof of Lemma 16

Analogous to Appendix D.4.9, we introduce some auxiliary values and reward functions to control

$$\sum_{j=h}^{H} \left\langle w_h^j, \mathsf{Var}_{\sum_{k=1}^{K} \alpha_k^K \underline{P}_{i,j}^{\widehat{\pi}^k, V}} \left( \mathbb{E}_{\pi \sim \zeta} \left[ V_{i,j+1}^\pi \right] \right) \right\rangle$$

as below: for any time step $h$ and the $i$-th agent

**Definition 5.** *For any time step $h \in [H]$ and the $i$-th agent, we denote $V_h^{\min} := \min_{s \in \mathcal{S}} \mathbb{E}_{\pi \sim \zeta} \left[ V_{i,h}^\pi(s) \right]$ as the minimum value of all the entries in vector $\mathbb{E}_{\pi \sim \zeta} \left[ V_{i,h}^\pi \right]$. We further define $V_h' := C_{i,h}^\pi - V_h^{\min} \mathbf{1}$ as the truncated value function. Eventually for reward function, we define $r_{i,h}^{\min} = \sum_{k=1}^{K} \alpha_k^K \mathbb{E}_{a_i \sim \pi_{i,h}^k} r_{i,h}^{\widehat{\pi}_{-i}^k}(\cdot, a_i) + \left( V_{h+1}^{\min} - V_h^{\min} \right) \mathbf{1}$ as the truncated reward function..*

Then applying the robust Bellman's consistency equation in (26) gives

$$V_h' = \mathbb{E}_{\pi \sim \zeta} \left[ V_{i,h}^{\pi, \sigma_i} \right] - V_h^{\min} \mathbf{1} = \sum_{k=1}^{K} \alpha_k^K r_{i,h}^{\widehat{\pi}^k} + \sum_{k=1}^{K} \alpha_k^K \underline{P}_{i,h}^{\widehat{\pi}^k, V} \mathbb{E}_{\pi \sim \zeta} \left[ V_{i,h+1}^{\pi, \sigma_i} \right] - V_h^{\min} \mathbf{1}$$

$$= \sum_{k=1}^{K} \alpha_k^K r_{i,h}^{\widehat{\pi}^k} + \left( V_{h+1}^{\min} - V_h^{\min} \right) \mathbf{1} + \sum_{k=1}^{K} \alpha_k^K \underline{P}_{i,h}^{\widehat{\pi}^k, V} V_{h+1}' \quad (143)$$

$$= r_{i,h}^{\min} + \sum_{k=1}^{K} \alpha_k^K \underline{P}_{i,h}^{\widehat{\pi}^k, V} V_{h+1}'. \quad (144)$$

The above fact leads to

$$\mathsf{Var}_{\sum_{k=1}^{K} \alpha_k^K \underline{P}_{i,h}^{\widehat{\pi}^k, V}} \left( \mathbb{E}_{\pi \sim \zeta} \left[ V_{i,h+1}^{\pi, \sigma_i} \right] \right)$$

$$\overset{(i)}{=} \mathsf{Var}_{\sum_{k=1}^{K} \alpha_k^K \underline{P}_{i,h}^{\widehat{\pi}^k, V}} \left( V_{h+1}' \right)$$

$$= \sum_{k=1}^{K} \alpha_k^K \underline{P}_{i,h}^{\widehat{\pi}^k, V} \left( V_{h+1}' \circ V_{h+1}' \right) - \left( \sum_{k=1}^{K} \alpha_k^K \underline{P}_{i,h}^{\widehat{\pi}^k, V} V_{h+1}' \right) \circ \left( \sum_{k=1}^{K} \alpha_k^K \underline{P}_{i,h}^{\widehat{\pi}^k, V} V_{h+1}' \right)$$

$$\overset{(ii)}{=} \sum_{k=1}^{K} \alpha_k^K \underline{P}_{i,h}^{\widehat{\pi}^k, V} \left( V_{h+1}' \circ V_{h+1}' \right) - \left( V_h' - r_{i,h}^{\min} \right)^{\circ 2}$$

$$= \sum_{k=1}^{K} \alpha_k^K \underline{P}_{i,h}^{\widehat{\pi}^k, V} \left( V_{h+1}' \circ V_{h+1}' \right) - V_h' \circ V_h' + 2 V_h' \circ r_{i,h}^{\min} - r_{i,h}^{\min} \circ r_{i,h}^{\min}$$

$$\leq \sum_{k=1}^{K} \alpha_k^K \underline{P}_{i,h}^{\widehat{\pi}^k, V} \left( V_{h+1}' \circ V_{h+1}' \right) - V_h' \circ V_h' + 2 \| V_h' \|_\infty \mathbf{1}, \quad (145)$$

where (i) follows from the fact that $\mathsf{Var}_{\sum_{k=1}^{K} \alpha_k^K \underline{P}_{i,h}^{\widehat{\pi}^k, V}}(V - b\mathbf{1}) = \mathsf{Var}_{\sum_{k=1}^{K} \alpha_k^K \underline{P}_{i,h}^{\widehat{\pi}^k, V}}(V)$ for any value vector $V \in \mathbb{R}^S$ and scalar $b$, (ii) holds by (144) and (26), and the last inequality arises from $r_{i,h}^{\min} \leq r_{i,h}^\pi \leq 1$ since $V_{h+1}^{\min} - V_h^{\min} \leq 0$ by definition.

Consequently, combining (145) and the definition of $w_h^j$ in (102), we arrive at

$$\sum_{j=h}^{H} \left\langle w_h^j, \mathsf{Var}_{\sum_{k=1}^{K} \alpha_k^K \underline{P}_{i,h}^{\widehat{\pi}^k, V}} \left( \mathbb{E}_{\pi \sim \zeta} \left[ V_{i,j+1}^{\pi, \sigma_i} \right] \right) \right\rangle$$

$$\leq \sum_{j=h}^{H} (w_h^j)^\top \left( \sum_{k=1}^{K} \alpha_k^K \underline{P}_{i,j}^{\widehat{\pi}^k, V} \left( V_{j+1}' \circ V_{j+1}' \right) - V_j' \circ V_j' + 2 \| V_h' \|_\infty \mathbf{1} \right)$$

$$\overset{(i)}{\leq} \sum_{j=h}^{H} \left[ (w_h^j)^\top \left( \sum_{k=1}^{K} \alpha_k^K \underline{P}_{i,j}^{\widehat{\pi}^k, V} \left( V_{j+1}' \circ V_{j+1}' \right) - V_j' \circ V_j' \right) \right] + 2H \| V_h' \|_\infty$$

$$= \sum_{j=h}^{H} \left[ (w_h^{j+1})^\top \left( V'_{j+1} \circ V'_{j+1} \right) - (w_h^j)^\top \left( V'_j \circ V'_j \right) \right] + 2H\|V'_h\|_\infty$$

$$\leq \|w_h^{H+1}\|_1 \left\| V'_{H+1} \circ V'_{H+1} \right\|_\infty + 2H\|V'_h\|_\infty$$

$$\leq 3H\|V'_h\|_\infty, \tag{146}$$

where (i) and the last inequality hold by the fact $\|V'_h\|_\infty \geq \|V'_{h+1}\|_\infty \geq \cdots \geq \|V'_H\|_\infty$ and basic calculus.

