# OpenReview forum: "Breaking the Curse of Multiagency in Robust Multi-Agent Reinforcement Learning"
_ICLR.cc/2025/Conference — Submitted to ICLR 2025_

### Official Review · Reviewer_DjTn · 2024-10-23

**Soundness:** 2
**Presentation:** 2
**Contribution:** 2
**Rating:** 5
**Confidence:** 3

**Summary:**

This paper studies robust reinforcement learning in the multi-agent setting. The authors propose a new uncertainty set called "fictitious uncertainty set", where depends on the joint policy of agents. Based on the new notion, they establish the existence of robust NE and CCE. After that, they propose the Robust-Q-FTRL algorithm, and under the generative model assumption, they establish sample complexity only scaling with the number of actions of all agents.

**Strengths:**

The robust multi-agent learning setting is interesting and reasonable to me.

Although under a different setting compared with previous works considering rectangular uncertainty set, the sample complexity results "break the curse of multi-agency".

**Weaknesses:**

1. **About paper writing**: I think there are several parts of the paper writing should be improved to avoid confusion.

* If I understand correctly, starting from Section 3.2, the definition of value functions are associated with the fictitious RMGs, instead of the rectangular uncertainty sets. In another word, one should interpret $V^{\pi,\sigma_i}_{i,h}$ through the definition in Eq.(26), instead of Eq. (5), which are different in the uncertainty set w.r.t. the inf operator.

    However, I did not find a declaration of this abuse of notation (and personally, I would recommend use different notations to avoid confusion). This is crucial because it makes significant difference in how to interpret the results. Under the definition of Eq.(26), the uncertainty set varies for different policies $\pi$, rather than that one first fix the uncertainty set by an arbitrary reference policy $\pi$ and then use that to define the NE/CCE.

* The "technical insights" paragraph in Section 4.3 does not explain clearly how the proposed algorithms avoid the curse of multi-agency. This is crucial to evaluate the technical novelty of this work (also see my second point in the following).

2. **The fundamental reason for avoiding curse of multi-agency is not clear to me**.

* Comparing with previous works, this paper consider a different uncertainty set, it is not clear to me whether it is the key variation making the problem more tractable than before. Especially, by definition, the new uncertainty set only quantifies the uncertainty of the (weighted) marginal transition function. If this is the essential reason, claiming avoiding curse of multi-agency as advantage over previous work would be unfair.

* Besides, this paper relies on generative model assumption, I'm also curious whether it is a common assumption in previous literature?

3. **The proposed uncertainty set may not be sufficient to quantify the uncertainty**. I'm curious what would be the benefits of considering new uncertainty set, and is it reasonable to consider such set in practice? For example, under what scenarios (what kind of model difference between simulator and practice), the propose uncertainty set would be useful.

Besides, technically speaking, the new uncertainty set may be insufficient to capture the sim-to-real gap. Note that the ratio $\pi_h(a_i,a_{-i}|s) / \pi_{i,h}(a_i|s)$ could varies a lot for different $a_{-i}$, given that this paper considers correlated policies. Consider the scenario where the sim-to-real gaps of $P^0_{h,s,\textbf{a}}$ are very large for some $(s,\textbf{a})$, while, coincidently, the learned CCE only has low ratio $\pi_h(a_i,a_{-i}|s) / \pi_{i,h}(a_i|s)$ on them. In that case, fictitious uncertainty set is not sufficient to quantify the uncertainty.

**Questions:**

1. Is the proposed Algorithm 1 overall computationally tractable? I found a related discussion in Sec. 4.2, but it is about the Q value estimation.

---

> ### Author Response · Authors · 2024-11-21
> **Response to Reviewer DjTn: Part One**
>
> We sincerely thank Reviewer DjTn for their valuable feedback on our work. We now address your concerns and questions together below.
>
>
> ### Q1. Clarification on the Definition of Value Function (a mismatch between Eq.(26) and Eq. (5))
>
> We sincerely thank the reviewer for these constructive suggestions for clarification. In response, we have thoroughly revised Section 2.2 in the updated version. The two equations are now labeled as Eq. (23) and Eq. (3) in the new version and are consistent.
>
> ### Q2. Is Generative Model a Common Assumption in the Literature?
>
> **Generative model is a widely-used fundamental setting in RL literature, and even this setting is heavily understudied.** Sampling through a generative model is a fundamental setting with a long line of research for decades  in RL ranging from all kinds of single-agent RL and multi-agent RL problems [1-3]. The widely studied data collection settings can be categoried into  "generative model" settings, "online," and "offline". Generative model setting plays an essential role in shaping the theoretical foundation of RL and are critically needed before addressing more complex or practical sampling settings (such as online).
>
> However, even the fundamental generative model setting for robust MARL is understudeid and open with only a few prior works [4-5]. This motivates us to take a step in this open question, where our theoretical findings are potential to be extended to more settings such as online, offline, or function approximation robust MARL.
>
> > [1] Shi, L., Li, G., Wei, Y., Chen, Y., Geist, M., & Chi, Y. (2024). The curious price of distributional robustness in reinforcement learning with a generative model. Advances in Neural Information Processing Systems, 36.\
> [2] Clavier, Pierre, Erwan Le Pennec, and Matthieu Geist. "Towards minimax optimality of model-based robust reinforcement learning." arXiv preprint arXiv:2302.05372 (2023).\
> [3] Kearns, M., & Singh, S. (1998). Finite-sample convergence rates for Q-learning and indirect algorithms. Advances in neural information processing systems, 11.\
> [4] Jose Blanchet, Miao Lu, Tong Zhang, and Han Zhong. Double pessimism is provably efficient
> for distributionally robust offline reinforcement learning: Generic algorithm and robust partial
> coverage. Advances in Neural Information Processing Systems, 36, 2024\
> [5] Laixi Shi, Eric Mazumdar, Yuejie Chi, and Adam Wierman. Sample-efficient robust multi-agent reinforcement learning in the face of environmental uncertainty. In Forty-first International Conference on Machine Learning, 2024
>
>
> ### Q3. Reason and Technical Contributions for Breaking the Curse of Multi-Agency
> Thanks to the reviewer for raising this key question.
> * Our work circumvents the curse of multiagency **by two designs: the introduction of a new class of fictitious RMGs, together with resorting to the algorithm design** --- a tailored adaptive sampling and online learning procedure. Lacking either one of these two won't lead to the provable algorithm without the curse of multiagency. Combining these two techniques provides a fresh perspective on learning RMGs.
>
>     Specifically, we propose a new class of games with fictitious uncertainty set, whose practical meanings and advantages of game modeling compared to previous robust MGs can be **refer to the General response**. Armed with this new class of games, which is not only more realistic in practice but also a formulation with promising algorithms with sample efficiency. But achieving this promising solution is also challenging, where we are inspired by adversarial online learning that is widely used in breaking the curse in standard MGs and tailored to the robust MGs. This requires us to design tailored sampling ways to overcome the additional statistical difficulties induced by the non-linear of agents' robust value function, rather than the linear value function in standard MGs w.r.t. the transition kernel.
>
> * **We work on a new class of games (realistic and tractable): no fair comparisons due to non-identical problems, so we don't compare sample complexity results**  Promoting robustness in Markov games (MARL) is an open area starting from the game formulation --- how to construct the uncertainty set so that the induced games are considering realistic uncertainty and also well-posed to enable tractable learning. Previous state-of-the-art algorithms for games with $(s,a)$-rectangular set can't break the curse of multi-agency of the sample complexity --- require samples exponentially increasing as the number of agents increases. Leveraging the features of the new proposed games, we propose an algorithm (Robust Q-FTRL) and demonstrate its sample complexity that breaks the curse of multi-agency for this new class of games. It is the first one to break the curse in the context of general robust Markov games, regardless of the chosen uncertainty set.

---

> > ### Author Response · Authors · 2024-11-21
> > **Response to Reviewer DjTn: Part Two**
> >
> > ### Q4. Insights of the prooposed uncertainty sets (games), the practical meaning, and the advantages compared to previous uncertainty sets
> > We appreciate the reviewer for raising this important question. So we provide a very detailed answer in the **General response** for all the reviewers, please refer to it.
> >
> > ### Q5. Will the proposed new fictitious uncertainty set be insufficient to capture the sim-to-real gap? (Consider the scenario where the sim-to-real gaps of  $P_{h,s,a}$  are very large for some $(s, a)$, while, coincidentally, the learned CCE only has low ratio $\frac{\pi_h(a_i,a_{-i}|s)}{\pi_{i,h}(a_i | s)}$. In that case, the fictitious uncertainty set is not sufficient to quantify the uncertainty.)
> >
> > This is a very important and interesting case, which we also used in the response to other reviewers. We want to thank the reviewer again for this constructive comments. This is actually a representative case that shows **the advantages of our new fictitious uncertainty set --- can quantify the same levels of sim-to-real gaps without being too conservative.**
> > * **Technically speaking, when using the same uncertainty level, our fictitious uncertainty set will be a larger uncertainty set than the previous $(s,a)$-rectangular uncertainty set.** Specifically, let's see the final objective associated with these two sets.
> >     *  For $(s,\textbf{a})$-rectangular set: the final objective with level $\sigma_1$ will be $$f(\sigma_1) = \mathbb{E}\_{a_i\sim \pi_{i,h}}[\mathbb{E}\_{a_{-i}\sim \pi_{-i,h}}[ \inf\_{P\in U^{\sigma_1}(P^0_{h,s,a})} PV ]]$$
> >     *  For our fictitious uncertainty set: the final objective with level $\sigma_2$ will be
> > $$ g(\sigma_2) = \mathbb{E}\_{a_i\sim \pi_{i,h}}[\inf\_{P\in (U^{\sigma_2}(\mathbb{E}\_{a_{-i} \sim \pi_{-i,h}}[P^0_{h,s,a}]))} PV ]$$
> >
> >     It is easily verified that $g(\sigma_1) \leq f(\sigma_1)$ due to Jensen's inequality. It implies that our fictitious uncertainty set is a larger uncertainty set than the previous $(s,a)$-rectangular uncertainty set, so that the worst-case performance is smaller (g(\sigma_1)). But note that we don't need to choose the same uncertainty level to quantify the same uncertainty scenarios
> >
> >
> > * The Fictitious uncertainty set can **quantify reasonable uncertainty of the environment P without being overconservative; the previous $(s,\textbf{a})$-rectangular set will be too overconservative.** See a representative case that for some joint action $a^*$, the sim-to-real gap (uncertainty) on $P_{h,s,a^*}$ is large, while for other actions $a$, the uncertianty of $P_{h,s,a}$ is very small.
> >     * **The fictitious uncertainty set can quantify the uncertainty of P without resulting in an overly large uncertainty set.** This is due to exactly what the reviewer mentioned --- a two rounds process to construct the uncertainty set. Before putting the second step (an infimum operator), we do a policy-based adjustment to the uncertainty set reference --- the expected transition kernel associated with the equilibrium policy. This finally leads to a heterogeneous uncertainty set over all actions, which tightly characterizes the consequences on the agent's objective induced by the uncertainty of environment P, but without being overwhelmed with unnecessary concerns. Heterogeneous to actions is great and should be determined by agents' policy, since the policy may not even visit some bad action $a^*$ with large uncertainty, so why should we consider this large uncertainty so much.
> >     * **The previous $(s,\textbf{a})$-rectangular set will be too overconservative** since all actions $a$ use the homogenuous uncertainty level $\sigma_1$. In order to quantify the large uncertainty of P in $a^*$, $(s,\textbf{a})$-rectangularity directly quantifies the large gaps in all actions, without considering other actions (except $a^*$) only have small uncertainty.
> >
> >
> >
> >
> > * **Explaination of the reviewer's confusing about the low ratio.** WLOG, we consider product policy $\pi$. Note that for those two uncertainty sets, such low ratio $\frac{\pi_h(a_i,a_{-i}|s)}{\pi_{i,h}(a_i | s)} = \pi_{-i,h}(a_i)$ associated with the equilibrium $\pi$ will all appear in the overall "uncertainty" of the objective functions (f,g). The difference is just within (ours) the uncertainty set or outside (the $(s,a)$) one. So it is not a good signal to see which uncertainty set capture more uncertainty.

---

> > > ### Author Response · Authors · 2024-11-21
> > > **Response to Reviewer DjTn: Part Three**
> > >
> > > ### Q6. Is Algorithm 1 computationally tractable?
> > >
> > > Yes, our proposed Algorithm 1 is computationally tractable, the overall computation is very similar to many algorithms in standard Markov games [1]. We will explain the computational tractability of each step of the algorithm in detail as follows:
> > >
> > >   Only three steps in our algorithm require non-trivial computation: line 7, line 8, and line 10. First, we can directly observe that the policy update (line 8) and the update of the optimistic value function $\widehat{V}_{i,h}$ can be computed with elementary operations. Therefore, we only need to consider the computational tractability of the estimation of the robust $Q$-function (Eq. (15)).
> > >
> > >   We note that $P\_{i,h}^k \big[\widehat{V}\_{i,h+1}\big]\_{\alpha} - \sigma_i \big(\alpha - \min\_{s'}\big[\widehat{V}\_{i,h+1}\big]\_{\alpha}(s') \big)$ is a piecewise linear function of $\alpha$. Thus, we have the following:
> > >   $$
> > >   \arg\max\_{\alpha \in [\min\_s \widehat{V}\_{i,h+1}(s), \max_s \widehat{V}\_{i,h+1}(s)]} P_{i,h}^k \big[\widehat{V}\_{i,h+1}\big]\_{\alpha} - \sigma_i \big(\alpha - \min\_{s'}\big[\widehat{V}\_{i,h+1}\big]\_{\alpha}(s') \big) \in \{\widehat{V}\_{i,h+1}(s)\}\_{s \in \mathcal{S}}.
> > >   $$
> > >   Since $\mathcal{S}$ is a finite set, we can compute the estimation of the $Q$-function with at most $\mathcal{O}(\mathcal{S}\log(S))$ computational complexity [2]. Thus, the entire algorithm is computationally tractable.
> > >
> > > > [1] Li, Gen, et al. "Minimax-optimal multi-agent RL in Markov games with a generative model." Advances in Neural Information Processing Systems 35 (2022): 15353-15367.
> > > > [2] Iyengar, Garud N. "Robust dynamic programming." Mathematics of Operations Research 30.2 (2005): 257-280.

---

> ### Comment · Reviewer_DjTn · 2024-11-25
>
> Thanks for the detailed responses. Most of my questions are addressed. However, I'm still concerning about the advantage of the proposed fictitious uncertainty set. By definition, the fictitious uncertainty sets can be much more conservative than rectangular uncertainty set on those (s,a) with low policy value (because the policy value is the weight term). More concretely, for a fixed threshold $\sigma_i$, suppose the policy value on some (s,a) is very small, then the transition function in the uncertainty set can varies significantly on that (s,a), which results in much more pessimistic uncertainty quantification than rectangular sets.
>
> Although sometimes the proposed fictitious uncertainty set may indeed provide much better and sufficient uncertainty quantification, it is not clear to me when it happens, and more importantly, can we know it is better in advance without knowledge the equilibrium policy or other information.
>
> Therefore, I tend to keep my score.

---

> > ### Author Response · Authors · 2024-11-26
> > **Response to Reviewer DjTn**
> >
> > We sincerely appreciate the reviewer's additional questions and active engagement in the discussion. We will address the concerns raised and warmly welcome further discussions.
> >
> > * **The reviewer's overconservative intuition fits the previous $(s,a)$-rectangular set but not our fictitious uncertainty set.**
> >
> > To clarify the previous claim and the advantages of our fictitious uncertainty set clearly, we will use a concrete example of two-player game as below:
> >
> > **Without loss of generality, we fix a state and ignore its corresponding notations. The 1st player only have one action $a_1$, and the 2nd player has two possible actions $a_2, \widetilde{a}\_2$. The second player's policy as $\pi_2(a_2) = \pi_2(\widetilde{a}\_2) = 0.5$. Then, we suppose the true possible sim-to-real gaps on the joint action $(a_1, a_2)$ is zero (no uncertainty on $P_{h,a_1,a_2}$), while on the action $(a_1,\widetilde{a}\_2)$ is $0.9$ ($P_{h,a_1,\widetilde{a}\_2}$ can potentially deviate from nominal one $P^o\_{h,a_1,\widetilde{a}\_2}$ by 0.9). In addition, we denote $P^{\star}\_{h,a_1,\widetilde{a}\_2}$ as the transition that leads to the worst-case performance of the 1st agent.**
> >
> >
> > Then the goal is to design uncertainty set that can cover the potential sim-to-real gap, but not too conservative to cover unnecessray uncertain cases.
> > * **For previous $(s,a)$-rectangular set: No uncertainty level $\sigma$ can handle this example or most cases with heterogeneous uncertainties across actions.** As we provide in previous **Q5**, when there are heterogenous sim-to-real gaps happens in different joint actions (0 for $(a_1, a_2)$, 0.9 for $(a\_1,\widetilde{a\_2})$). Since $(s,a)$-rectangular set use a homogenuous uncertainty level $\sigma_1$ for all joint actions (no prior information leads to this default setting), we can never reach a reasonable uncertainty set for this task. If we set $\sigma = 0.9$, we can cover the worst-case sim-to-real gap towards $P^*_{h,a_1,\widetilde{a_2}}$ in $(a_1,\widetilde{a_2})$ but becomes overconservative for $(a_1, a_2)$; if we set $0\leq \sigma<0.9$, we can't cover the worst-case $P^*_{h,a_1,\widetilde{a_2}}$ in $(a_1,\widetilde{a_2})$. This is due to the assumption of homogeneity across all joint actions.
> > * **For fictitious uncertainty set: a promising tight uncertainty set that allows heterogeneous.** When it comes to fictitious uncertainty set, recall the uncertainty set $$ U^{\sigma_2}(\mathbb{E}\_{a_{-i} \sim \pi_{-i,h}}[P^0\_{h,s,a}]) =U^{\sigma_2}[  0.5 P^o\_{h,a_1,\widetilde{a}\_2} + 0.5 P^o\_{h,a_1,a_2}].$$
> > Note that $0.5 P^*_{h,a_1,\widetilde{a_2}} + 0.5 P^o_{h,a_1,a_2}$ is the case that the transition shift 0.9 at $(a_1,\widetilde{a_2})$ but don't shift at $(a_1,a_2)$, which is exactly the worst-case that we want to cover. And the fictitious uncertainty set can cover this case when we set $\sigma =0.45$, and $|| 0.5 P^*_{h,a_1,\widetilde{a_2}} + 0.5 P^o_{h,a_1,a_2} -  ( 0.5 P^o\_{h,a_1,\widetilde{a}\_2} + 0.5 P^o\_{h,a_1,a_2})||\_1 = 0.45$ (i.e., this case is at the boundary of the fictitious uncertainty set). In addition, if $\arg\inf_{P\in U^{\sigma_2}[  0.5 P^o\_{h,a_1,\widetilde{a}\_2} + 0.5 P^o\_{h,a_1,a_2}] }PV ] = 0.5 P^*_{h,a_1,\widetilde{a_2}} + 0.5 P^o_{h,a_1,a_2}$, then the fictitious uncertainty set with $\sigma_2 = 0.45$ precisely captures the worst-case sim-to-real gap without excessive conservatism. While this is the most favorable scenario, it demonstrates the potential of fictitious uncertainty sets in addressing many cases effectively --- benefits from allowing heterogeneous over different actions and depends on the agents' policy (a reasonable weights used for the heterogeneity). And $\sigma_2$ can be finetuned until satisfying, so there is hope to cover the uncertainty without overconservative.
> >
> >
> >
> > * **Generally, no prior information on sim-to-real gaps and policy: requiring fine-tuning of the uncertainty level $\sigma$** In most cases, specific tasks lack prior information about both sim-to-real gaps and policies. This limitation applies not only to our uncertainty set but to all uncertainty sets. As a result, practitioners often need to adjust the uncertainty level $\sigma$ to identify an appropriate uncertainty set for the task.

---

> ### Comment · Reviewer_DjTn · 2024-11-26
>
> Thanks for the reply. However, my main concern is still there: we may not know what the resulting policy to compute fictitious uncertainty set could be.
>
> Usually, we do not know the equilibrium policy in advance, and it could be a near-deterministic one. Suppose we only know $\pi(a|s) > \epsilon$ for some very small $\epsilon$. If I understand correctly, in your example provided, to ensure the true dynamic is included, for fictitious uncertainty set, we should choose $\sigma_i \approx 0.9 * (1-\epsilon)$. Then, for those state actions with policy $\pi(a|s) = O(\epsilon)$, those transition functions satisfying the fictitious uncertainty constraints can lead to variation at level $\sigma_i / \epsilon \approx 0.9/\epsilon$, which can be much more conservative comparing with rectangular uncertainty set.

---

> ### Author Response · Authors · 2024-11-26
> **Reponse to Reviewer DjTn**
>
> We sincerely appreciate the insightful discussions and hope this new clarification will be helpful. We agree with the reviewer that there is no prior information about the policies of all agents and all of our arguments obey this assumption. We sincerely thank the reviewer for **raising this new case with $\epsilon$, which further highlights the advantages of the fictitious uncertainty set.**
>
> For the raised example, if $\pi(a|s) > \epsilon$, in contrast, **we only need to choose $\sigma_2 \approx 0.9 * \epsilon$ rather than $0.9/\epsilon$**. We only need a very small uncertainty level for the fictitious uncertainty set, which is intuitively reasonable. Conditioned on the agents' policies, they are unlikely to visit an action $a$ with high uncertainty (e.g., 0.9), so a large uncertainty level is unnecessary to account for the consequences of that uncertainty. That is exactly the case when the fictitious set is more powerful than $(s,a)$-rectangular set. Additionally, for any sim-to-real gaps to be covered, a fictitious uncertainty set always requires **a strictly smaller or equal uncertainty level** compared to the $(s,a)$-rectangular uncertainty set, not only in the mentioned examples. The fictitious uncertainty set provides the potential for a better (tighter) uncertainty set without requiring prior knowledge of agents' policies. It can automatically adjust its shape based on the behaviors (policies) of other agents, offering the possibility of selecting an appropriate uncertainty level  $\sigma$ to construct a tight uncertainty set for the true sim-to-real gaps. While we are not claiming that such an uncertainty set can always be achieved, it is potentially feasible in many cases.
>
> * **More advantages of the fictitious set: realistic to predict human behaviors.**  And such fictitious uncertainty set is also typically the way that humans would consider the risk of uncertainty, observed from behavior economics (We have a more detailed discussion of this point in the **General Response**).

---

> > ### Comment · Reviewer_DjTn · 2024-11-26
> >
> > I do not think $\sigma = 0.9 * \epsilon$ is enough. If you do not know which policy it will result in and do not know which (s,a) has the largest 0.9 uncertainty, when you choose $\sigma$, you should consider the worst case, i.e. the largest uncertainty meets with the largest policy probability, which is $\approx 0.9 * (1-\epsilon)$.
> >
> > After choosing that $\sigma$, for those $(s,a)$ with low probability $\approx\epsilon$, the range for variation would be $0.9(1-\epsilon)/\epsilon$ (under the constraint that the total variation for all state-actions weighted by $\pi$ is bounded by $\sigma$).

---

> ### Author Response · Authors · 2024-11-26
> **Reponse to Reviewer DjTn**
>
> We appreciate the reviewer's comments and active discussions. I apologize for misunderstanding your question. I was thinking that the reviewer was mentioning $\pi_2(\widetilde{a}\_2)$ in the example.
>
> I want to clarify further that **no matter what is the policy, $0.9$ is always enough for the fictitious uncertainty set, no need for any dividing by $/\epsilon$ or so anytime**. Taking the example for illustration, setting $\sigma_2=0.9$, $\pi\_2(\widetilde{a}\_2) P^*_{h,a_1,\widetilde{a}\_2} + (1-\pi\_2(\widetilde{a}\_2))  P^o_{h,a_1,a_2} \in U^{\sigma_2}[  \pi\_2(\widetilde{a}\_2) P^o\_{h,a_1,\widetilde{a}\_2} + (1-\pi\_2(\widetilde{a}\_2)) P^o\_{h,a_1,a_2}].$, even if $\pi\_2(\widetilde{a}\_2)=1$, i.e., $\epsilon=0$. So we can always cover the worst-case $P^*_{h,a_1,\widetilde{a_2}} $ for $\widetilde{a}\_2$.
>
> Hope this will be helpful. And we warmly welcome any further discussions.

---

### Official Review · Reviewer_pYxc · 2024-10-29

**Soundness:** 3
**Presentation:** 2
**Contribution:** 2
**Rating:** 5
**Confidence:** 4

**Summary:**

This paper proposes a sample-efficient algorithm for robust general-sum and Markov games (RMGs), breaking the curse of dimensionality in RMGs and achieving optimal sample complexity for CCE with respect to the action space size.

**Strengths:**

This paper proposes a sample-efficient algorithm for robust general-sum and Markov games (RMGs), breaking the curse of dimensionality in RMGs and achieving optimal sample complexity for CCE with respect to the action space size.

**Weaknesses:**

The sample complexity optimality with respect to $A_i$ is achieved at the expense of an increased horizon $H$. Additionally, while the existence of a robust Nash equilibrium (NE) is proven, only the sample complexity for robust CCE is derived, which affects the completeness of the work.

**Questions:**

1. Could the authors discuss the challenges involved in deriving sample complexity bounds for robust NE within the existing framework? Additionally, proposing the derivation of robust NE sample complexity as a direction for future work would help address the question of completeness and open avenues for further research.
2. Could the authors include a concrete example or comparison showing how the policy-induced $(s, a_i)$-rectangularity condition improves sample complexity relative to the $(s, a)$-rectangularity condition used in previous work? Such a comparison would help readers understand the practical impact of this new approach.
3. Could the authors clarify why separate constraints on $N$ and $K$ are presented, rather than combining them as $N = KH$? If there are specific advantages or insights gained from keeping these constraints separate, an explanation would enhance the clarity and completeness of the analysis.
4. The paper lacks clear organization and contains numerous typos. For instance, on page 3, it refers to "Section 2.2 for details," whereas the relevant content is actually in Section 3.1. Additionally, the algorithm description does not appear until page 8. A thorough proofreading pass to correct section references and typographical errors is also recommended to enhance readability.

---

> ### Author Response · Authors · 2024-11-21
> **Response to Reviewer pYxc: Part One**
>
> We sincerely thank Reviewer pYxc for the valuable feedback on our work. We now address your questions and concerns collectively, as outlined below:
>
> ### Q1: Is the sample complexity optimality with respect to $A_i$ achieved at the expense of increased horizon $H$?
>
> Thanks for this clarification question. The answer is no since we are not supposed to compare the sample complexity results with any works. **We work on a new class of games (realistic and tractable): no fair comparisons to prior works due to non-identical problems.** Previous state-of-the-art algorithms study a heuristic class of robust MARL problems using $(s,a)$-rectangular uncertainty set and can't ensure tractable learning --- have sample complexity with exponential dependency on the action space $A_i$.
>
> This motivates this work to offer a new design of robust MGs (problems) with fictitious uncertainty sets that are both realistic and predictive of human behaviors in the real world and can be learned with tractable data size. We propose an algorithm (Robust Q-FTRL) and demonstrate its sample complexity that breaks the curse of multi-agency (linear dependency on $A_i$) for this new class of games. It is the first one to break the curse in the context of general robust Markov games, regardless of the chosen uncertainty set. Further improving the sample complexity dependency on horizon length $H$ is a very interesting future direction to consider.
>
>
> ### Q2. Can we design algorithms with sample complexity guarantees for Robust NE instead of robust CCE? What's the challenges?
> Thanks for raising this clarification question.
> * **The computation intractability of Nash in general-sum multi-agent games hinders the algorithm design interests.** Computing the Nash equilibrium (NE) in standard Markov games (MGs) is known to be PPAD-hard --- computationally intractable [1]. Consequently, when considering general-sum multi-agent standard MGs, prior works (such as [2,3,4,5]) typically focus on developing algorithms with sample complexity for Correlated Nash Equilibrium (CCE) that can be computed in polynomial time. As NE in standard MGs is a special case of the robust NE of our robust MGs, computing robust NE is also computationally intractable. So we follow the standard literature to consider robust CCE as well.
> * **Proving the existence of robust NE shows our proposed game is well-posed.** We propose a new class of games with a fictitious uncertainty set that involves more correlations between agents. There are no existing results that show that this class of games has the existence of a natural optimal solution. So we prove the existence of NE to ensure that there at least exists a natural optimal solution that ensures each agent is **independently** rationally satisfied. This means the proposed games are realistic and well-posed for individuals to solve. The existence of NE directly implies the existence of CCE.
>
>
> > [1]Constantinos Daskalakis. On the complexity of approximating a nash equilibrium. ACM Transactions on Algorithms (TALG), 9(3):23, 2013.\
> > [2]Daskalakis, C., Golowich, N., and Zhang, K. (2023). The complexity of markov equilibrium in stochastic games. In The Thirty Sixth Annual Conference on Learning Theory,\
> >[3]Jin, C., Liu, Q., Wang, Y., and Yu, T. (2021). V-learning-a simple, efficient, decentralized algorithm for multiagent RL. arXiv preprint arXiv:2110.14555\
> >[4]Cui, Q., Zhang, K., and Du, S. (2023). Breaking the curse of multiagents in a large state space: Rl in markov games with independent linear function approximation. In The Thirty Sixth Annual Conference on Learning Theory, \
> >[5]Wang, Y., Liu, Q., Bai, Y., and Jin, C. (2023). Breaking the curse of multiagency: Provably efficient decentralized multi-agent rl with function approximation. In The Thirty Sixth Annual Conference on Learning Theory,

---

> > ### Author Response · Authors · 2024-11-21
> > **Response to Reviewer pYxc: Part Two**
> >
> > ### Q3. Insights of the prooposed uncertainty sets (games), the sample complexity comparing to previous uncertainty sets, and the practical meaning of the games.
> > Thank you for raising this insightful question.
> > * **We work on a new class of games (realistic and tractable): no fair comparisons of sample complexity due to non-identical problems.** Promoting robustness in Markov games (MARL) against uncertainty is an open challenge, starting from the game formulation—specifically, how to construct uncertainty sets that make the induced games realistic and well-posed for tractable learning. Previous state-of-the-art algorithms address a heuristic class of robust MARL problems but fail to ensure learning with a feasible data size.
> >
> >     This motivates this work to offer a new design of robust MGs (problems) with fictitious uncertainty sets that are both realistic and predictive of human behaviors in the real-world and can be learned with tractable data size. Our contribution lies within the general context of robust MGs. Importantly, in this broader perspective, our proposed algorithm is the first to break the curse of multi-agency in sample complexity, regardless of the game class.
> > * **Practical significance of the proposed robust MGs with fictitious uncertainty sets (realistic and natural)**: Please refer to the first bullet point in the **general response**.
> >
> >
> > ### Q4. Explanation on the Constraints of $N$ and $K$
> > There is no particular reason for presenting separate constraints on $N$ and $K$, and we indeed provide the upper bound for $N = KH$ in our submitted paper. We present separate constraints solely to offer further details regarding the parameter selection in the algorithm. In our approach, $N$ represents the number of samples required to estimate the transition model for each episode, while $K$ denotes the number of episodes needed to learn the optimal policy. So the total number of samples is their multiplication.
> >
> >   In other words, as most RL algorithms involve two stages---"planning" (finding the optimal policy) and "exploration" (learning the environment and model parameters)---the constraint on $N$ reflects the sample complexity of the "exploration" phase, while $K$ corresponds to the sample complexity of the "planning" phase. Furthermore, we believe that presenting the constraints separately provides additional insight into the parameter selection process of the algorithm, as it is crucial for any algorithmic work to specify the constraints on all the parameters involved.
> >
> > ### Q5. Do a thorough proofreading pass
> >
> > Thank you for the careful review and pointing out the typos. We have conducted a thorough grammar and typo check and have corrected the issues in the updated version of our paper. The late appearance of the algoriothm details is due to we highlight the new class of games and its basic properties (e.g., the existence of Nash). The formulation of this new game is also one of the most important contribution of this work, providing a fresh view of game design.

---

> > ### Comment · Reviewer_pYxc · 2024-11-26
> >
> > Thank you for your response. After reading the response and comments from other reviewers, I decide to keep my score.

---

> > > ### Author Response · Authors · 2024-11-27
> > > **Response to Reviewer pYxc**
> > >
> > > We sincerely thank the reviewer for their time and thoughtful evaluation of our work. We are currently engaged in ongoing discussions with reviewer DjTn, though a consensus has not yet been reached. We kindly invite you to follow these discussions and would greatly value any further insights, comments, or questions you may wish to share.

---

### Official Review · Reviewer_BsE9 · 2024-11-03

**Soundness:** 3
**Presentation:** 3
**Contribution:** 2
**Rating:** 5
**Confidence:** 4

**Summary:**

This paper addresses the challenge of the “curse of multiagency” in robust MARL by introducing a new class of robust Markov games with a fictitious uncertainty set. This kind of uncertainty set allows any one agent to consider uncertainties arising from both the environment and the policies of other agents. Then, the authors prove the existence of robust Nash equilibrium and propose an algorithm Robust-Q-FTRL, which learns an approximate coarse correlated equilibrium with polynomial sample complexity, thus breaking the curse of multiagency in RMG for the first time. No experiment validation is provided.

**Strengths:**

Strengths:

1. The main body of this paper is well-written.
2. The proposed uncertainty set is interesting.
3. The proposed algorithm effectively reduces sample complexity, avoiding the exponential growth issue in multi-agent systems caused by the joint action space.

**Weaknesses:**

Weakness:

1. The approach assumes access to a generative model with a true nominal transition kernel for sample generation, which is infeasible in real-world applications.
2. Although the paper provides theoretical evidence for overcoming the curse of multiagency, further experimental validation is needed to determine whether the algorithm itself breaks the curse or if the new problem formulation inherently possesses this advantage, potentially enabling other algorithms to achieve similar results. In addition, additional experimental evidence is needed to demonstrate that the new uncertainty set yields more robust policies compared to the previously classic (s, a)-rectangularity uncertainty sets.
3. While reading through the proof section, I noticed typos and unclear expressions, e.g., on lines 1044, 1170, and 1175. Additionally, the proofs lack tight cohesion between sections as seen in the main body, and there are missing references to some essential equations, which makes the reading experience somewhat challenging.

**Questions:**

Questions:

1. Does the definition of the policy imply that the final solution can be an un-stationary policy?
2. Are the fictitious uncertainty sets un-stationary? That is, does each period have a distinct uncertainty set?
3. In a robust Markov game (MG) setting where the uncertainty set differs at each time step, is it valid to interchange the max operators in the definition of the optimal robust value function at each time step?
4. You assume that sampling from the true nominal transition kernel is possible. However, if this is feasible, why is robustness necessary? To address the sim-to-real gap, wouldn’t it be more appropriate to assume that sampling from the true nominal transition kernel isn’t possible?
5. Additionally, could you explain why the estimation method in Algorithm 1 differs from that in standard MG settings?
6. My understanding of the fictitious uncertainty set is that it involves a policy-based adjustment to the classic reference transition kernel (nominal transition kernel), which is then used as the basis for constructing the uncertainty set. This implies two rounds of adjustment in creating the uncertainty set, adding an extra step compared to traditional uncertainty sets. Does this make the new uncertainty set more conservative? How do you quantify the difference between the fictitious uncertainty set and a conventional uncertainty set without the initial adjustment?
7. Could you elaborate on the reason for breaking the curse of multiagency? Is it mainly due to the FTRL algorithm, or does the fictitious uncertainty set contribute to this outcome?
8. Could additional experimental results be provided to verify the convergence of the algorithm and demonstrate that the robust policy in this setting achieves better robustness compared to the classical (s, a)-rectangularity setting?
9. Line 65, Kearns & Singh, 1999 is not about solving finite-horizon multi-player general-sum Markov games, why cite this?

---

> ### Author Response · Authors · 2024-11-21
> **Reply to Reviewer BsE9: Part One**
>
> We appreciate the reviewer for their valuable suggestions and for recognizing the contributions of our proposed new game and efficient learning algorithm.
>
> ### Q1. Generative Model as the sampling mechanisum
>
> **Generative model is a widely-used fundamental setting in RL literature, and even this setting is heavily understudied.** Sampling through a generative model is a fundamental setting with a long line of research for decades  in RL ranging from all kinds of single-agent RL and multi-agent RL problems [1-3]. The widely studied data collection settings can be categoried into  "generative model" settings, "online," and "offline". Generative model setting plays an essential role in shaping the theoretical foundation of RL and are critically needed before addressing more complex or practical sampling settings (such as online).
>
> However, even the fundamental generative model setting for robust MARL is understudeid and open with only a few prior works [4-5]. This motivates us to take a step in this open question, where our theoretical findings are potential to be extended to more settings such as online, offline, or function approximation robust MARL.
>
> > [1] Shi, L., Li, G., Wei, Y., Chen, Y., Geist, M., & Chi, Y. (2024). The curious price of distributional robustness in reinforcement learning with a generative model. Advances in Neural Information Processing Systems, 36.\
> [2] Clavier, Pierre, Erwan Le Pennec, and Matthieu Geist. "Towards minimax optimality of model-based robust reinforcement learning." arXiv preprint arXiv:2302.05372 (2023).\
> [3] Kearns, M., & Singh, S. (1998). Finite-sample convergence rates for Q-learning and indirect algorithms. Advances in neural information processing systems, 11.\
> [4] Jose Blanchet, Miao Lu, Tong Zhang, and Han Zhong. Double pessimism is provably efficient
> for distributionally robust offline reinforcement learning: Generic algorithm and robust partial
> coverage. Advances in Neural Information Processing Systems, 36, 2024\
> [5] Laixi Shi, Eric Mazumdar, Yuejie Chi, and Adam Wierman. Sample-efficient robust multi-agent reinforcement learning in the face of environmental uncertainty. In Forty-first International Conference on Machine Learning, 2024\
>
> ### Q2. Does other algorithm can also break the curse of multiagency for the proposed robust MGs? Will the proposed new uncertainty set yields more robust policies compared to the previously $(s, a)$-rectangularity uncertainty set? Additional numerical examples will be useful for those questions
> Thanks for the insightful questions.
> * There **may exists other algorithms** that can break the curse of multiagency for the proposed robust MGs. Our proposed Robust-Q-FTRL serves as the first success case, which inspires more works to continue working on data-tracatable algorithms.
> * **We work on a new class of games (realistic and tractable): no fair comparisons due to non-identical robust considerations.** Basically, these two uncertainty sets consider robustness against different types of environmental uncertainty, so it is hard to find scenarios for a fair comparisons. Promoting robustness in Markov games (MARL)is an open area starting from the game formulation --- how to construct the uncertainty set so that the induced games are considering realistic uncertainty and also well-posed to enable tractable learning. Previous state-of-the-art algorithms study a heuristic class of robust MARL problems and can't ensure learning with tractable data size. This motivates this work to offer a new design of robust MGs (problems) with fictitious uncertainty sets that are both realistic and predictive of human behaviors in the real-world and can be learned with tractable data size.
> * **We focus on establishing the theoretical feasibility of breaking the curse of multi-agency in robust MARL.** Similar to existing standard pipeline of theoretical MARL research [1-2] that do not include numerical experiments, our main objective is to establish -- from the theoretical perspective -- that indeed it is possible to design a robust MG with a provably efficient algorithm that breaks the curse of multi-agency.
>
> ### Q3. Improving the presentation in the proof (While reading through the proof section, I noticed typos and unclear expressions, e.g., on lines 1044, 1170, and 1175. Additionally, the proofs lack tight cohesion between sections as seen in the main body, and there are missing references to some essential equations, which makes the reading experience somewhat challenging.)
>
>
> Thank you for your valuable feedback and suggestions. We have addressed the issue you raised and have thoroughly revised the proof. Additionally, we have highlighted the modifications made in the revised version of our paper for your convenience.

---

> > ### Author Response · Authors · 2024-11-21
> > **Reply to Reviewer BsE9: Part Two**
> >
> > ### Q4. Questions about non-stationary
> > * Does the definition of the policy imply that the final solution can be an un-stationary policy?
> > * Are the fictitious uncertainty sets un-stationary? Does each period have a distinct uncertainty set?
> >
> > Thanks for raising the clarification questions. If the authors do not misunderstand, the reviewer is asking about **the non-stationary with respect to different time step $h=1,2,\cdots, H$.**
> > * Yes, the strategy class of each agent $i$ is all non-stationary policy: $\pi\_i = \{\pi\_{i,h} :\mathcal{S}   \mapsto \Delta(\mathcal{A}\_i)  \}\_{1\leq h\leq H}$, which means at each time step $h\in 1,2,\cdots, H$, the agent can choose different policy.
> > * Yes, the uncertainty set is also non-stationary w.r.t the time step, i.e., at different $h$, we can have different uncertainty set over each $(s, \textbf{a})$ due to non-stationary of policy and nominal transitions, see the definition for details ((11）in Definition 1). But note that the uncertainty set definition is prescribed and won't change.
> >
> > ### Q5. In a robust Markov game (MG) setting where the uncertainty set differs at each time step, is it valid to interchange the max operators in the definition of the optimal robust value function at each time step?
> > Thanks to the reviewer for this important question. If the author does not misunderstand, the reviewer's question is whether the maximum and infimum operator in  the maximum of robust value function can be exchanged:
> > $$ V\_{i,h}^{\star,\pi\_{-i}, \sigma_i}(s) = \max\_{\pi'_i: \mathcal{S} \times [H] \mapsto \Delta(\mathcal{A}\_i)} \inf\_{P\in U\_{\rho}^{\sigma_i}(P^{0})} V\_{i,h}^{\pi\_i' \times \pi\_{-i},P} (s)$$
> >
> > The answer is partial: we can exchange these two operators in a bootstrapping manner by separating the partial application of the maximum operator and moving it inside the infimum operator. However, the ability to partially exchange them is not due to the uncertainty sets but rather the non-stationarity and independence of the policy at each time step. The partial exchange can be demonstrated as follows:
> >
> > $$\text{exchange} \rightarrow \max\_{\pi'\_{i,h^-}: \mathcal{S} \times [h] \mapsto \Delta(\mathcal{A}\_i)}   \inf\_{P\in U\_{\rho}^{\sigma\_i}(P^{0})} \;\; \max\_{\pi'\_{i,h^+}: \mathcal{S} \times [H]\setminus[h] \mapsto \Delta(\mathcal{A}\_i)} V\_{i,h}^{\pi_i' \times \pi_{-i},P} (s),$$
> > here we denote $\pi'\_{i,h^-} = \{\pi'\_{i,t}\}\_{t=1}^h$, and $\pi'\_{i,h^+} = \{\pi'\_{i,t}\}_{t=h+1}^H$.
> >
> > ### Q6.  Why assuming sampling from true nominal transition kernel is possible?
> > Thanks for this clarification question. Recall that robust MARL considers the problems that the testing environment may deviate from the training environment (the true nominal transition kernel), and the goal is to perform well in the testing environment (not the true nominal environment) in the worst-case. So it is reasonable to assume that we can access the training environment (true nominal transition) to sample from and collect information. However, note that we can never sample from the testing environment, which is the one that we really want to solve.
> >
> >
> > ### Q7. Why the estimation method in Algorithm 1 differs from that in standard MG settings?
> >
> > This is an insightful question that requires further discussion. A brief one: the tailored estimation method in Algorithm 1 is to overcome additional statistical estimation induced by robust MGs.
> >
> > Prior approaches for standard MG assume a linear relationship between the value function and the transition kernel, allowing statistical errors across $K$ iterations to cancel out. However, for our new robust MGs, the robust value function, due to its distributionally robust requirement, is highly nonlinear and often lacks a closed form, making it impossible to linearly aggregate statistical errors. To tackle the nonlinear challenges in RMGs, we use Algorithm 1 to increase the number of samples during each estimation and then design a new variance-style bonus term (in the main Algorithm 2) through non-trivial decomposition and control of auxiliary statistical errors caused by nonlinearity, resulting in a tight upper bound on regret during the online learning process.

---

> > > ### Author Response · Authors · 2024-11-21
> > > **Reply to Reviewer BsE9: Part Three**
> > >
> > > ### Q8. The fictitious uncertainty set involves a policy-based adjustment to the classic reference transition kernel (nominal transition kernel), which is then used as the basis for constructing the uncertainty set. This implies two rounds of adjustment in creating the uncertainty set, adding an extra step compared to traditional uncertainty sets. Does this make the new uncertainty set more conservative? How do you quantify the difference between the fictitious uncertainty set and a conventional uncertainty set without the initial adjustment?
> > >
> > >
> > > Thanks for raising this question. The reviewer's intuition is very insightful. Here, we provide a thorough comparison between our fictitious uncertainty set and the previous $(s,a)$-rectangular uncertainty set.
> > >
> > >
> > > * **Technically speaking, when using the same uncertainty level, our fictitious uncertainty set will be a larger uncertainty set than the previous $(s,a)$-rectangular uncertainty set.** Specifically, let's see the final objective associated with these two sets.
> > >     *  For $(s,\textbf{a})$-rectangular set: the final objective with level $\sigma_1$ will be $$f(\sigma_1) = \mathbb{E}\_{a_i\sim \pi_{i,h}}[\mathbb{E}\_{a_{-i}\sim \pi_{-i,h}}[ \inf\_{P\in U^{\sigma_1}(P^0\_{h,s,a})} PV ]]$$
> > >     *  For our fictitious uncertainty set: the final objective with level $\sigma_2$ will be
> > > $$ g(\sigma_2) = \mathbb{E}\_{a_i\sim \pi_{i,h}}[\inf\_{P\in (U^{\sigma_2}(\mathbb{E}\_{a_{-i} \sim \pi_{-i,h}}[P^0_{h,s,a}]))} PV ]$$
> > >
> > >     It is easily verified that $g(\sigma_1) \leq f(\sigma_1)$ due to Jensen's inequality. It implies that our fictitious uncertainty set is a larger uncertainty set than the previous $(s,a)$-rectangular uncertainty set, so that the worst-case performance is smaller (g(\sigma_1)). But note that we don't need to choose the same uncertainty level to quantify the same uncertainty scenarios
> > >
> > >
> > > * The Fictitious uncertainty set can **quantify reasonable uncertainty of the environment P without being overconservative; the previous $(s,\textbf{a})$-rectangular set will be too overconservative.** See a representative case that for some joint action $a^*$, the sim-to-real gap (uncertainty) on $P_{h,s,a^*}$ is large, while for other actions $a$, the uncertianty of $P_{h,s,a}$ is very small.
> > >     * **The fictitious uncertainty set can quantify the uncertainty of P without resulting in an overly large uncertainty set.** This is due to exactly what the reviewer mentioned --- a two rounds process to construct the uncertainty set. Before putting the second step (an infimum operator), we do a policy-based adjustment to the uncertainty set reference --- the expected transition kernel associated with the equilibrium policy. This finally leads to a heterogeneous uncertainty set over all actions, which tightly characterizes the consequences on the agent's objective induced by the uncertainty of environment P, but without being overwhelmed with unnecessary concerns. Heterogeneous to actions is great and should be determined by agents' policy, since the policy may not even visit some bad action $a^*$ with large uncertainty, so why should we consider this large uncertainty so much.
> > >     * **The previous $(s,\textbf{a})$-rectangular set will be too overconservative** since all actions $a$ use the homogenuous uncertainty level $\sigma_1$. In order to quantify the large uncertainty of P in $a^*$, $(s,\textbf{a})$-rectangularity directly quantifies the large gaps in all actions, without considering other actions (except $a^*$) only have small uncertainty.
> > >
> > >
> > > ### Q9. The technical insights of breaking the curse of multiagency. Is it mainly due to the FTRL algorithm, or does the fictitious uncertainty set contribute to this outcome?
> > >
> > > We appreciate the reviewer's interest in the technical contributions. The general answer is: that both the problem formulation and proposed algorithms play a key role in the results. Our work circumvents the curse of multiagency by the introduction of a new class of robust MGs that is more realistic in the real-world (predictive of human decision-making) than prior robust MGs, together with resorting to a tailored adaptive sampling and online learning procedure, providing a fresh perspective to learning RMGs.
> > >
> > > We have added **the practical meaning and motivation of the proposed new games** with comparisons to the prior works in the new version (Section 3.1), please also refer to the **general response Q1** for more details.
> > >
> > > ### Q10. Line 65, Kearns & Singh, 1999 is not about solving finite-horizon multi-player general-sum Markov games, why cite this?
> > > Thanks for the clarification questions. The reviewer is correct that [Kearns & Singh, 1999] is about single-agent RL. In Line 65, we are talking about the generative model sampling mechanism, so we cite this classical work [Kearns & Singh, 1999] in support of which the generative model is a common and widely-used sampling way in not only multi-agent RL, but widely in single-agent RL literature as well.

---

### Official Review · Reviewer_rZ7M · 2024-11-03

**Soundness:** 2
**Presentation:** 3
**Contribution:** 2
**Rating:** 5
**Confidence:** 3

**Summary:**

The authors propose a new multi-agent problem class, which is a new class of robust Markov Games with fictitious uncertainty sets. The authors define solution concepts – robust Nash equilibrium and robust coarse correlated equilibrium for this new class of games and also prove the existence of these. The authors then propose a novel algorithm called Robust-Q-FTRL to find robust CCE for their class of robust Markov Games and also establish its sample complexity. Using their approach in their class if problems, the authors break the curse of multi-agency in MARL.

**Strengths:**

1. The paper is well written. All concepts and claims are clearly defined and claims are backed by theoretical justifications.
2. While the authors provide a detailed theoretical analysis, I am unable to verify the correctness of their results, though I could not find any error.
3. The authors clearly state their algorithm which can be applied to practical problems.

**Weaknesses:**

1. The authors do not provide any numerical examples. While they claim that their approach is scalable, some numerical examples will be useful to understand scalability and the sample efficiency claims better. Can the authors include a small simulation study to demonstrate the scalability and sample efficiency of their approach compared to baselines, even if only on a toy problem?
2. The authors do not provide any comparison of their approach with other approaches in literature in terms of numerical computations.
3. The authors do not comment on whether the policies learned using their algorithms are strategy proof from a learning perspective, i.e., do they perform well when other agents use different learning algorithms.

**Questions:**

1. What is the $(s, \mathbf{a})$-rectangularity condition? An explicit definition for this would be useful to readers with limited background in this class of problems.  Can the authors provide a brief formal definition of the $(s,a)$-rectangularity condition in the main text, along with a sentence or two explaining its significance in robust MDPs/Markov games?
2. How are the uncertainty sets defined in this work practically meaningful? Can the authors to provide 1-2 concrete examples of how their fictitious uncertainty sets could model real-world uncertainties in multi-agent systems, and how this compares to uncertainty modeling in previous work?
3. What are the real-world scenarios which are modelled well by this class of RMGs with fictitious uncertainty sets?
4. When the authors compare with prior art in Table 1, aren’t the game models also different in the different works? Are there any trade-offs due to this?
5. Is the sample complexity improvement obtained by the authors primarily dependent on their robust Markov game definition which is different from the other works in literature?
6. Doesn’t the uncertainty set become very large and potentially intractable in problems with large state spaces?
7. Is it necessary that the uncertainty set be defined as part of the problem? In practical use cases, how does one define these sets? Specifically, can the authors discuss guidelines or heuristics for how practitioners could define appropriate uncertainty sets for real-world multi-agent problems, and whether there are ways to learn or adapt the uncertainty sets from data?
8. While I agree that the proofs are too long to be included in the main paper, but considering the fact that the primary contribution of this work is theoretical, it would be very helpful if the authors can provide short proof sketches or key proof ideas for all their claims in the main paper itself.

**Details Of Ethics Concerns:**

The ethical concerns are only limited to the concerns for any MARL algorithm: that caution must be exercised regarding potential biases and harmful outcomes when using these in real-world use cases.

---

> ### Author Response · Authors · 2024-11-21
> **Reply to Reviewer rZ7M: Part One**
>
> We sincerely thank the reviewer for the valuable suggestions and the kind recognition of our contribution and presentation.
>
> ### Q1. Additional numerical examples will be useful. There are no numerical comparison between the algorithm and other baselines in the literature.
> We appreciate the reviewer for this valuable suggestion.
> * **We work on a new class of games (realistic and tractable) so there are no fair comparisons due to non-identical problems.** Promoting robustness in Markov games (MARL) against uncertainty is an open area starting from the game formulation --- how to construct the uncertainty set so that the induced games are realistic and also well-posed to enable tractable learning. Previous state-of-the-art algorithms study a heuristic class of robust MARL problems and can't ensure learning with tractable data size. This motivates this work to offer a new design of robust MGs (problems) with fictitious uncertainty sets that are both realistic and predictive of human behaviors in the real-world and can be learned with tractable data size.
> * **We focus on establishing the theoretical feasibility of breaking the curse of multi-agency in robust MARL.** Similar to existing standard pipeline of theoretical MARL research [1-2] that do not include numerical experiments, our main objective is to establish -- from the theoretical perspective -- that indeed it is possible to design a robust MG with a provably efficient algorithm that breaks the curse of multi-agency.
> > [1] Jin, Chi, et al. "V-Learning--A Simple, Efficient, Decentralized Algorithm for Multiagent RL." arXiv preprint arXiv:2110.14555 (2021).\
> > [2] Bai, Yu, and Chi Jin. "Provable self-play algorithms for competitive reinforcement learning." International conference on machine learning. PMLR, 2020.
>
> ### Q2. Does the proposed algorithm perform well when other agents use different learning algorithms?
>
> Thank you for raising this interesting question. The short answer is that we are not certain. However, this question is not directly related to the primary goal of this work, which is focused on learning an equilibrium in games.
> * **We can decide learning algorithms for all agents.** The goal of this work is learning in a game: after the game's structure is fixed (each agent's objective), the aim is to design an overall learning process where agents update their strategies to eventually converge to an equilibrium (where each agent is rationally satisfied). Agents can run different or identical sub-algorithms (e.g., our proposed Robust Q-FTRL). Since we control the design of learning algorithms, there is no reason to use suboptimal algorithms that could hinder the training process. If some agents deviate from the designed process, convergence to a fixed point—and especially to a good equilibrium—cannot be guaranteed.
> *  **Ensuring convergence in games is challenging, which our designed algorithm can achieve** Reaching equilibrium in games is generally difficult, and convergence is not guaranteed. For example, following gradient descent in a two-player rock–paper–scissors game leads to chaos [1], without convergence, let alone a good equilibrium. A key advantage of our proposed algorithm is its ability to ensure fast convergence of the learning process.
>
> > [1] Sato, Yuzuru, Eizo Akiyama, and J. Doyne Farmer. "Chaos in learning a simple two-person game." Proceedings of the National Academy of Sciences 99.7 (2002): 4748-4751.
>
> ### Q3. Adding the definition and comments on $(s,\mathbf{a})$-rectangular uncertainty set
>
> Thanks for your valuable suggestion. We have already add the definition and comments on $(s,\mathbf{a})$-rectangular uncertainty set in the revised version of our paper. Specifically, we include the following explanation in our paper:
>
> Previous work on provably sample-efficient algorithms has focused on a different class of uncertainty sets, characterized by the $(s, \mathbf{a})$-rectangularity condition [1][2][3]. Mathematically, the $(s, \mathbf{a})$-rectangularity condition is defined as follows: for all $i \in [n]$,
>
> $$\mathcal{U}\_\rho^{\sigma_i}(P^0) := \otimes \mathcal{U}^{\sigma_i}(P\_{h,s,\mathbf{a}}^0),$$
> where
> $$\mathcal{U}^{\sigma_i}(P\_{h,s,\mathbf{a}}^0) = \\{ P_{h,s,\mathbf{a}} \in \Delta(\mathcal{S}) : \rho(P_{h,s,\mathbf{a}}, P_{h,s,\mathbf{a}}^0) \leq \sigma_i \\}.$$
>
> This condition enables each agent to independently select its own uncertainty set, which can be decomposed as a product of subsets over each state-action pair. When $n = 1$, the $(s, \mathbf{a})$-rectangularity condition reduces to the $(s, a)$-rectangularity condition in the context of robust MDPs [4]. The $(s, \mathbf{a})$-rectangularity model is widely studied in both single-agent and multi-agent robust RL and characterizes environmental uncertainty that is independent of the agent's behavior policy.

---

> > ### Author Response · Authors · 2024-11-21
> > **Reply to Reviewer rZ7M: Part Two**
> >
> > ### Q4. The practical significance of the proposed fictitious uncertainty set and its advantages over previous uncertainty sets
> >
> > We appreciate this insightful question, which has helped us enhance the clarity of our motivation and the distinctions of this work. In the revised version (Section 3.1), we have elaborated on the practical motivation behind our proposed uncertainty set.
> > * The practical meaning of the proposed fictitious uncertainty set (and the proposed robust MGs):
> >     *  **Realistic and predictive of human decisions inspired by behavioral economics**: It is meaningful to consider a realistic class of robust stochastic games predictive of human decision-making, as modern multi-agent systems often involve people, such as in social media networks and autonomous driving systems. Observations from experimental data in behavioral economics [1-3] reveal that in many games accounting for agents' randomness, people handle other players' uncertainty using a risk metric outside of their expected outcomes (i.e., $\mathbf{Risk}(\mathbb{E}\_{a_{-i} \in \pi_{-i}}[u_i(a_i, \mathbf{a_{-i}})])$), rather than taking the expectation of the risk metric over the outcome of each joint action (i.e., $\mathbb{E}\_{a_{-i} \in \pi_{-i}}[\mathbf{Risk}(u_i(a_i, \mathbf{a_{-i}}))]$). Notably, the former approach, which aligns with realistic human behavior, corresponds to our fictitious uncertainty set, whereas the latter corresponds to the traditional $(s,\mathbf{a})$-rectangular uncertainty set.
> >     *  **A natural viewpoint for an individual: reduction to single-agent robust RL**: When fixing other agents' policy $\pi_{-i}$, from the perspective of each individual agent $i$,  our proposed games with fictitious uncertainty set will degrade to the original single-agent robust RL problem with the widely used $(s,a_i)$-rectangularity condition in the literature [4]. Namely, from any agent $i$'s viewpoint, it has an "overall environment" player that can not only manipulate the environmental dynamics but also other players' policy $\pi_{-i}$. This can't be achievd by the previous $(s,\mathbf{a})$-rectangular uncertainty set.
> >
> > > [1] Friedman, Evan, and Felix Mauersberger. "Quantal response equilibrium with symmetry: Representation and applications." Proceedings of the 23rd ACM Conference on Economics and Computation. 2022.\
> > > [2] Sandomirskiy, Fedor, et al. "Narrow Framing and Risk in Games." (2024).\
> > > [3] Goeree, Jacob K., Charles A. Holt, and Thomas R. Palfrey. "Regular quantal response equilibrium." Experimental economics 8 (2005): 347-367.\
> > > [4] Garud N Iyengar. Robust dynamic programming. Mathematics of Operations Research, 30(2): 257–280, 2005.
> >
> > ### Q5. Comparisons between the results and previous works: in game model and sample complexity
> > We thank the reviewer for raising this important question, which aids in differentiating our work.
> > * **Comparison regarding the game model: this work's game model is realistic and natural.** The game model is defined by the uncertainty set. Please refer to our earlier responses (for **Q4**) for a detailed discussion on the advantages of our proposed game model.
> > * **Comparison regarding sample complexity: no direct fair comparison due to non-identical problems.** We apologize for the confusion caused by the term "comparing" in the original manuscript. We have **revised the text to avoid this term in the new version**. This work focuses on a new class of robust MGs, distinct from previous robust MGs with $(s,\mathbf{a})$-rectangular uncertainty sets. As such, a fair comparison of sample complexity is not feasible. For clarification and convenience, we provide (in Table 1) an overview of existing sample complexity results in the broader context of robust MGs, even if they target different classes of games under varying uncertainty set formulations.
> >
> >     Our contribution lies within the general context of robust MGs. Importantly, in this broader perspective, our proposed algorithm is the first to break the curse of multi-agency in sample complexity, regardless of the game class. This breakthrough is made possible through the introduction of a new class of robust MGs, combined with a tailored adaptive sampling and online learning procedure, offering a fresh perspective on learning RMGs.

---

> > > ### Author Response · Authors · 2024-11-21
> > > **Reply to Reviewer rZ7M: Part Three**
> > >
> > > ### Q6. Will the uncertainty set become very large in problems with large state spaces?
> > > Thanks for raising this question. In brief: while the overall uncertainty set will have a high dimension, it will not be overly conservative from the player's perspective.
> > >
> > > The reviewer's intuition is correct that the space of the overall uncertainty set is a product over all states, meaning its dimensions increase with the number of states. However, from the player's perspective, the effects of the entire uncertainty set can be seen as an expectation over its subsets across all states. At any time step, the player operates under a distribution over the state space, so the uncertainty is expressed as $\mathbb{E}_{s\in \mathcal{S}}[\text{Uncertainty over } s]$. Therefore, while the number of states affects the support of this distribution, it does not lead to intractably large uncertainty.
> > >
> > >
> > >
> > > ### Q7. Is it necessary that the uncertainty set be defined as part of the problem? How to define a reasonable uncertainty set in practice, any heuristic guidelines or it can be learned from data?
> > > Thank you to the reviewer for this important question.
> > > * **Distributionally robust Markov games (RMGs) inherently require an uncertainty set.** Promoting robustness in MARL against environmental uncertainty can be approached in various ways. This work focuses on distributionally robust Markov games (RMGs) due to their popularity in both theory and practice. In the context of RMGs, the uncertainty set is a necessary component and plays a central role, otherwise not.
> > > * **Practical uncertainty sets must be task-specific (heuristic or learning-based approaches are both viable).** As promoting robustness in MARL is still an emerging area with limited literature, we draw insights from robust single-agent RL studies. In practice, different tasks require tailored uncertainty sets that balance capturing potential uncertainty without being overly conservative. Such uncertainty sets can be constructed by:
> > >     * Using heuristic uncertianty sets: Adjust the uncertainty level and divergence functions (e.g., f-divergence, Wasserstein distance).
> > >     * Learning from available data: Identify influential parameters from data and construct a "learning-based" uncertainty set by applying data augmentation, shifting these parameters as needed [1].
> > >
> > > > [1] Ding, Wenhao, et al. "Seeing is not believing: Robust reinforcement learning against spurious correlation." Advances in Neural Information Processing Systems 36 (2024).

---

> > > > ### Author Response · Authors · 2024-11-21
> > > > **Reply to Reviewer rZ7M: Part Four**
> > > >
> > > > ### Q8. It would be very helpful if the authors can provide short proof sketches or key proof ideas
> > > >
> > > >
> > > > Thank you for your valuable feedback. As the reviewer suggestesd, we have added a proof sketch in the revised version of our paper, while is currently in Appendix B due to limited space. We will merge the proof sketch into the main context in the final version.
> > > >
> > > > ### Proof Sketch
> > > >
> > > > 1. **Step 1: decompose regret.** We first decompose the regret with the following equation.
> > > >    $$\mathbb{E}\_{\pi \sim \widehat{\xi}}[V\_{i,h}^{\star, \pi_{-i}}] - \mathbb{E}\_{\pi \sim \widehat{\xi}} [V\_{i,h}^{\pi}]
> > > > \leq \underbrace{\mathbb{E}\_{\pi\sim \widehat{\xi}}[V\_{i,h}^{\star,\pi_{-i}}] - \mathbb{E}\_{\pi\sim \widehat{\xi}}[\overline{V}\_{i,h}^{\tilde{\pi}\_i^\star,\pi_{-i}}]}_{A}+ \underbrace{\mathbb{E}\_{\pi\sim \widehat{\xi}}[ \overline{V}\_{i,h}^{\star,\pi\_{-i}}]-\mathbb{E}\_{\pi\sim \widehat{\xi}}[ \overline{V}\_{i,h}^{\pi} ]}\_{B}+ \underbrace{\mathbb{E}\_{\pi\sim \widehat{\xi}}[\overline{V}\_{i,h}^{\pi}] - \mathbb{E}\_{\pi\sim \widehat{\xi}}[V\_{i,h}^{\pi}]}\_{C},$$
> > > >
> > > >
> > > >    where $\mathbb{E}\_{\pi\sim \widehat{\xi}}[\overline{V}\_{i,h}^{\tilde{\pi}\_i^\star,\pi\_{-i}}]$ and $\mathbb{E}\_{\pi\sim \widehat{\xi}}[\overline{V}\_{i,h}^{\pi}]$ denote the empirical estimation of $\mathbb{E}\_{\pi\sim \widehat{\xi}}[V\_{i,h}^{\star,\pi_{-i}}]$ and $\mathbb{E}\_{\pi\sim \widehat{\xi}}[V\_{i,h}^{\pi}]$ correspondingly. $\mathbb{E}\_{\pi\sim \widehat{\xi}}[ \overline{V}\_{i,h}^{\star,\pi\_{-i}}]$ denotes the empirical benchmark value function satisfying $\mathbb{E}\_{\pi\sim \widehat{\xi}}[ \overline{V}\_{i,h}^{\star,\pi\_{-i}}]=\max\_{\pi_i^\prime}\mathbb{E}\_{\pi\sim \widehat{\xi}}[ \overline{V}\_{i,h}^{\pi_i^\prime,\pi_{-i}}]$. We aim to bound the three terms $A$, $B$, $C$ separately.
> > > >
> > > > 2. **Step 2: bound estimation error of term $A$ and term $C$.** To bound the estimated value function and the true value function, it suffices to bound the estimation errors of the transition function and the reward function. Therefore, we aim to establish an upper bound for
> > > >    $$
> > > >    |\min\_{\mathcal{P} \in \mathcal{U}(P\_{i,h,s,a_i}^{\pi_{-i}^k})} \mathcal{P}V - \min_{\mathcal{P} \in \mathcal{U}(\widehat{P}\_{i,h,s,a_i}^{\pi_{-i}^k})} \mathcal{P}V|
> > > >    $$
> > > >    for all value vectors $V: \mathcal{S} \to [0,H]$, as well as
> > > >    $$
> > > >    |r\_{i,h}^k(s,a_i) - \mathbb{E}\_{a_{-i} \sim \pi_{-i}^k} r_{i,h}(s, \mathbf{a})|.
> > > >    $$
> > > >    With these bounds on the estimation errors of the reward and transition functions, we can derive upper bounds for $A$ and $C$.
> > > >
> > > > 3. **Step 3: apply FTRL to bound term $B$.** Initially, we apply the results from FTRL to prove that
> > > >    $$\textstyle\max\_{a\_i\in\mathcal{A}\_i}\sum\_{k=1}^K\alpha_k^K\Big[ r\_{i,h}^k(s,a_i)+\inf\_{\mathcal{P}\in\mathcal{U}^{\sigma\_i}(P\_{i,h,s,a_i}^k)}\mathcal{P}\widehat{V}\_{i,h+1}\Big]-\textstyle\sum\_{k=1}^K\alpha_k^K\mathbb{E}\_{a_i\sim\pi\_{i,h}^k(s)}\Big[r\_{i,h}^k(s,a_i)+\inf\_{\mathcal{P}\in\mathcal{U}^{\sigma_i}(P\_{i,h,s,a_i}^k)}\mathcal{P}\widehat{V}\_{i,h+1}\Big]\leq \beta_{i,h}(s).
> > > >    $$
> > > >    We then proceed by induction to prove that $\widehat{V}\_{i,h}(s) \geq \mathbb{E}\_{\pi \sim \widehat{\xi}}[\overline{V}\_{-i}^{\star,\pi}(s)]$. Finally, we further upper-bound $B$ with $\widehat{V}\_{i,h}(s) - \mathbb{E}\_{\pi \sim \widehat{\xi}}[\overline{V}\_{i,h}^{\pi}(s)]$.

---

> ### Author Response · Authors · 2024-11-26
> **Response to Reviewer rZ7M**
>
> We sincerely thank the reviewer for actively engaging in the discussion. We value the reviewer’s perspective and metric and aim to provide additional clarifications to avoid any potential misunderstandings about our work. We warmly welcome further questions and comments.
>
> * Regarding numerical experiments, we plan to include them in a future version. However, consistent with the standard theoretical MARL research pipeline [1-2], that do not include numerical experiments, our main objective is to establish -- from the theoretical perspective -- that indeed it is possible to design a robust MG with a provably efficient algorithm that breaks the curse of multi-agency.
> * As for concrete examples of promoting robustness in MARL systems, there are numerous cases. For example, in economics, robustness (or risk aversion) is commonly sought in scenarios like matching pennies games [7], where uncertainty is modeled as a risk function outside of expected outcomes [4-6]— similar to our proposed uncertainty set. In real-world applications, examples include accounting for environmental uncertainty when defending against illegal fishing (as illustrated in Figure 1 of [6]).
>
> > [1] Jin, Chi, et al. "V-Learning--A Simple, Efficient, Decentralized Algorithm for Multiagent RL." arXiv preprint arXiv:2110.14555 (2021).\
> > [2] Bai, Yu, and Chi Jin. "Provable self-play algorithms for competitive reinforcement learning." International conference on machine learning. PMLR, 2020.\
> >[3] Friedman, Evan, and Felix Mauersberger. "Quantal response equilibrium with symmetry: Representation and applications." Proceedings of the 23rd ACM Conference on Economics and Computation. 2022.\
> >[4] Sandomirskiy, Fedor, et al. "Narrow Framing and Risk in Games." (2024).\
> >[5] Goeree, Jacob K., Charles A. Holt, and Thomas R. Palfrey. "Regular quantal response equilibrium." Experimental economics 8 (2005): 347-367.\
> > [6] Shi, Laixi, et al. "Sample-Efficient Robust Multi-Agent Reinforcement Learning in the Face of Environmental Uncertainty." arXiv preprint arXiv:2404.18909 (2024).\
> > [7] Goeree, Jacob K., Charles A. Holt, and Thomas R. Palfrey. "Risk averse behavior in generalized matching pennies games." Games and Economic Behavior 45.1 (2003): 97-113.

---

### Author Response · Authors · 2024-11-21
**General response**

### Insights of the prooposed uncertainty sets (games), the practical meaning, and the advantages compared to previous uncertainty sets

We appreciate this insightful question, which has been raised by multiple reviewers and greatly contributes to clarifying the motivation and distinctiveness of our work. In response, we have summarized the comparisons between the proposed robust Markov Games (MGs) and prior games with $(s,a)$-rectangular set. **These updates can be found in the revised manuscript (Section 3.1)**.
* The practical meaning of the proposed robust MGs with fictitious uncertainty set **(realistic and natural)**:
    *  **Realistic and predictive of human decisions: inspired by behavioral economics.** It is meaningful to consider a realistic class of robust stochastic games predictive of human decision-making, as modern multi-agent systems often involve people, such as in social media networks and autonomous driving systems. Observations from experimental data in behavioral economics [1-3] reveal that in many games accounting for agents' randomness, people handle other players' uncertainty using a risk metric outside of their expected outcomes (i.e., $\mathbf{Risk}(\mathbb{E}\_{a\_{-i}\in\pi\_{-i}}[u_i(a_i, \mathbf{a_{-i}})])$), rather than taking the expectation of the risk metric over the outcome of each joint action (i.e., $\mathbb{E}\_{a_{-i} \in \pi_{-i}}[\mathbf{Risk}(u_i(a_i, \mathbf{a_{-i}}))]$). Notably, the former approach, which aligns with realistic human behavior, corresponds to our fictitious uncertainty set, whereas the latter corresponds to the traditional $(s,\mathbf{a})$-rectangular uncertainty set.
    *  **A natural viewpoint for an individual: directly adapted from single-agent RL**: When fixing other agents' policy $\pi_{-i}$, from the perspective of each individual agent $i$,  our proposed games with fictitious uncertainty set will degrade to the original single-agent robust RL problem with the widely used $(s,a_i)$-rectangularity condition in the literature [4]. Namely, from any agent $i$'s viewpoint, it has an "overall environment" player that can not only manipulate the environmental dynamics but also other players' policy $\pi_{-i}$. This can't be achievd by the previous $(s,\mathbf{a})$-rectangular uncertainty set.
* **The proposed new games enable tractable sample complexity**: with provable algorithms that overcome the curse of multi-agency. Previous state-of-the-art algorithms for games with $(s,a)$-rectangular sets cannot overcome the curse of multi-agency in sample complexity, as they require samples that grow exponentially with the number of agents. By leveraging the features of the proposed new games, we introduce an algorithm (Robust Q-FTRL) and demonstrate its sample complexity, which successfully breaks the curse of multi-agency for this new class of games. This is the first algorithm to achieve this in the context of general robust Markov games, irrespective of the chosen uncertainty set. This breakthrough is made possible by introducing a new class of robust MGs, combined with a tailored adaptive sampling and online learning procedure, offering a fresh perspective on learning RMGs.


> [1] Friedman, Evan, and Felix Mauersberger. "Quantal response equilibrium with symmetry: Representation and applications." Proceedings of the 23rd ACM Conference on Economics and Computation. 2022. \
> [2] Sandomirskiy, Fedor, et al. "Narrow Framing and Risk in Games." (2024).\
> [3] Goeree, Jacob K., Charles A. Holt, and Thomas R. Palfrey. "Regular quantal response equilibrium." Experimental economics 8 (2005): 347-367.\
> [4] Garud N Iyengar. Robust dynamic programming. Mathematics of Operations Research, 30(2): 257–280, 2005.

---

### Meta-Review · Area_Chair_mdfi · 2024-12-23

**Metareview:**

This paper investigates the distributionally robust Markov games (RMGs) settings and focuses on a new uncertainty set called "fictitious uncertainty set". Then, the authors propose an algorithm named Robust-Q-FTRL to find robust approximate CCE and provide theoretical guarantees about the upper bound of sample complexity. The sample complexity shows that it breaks the curse of multi-agency in MARL.

The paper raises novel settings and provides detailed theoretical guarantees. However, as challenged by the reviewers, how the new uncertainty set notion "fictitious uncertainty set" can capture real-world applications is not well justified. I suggest elaborating further on the reasons for "breaking the curse of multi-agency in MARL" and explaining how the new uncertainty set notion relates to applications in the future.

**Additional Comments On Reviewer Discussion:**

* Reviewers rZ7M and BsE9 hope the authors can provide some numerical experiment results to demonstrate better performance. The authors state that conducting experiments is difficult because there are no fair comparisons due to non-identical problems.

* Reviewers DjTn and BsE9 questioned the reason for avoiding the curse of multi-agency. The authors provided explanations during the rebuttal period and added the practical meaning and motivation of the proposed new games with comparisons to the prior works in the new version (Section 3.1).

---

### Decision · Program_Chairs · 2025-01-22

Reject